# Controlled beat-wave Brillouin scattering in the ionosphere

B. Eliasson [1 ✉], A. Senior [2], M. Rietveld[3,7], A. D. R. Phelps [1], R. A. Cairns [4], K. Ronald[1], D. C. Speirs[1], R. M. G. M. Trines [5], I. McCrea[5], R. Bamford[5], J. T. Mendonça [6] & R. Bingham [1,5]

Stimulated Brillouin scattering experiments in the ionospheric plasma using a single electromagnetic pump wave have previously been observed to generate an electromagnetic sideband wave, emitted by the plasma, together with an ion- acoustic wave. Here we report results of a controlled, pump and probe beat-wave driven Brillouin scattering experiment, in which an ion-acoustic wave generated by the beating of electromagnetic pump and probe waves, results in electromagnetic sideband waves that are recorded on the ground. The experiment used the EISCAT facility in northern Norway, which has several high power electromagnetic wave transmitters and receivers in the radio frequency range. An electromagnetic pump consisting of large amplitude radio waves with ordinary (O) or extraordinary (X) mode polarization was injected into the overhead ionosphere, along with a less powerful probe wave, and radio sideband emissions observed on the ground clearly show stimulated Brillouin emissions at frequencies agreeing with, and changing with, the pump and probe frequencies. The experiment was simulated using a numerical full-scale model which clearly supports the interpretation of the experimental results. Such controlled beat-wave experiments demonstrate a way of remotely investigating the ionospheric plasma parameters.

[1] Department of Physics, SUPA, University of Strathclyde, Glasgow G4 0NG, United Kingdom. [2] Independent Researcher, Lancaster, United Kingdom. [3] EISCAT, Ramfjordmoen, N-9027 Ramfjordbotn, Norway. [4] School of Mathematics and Statistics, University of St Andrews, St Andrews KY16 9SS, United Kingdom. [5] STFC Rutherford Appleton Laboratory, Harwell, Oxford, Didcot OX11 0QX, United Kingdom. [6] Instituto Superior Técnico, Av. Rovisco Pais, N° 1, 1049-001 Lisboa, Portugal. [7] Present address: Department of Physics and Technology, UiT, The Arctic University of Norway, 9037 Tromsø, Norway. ✉email: bengt.eliasson@strath.ac.uk

Nonlinear mixing of electromagnetic (EM) waves in plasmas has a wide range of applications in laboratory and ionospheric plasma experiments, including beat-wave-particle acceleration, plasma diagnostics, heating of magnetically confined, and laser plasmas, and beat-wave current drive in magnetic fusion devices[1–6]. The stimulated Brillouin scattering (SBS) instability is of profound importance in laser-plasma interactions, laser-driven inertial confinement fusion schemes, and a variety of optical systems[7–14]. In laser-driven Brillouin amplification[15–18], a relatively long pump pulse interacts with a counter-propagating, frequency-downshifted probe (or seed) pulse to resonantly drive ion-acoustic (IA) waves. Under favorable conditions, wave energy is transferred from the pump to the probe, resulting in an amplification of the probe pulse.

In ionospheric heating experiments[19,20], large-amplitude EM waves in the MHz range are injected into the overhead ionospheric plasma to study nonlinear wave-plasma interactions. Theoretical analysis of SBS interactions in unmagnetized plasma[21–23] and magnetized plasma[24–30] have predicted that such interactions would be possible and that nonlinear sidebands would be observed in the form of stimulated EM emissions (SEE) escaping the ionosphere. The first observations of stimulated Brillouin scattering in ionospheric experiments were reported from the Jicamarca radar and at Arecibo[31,32] when the transmitted frequency $f_0$ was significantly higher than the maximum plasma frequency of the ionosphere, making the ionosphere transparent to the EM wave. More recently, there were also clear observations of SBS during ionospheric heating experiments for $f_0$ below the maximum plasma frequency so that the ionosphere was opaque to the EM wave[33–36]. In addition, experiments have demonstrated scattering of the EM wave off ion cyclotron/Bernstein waves[37–47]. Beat waves are being employed in ionospheric heating experiments to excite low-frequency EM waves in the kilohertz range[48–50], with applications to ionospheric and magnetospheric remote sensing, and long-distance communication.

However, a large-amplitude O mode wave can be anomalously absorbed at the F2 layer[51], even though the plasma is only weakly collisional there (see Supplementary Information). Within a few milliseconds, there is a drop of about 10 dB or more in the reflected power. This is attributed to the fact that the O mode wave first excites short-wavelength electrostatic high-frequency Langmuir waves (electron plasma oscillations) and low-frequency ion density fluctuations via the oscillating two-stream instability (OTSI) and parametric decay instability (PDI)[52,53] near the critical layer where the transmitted frequency equals the electron plasma frequency. The ion fluctuations work as a grating on which the O mode is continuously converted to Langmuir waves, leading to an anomalous resistivity[54] that absorbs the O mode. On a longer timescale of about a second, there is a further drop of 10–15 dB in the reflected power. This absorption is due to magnetic field-aligned striations (density cavities) formed via a thermal instability, on which the O mode is converted to high-frequency upper hybrid waves[55–57] at the upper hybrid layer, a few km below the critical layer. The electrostatic waves remain in the plasma and dissipate their energy by heating the electrons and to a lesser extent the ions. In contrast, the X mode wave is reflected by the ionosphere below these resonant layers and is therefore not absorbed, leading to significant differences with respect to wave absorption, ionospheric turbulence, and electron heating.

Here, we present the results of a controlled SBS beat-wave experiment using the EISCAT facility at Ramfjordmoen, Norway (cf. the map in Fig. 1), on 29 November 2014, and analyse the results by means of numerical simulations. To avoid saturation by the direct signal from the transmitter (if the reception of the probe was to be carried out at Ramfjordmoen using the HF receiver), the remote SEE receiver[36,39] installed at Kroken was instead used to record the signal. As seen in Fig. 1, Kroken is located near Tromsø, about 13 km north-northwest (NNW) of Ramfjordmoen separated by a mountainous region. The purpose of the experiment is to study the beat-wave Brillouin interactions by varying the frequency between an EM pump wave and a frequency downshifted probe wave to resonantly excite ion-acoustic (IA) waves in the ionosphere, employing ordinary (O), and extraordinary (X) mode polarization of the injected EM waves. At the frequencies used in the experiment, the ionosphere may work as a reflector, leading to the resonant interaction between the reflected, descending pump, and probe waves and the ascending waves, to drive IA waves. The IA waves constitute a grating in the ionospheric plasma on which the counter-propagating pump and probe waves are scattered into each other. The different propagation and interaction mechanisms for O and X mode polarizations are investigated with the help of numerical simulations and are found to be important when interpreting the observed signals escaping the plasma.

## Results

**Theory of resonant SBS interaction.** In the present experiment, the three waves taking part in resonant SBS interaction satisfy the frequency and wave vector matching conditions

$$\omega_0 - \omega_1 = \omega_2, \tag{1a}$$

$$\mathbf{k}_0 - \mathbf{k}_1 = \mathbf{k}_2, \tag{1b}$$

where the subscript 0 refers to the transmitted pump EM wave, 1 to the transmitted probe EM wave and 2 to the stimulated IA wave nonlinearly driven in the ionosphere by waves 0 and 1. Here $\omega_0$ and $\omega_1$ are the transmitted angular frequencies ($\omega = 2\pi f$) of the pump and probe waves, and $\mathbf{k}_0$ and $\mathbf{k}_1$ are the wave vectors that the pump and probe waves have as they propagate through the dispersive plasma medium of the ionosphere, and where the respective frequency-wave vector pairs ($\omega_0$, $\mathbf{k}_0$) and ($\omega_1$, $\mathbf{k}_1$) obey the Appleton–Hartree dispersion relation[58] (see Methods Eq. (5)). Through nonlinear wave mixing, a low-frequency IA wave is driven with wave frequency and wave vector ($\omega_2$, $\mathbf{k}_2$), fulfilling the wave matching conditions (1a) and (1b). The controlled IA wave sets up a time-oscillating density grating in the ionospheric plasma, on which the pump and probe can be scattered into each other and into EM waves with different frequencies and wave vectors producing EM sidebands at sum and difference frequencies $\omega_0 \pm \omega_2$ and $\omega_1 \pm \omega_2$. The newly created sidebands can again be scattered by the IA wave to new frequencies $\omega_0 \pm 2\omega_2$ and $\omega_1 \pm 2\omega_2$, and so on, leading to a cascade of sidebands at different frequencies, which can escape the ionosphere as SEE. At *resonant interaction*, the IA wave frequency and wave vector ($\omega_2$, $\mathbf{k}_2$) also approximately obey the IA dispersion relation, which takes place at certain altitudes of the ionosphere (see Supplementary Fig. 5). In previous Brillouin experiments[33–47], only the pump wave 0 was transmitted while wave 1 occurred naturally as SEE escaping the ionosphere and was recorded on the ground as a downshifted sideband. The physical process of natural SBS involves a nonlinear interaction in which the incident (pump) EM wave 0 decays into an electrostatic IA wave 2 and a scattered EM wave 1 that grow via the SBS instability[14,40], where the electrostatic IA wave 2 is initially excited by noise-like fluctuations in the plasma. In this three-wave resonant interaction processes, the wave matching conditions are again satisfied: $\omega_0 = \omega_1 + \omega_2$ and $\mathbf{k}_0 = \mathbf{k}_1 + \mathbf{k}_2$, where $\omega_1$ and $\omega_2$ become complex valued, resulting in exponential growth in a time of the waves during the linear phase of the instability[21–23].

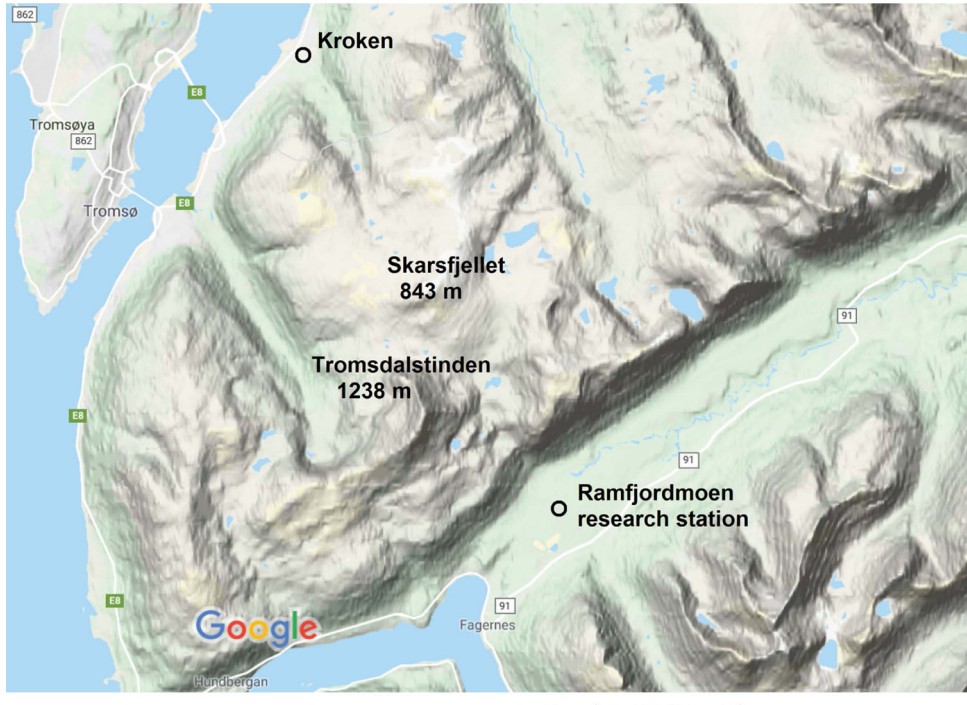

**Fig. 1 The location of the experiment.** The EISCAT heating facility is located at *Ramfjordmoen* and the Lancaster University SEE receiver at *Kroken*, about 13 km NNW of Ramfjordmoen, separated by a mountainous region containing the peaks *Tromsdalstinden* and *Skarsfjellet*. Map data © 2019 Google.

**Table 1 Parameters of the experiment on 29 November 2014.**

| Run | Time On (UT) | Time Off (UT) | #Exp. cycles | Pump $f_0$ (MHz) | Polari- Zation | ERP (MW) | Beamwidth |
|-----|--------------|---------------|--------------|------------------|----------------|----------|-----------|
| I | 10:52 | 11:07 | 45 | 6.300 | O | 554.1 | 6.9° |
| II | 11:10 | 11:15 | 15 | 6.300 | X | 429.8 | 6.6° |
| III | 11:18 | 11:23 | 15 | 6.300 | X | 453.7 | 6.7° |
| IV | 11:45 | 11:55 | 29 | 7.953 | O | 685.7 | 6.3° |
| V | 11:58 | 12:12 | 42 | 7.100 | O | 617.8 | 6.6° |
| VI | 12:15 | 12:30 | 40 | 7.100 | X | 400.2 | 6.8° |

The experiment was conducted with the heater array 1, where transmitters 3-12 were used for the pump and Transmitters 1 and 2 for the probe, with vertical incidence of the beams. The power was 80 kW/transmitter for both the pump and probe, using either O or X polarization. The Effective Radiated Power (ERP) and array beamwidth have been extrapolated for Run I where data was missing.

To a first approximation for long wavelengths, IA waves are dispersion-less and propagate at the phase speed $v_{IA} = C_s$, where $C_s = \sqrt{k_B(T_e + 3T_i)/m_i}$ is the IA speed, $T_e$ and $T_i$ are the electron and ion temperatures, $m_i$ is the ion mass and $k_B$ is Boltzmann's constant. In free space, the EM waves propagate with the phase speed $v_{EM}$ equal to the speed of light $c$, but in the plasma $v_{EM}$ is further increased and rises sharply near the reflection height. Since the IA wave frequency is small compared to the EM wave frequencies, the two EM waves have almost the same frequency and thus propagate at almost the same phase speed (but in opposite directions). Under these assumptions, it can be shown that the IA wave frequency is[14]

$$\omega_2 = (\omega_0 + \omega_1)\frac{v_{IA}}{v_{EM}} \approx 2\omega_0 \frac{v_{IA}}{v_{EM}}, \qquad (2)$$

For typical parameters $T_e = 1500$K, $T_i = 1000$K, $m_i = 16u$ ($O^+$) and a radio frequency of 6 MHz, one finds that the maximum value of the resonant IA wave frequency is about 60 Hz. In the previous heating experiments, the observed SBS lines in the SEE spectrum were typically at a frequency a few

times less than the maximum IA frequency, demonstrating the increase in $v_{EM}$ close to the reflection height of the EM wave.

**Experimental setup.** The parameters used in the experiment are summarized in Table 1. The experiment was carried out employing EISCAT's antenna array 1, encompassing $12 \times 12$ crossed dipoles fed by 12 transmitters producing 80 kW power each. Transmitters 1 and 2, connected to the first and second rows of antennas, were used for the probe signal, while the transmitters 3–12 were used for the pump. The experiment was divided into 6 runs using frequencies 6.3, 7.1, and 7.953 MHz, with the pump and probe beams either X or O mode polarized. In Tromsø, the geomagnetic field is directed almost vertically downwards, and with respect to the vertical, the X, or O mode waves are left-hand and right-hand circularly polarized, respectively. Each run consisted of a number of experimental cycles (see Supplementary Figures 7–9) of 20 seconds each. As detailed in Table 2, the pump was within each experimental cycle switched on for 10 s and then off for 10 s, and the probe wave was stepped down in frequency during the first 9 s and then switched off for the last 11 s. The transmitted frequencies were chosen far enough

from the electron gyroharmonics (cf. Supplementary Figure 2) such that gyroharmonic effects can be neglected. The driven IA waves also have frequencies far enough from ion gyroharmoics and propagate at small angles to the magnetic field so that magnetic field effects can be neglected for the IA waves (cf. Eq. (S5) of the Supplementary Information).

**Experimental observations of SBS interaction**. Figure 2 shows power spectrograms (see Methods) in logarithmic scale obtained in the experiment Runs I–VI averaged over the available experimental cycles (shown Supplementary Figs 7–9). Visible in the spectrograms are the reflected pump wave, and reflected probe at a negative frequency shift $\Delta f$ (cf. Table 2) relative to the

| Time (s) | Pump | Probe $\Delta f$ (Hz) |
|---|---|---|
| 0–1 | On | −1 |
| 1–2 | On | −2 |
| 2–3 | On | −4 |
| 3–4 | On | −6 |
| 4–5 | On | −8 |
| 5–6 | On | −10 |
| 6–7 | On | −14 |
| 7–8 | On | −18 |
| 8–9 | On | −100 |
| 9–10 | On | Off |
| 10–20 | Off | Off |

**Table 2 Transmitter experimental cycle.**

The pump is on 10 s and then off 10 s at a fixed frequency (cf. Table 1), and the probe is stepped down in frequency by a shift $\Delta f$ relative to the pump frequency $f_0$ (cf. Table 1) during the first 9 seconds and is then off for the following 11 seconds.

pump frequency. The probe and pump spectra are partially overlapping during the first few seconds. Also seen in Fig. 2 are downshifted sidebands with frequencies below the probe frequency and upshifted sidebands above the pump frequency, indicating nonlinear wave mixing in the ionosphere generating SEE escaping the ionosphere; see the discussion of Eq. (1a, b) and of the numerical results in Fig. 7 below. Repetition of the frequency spectra at multiples of 50 Hz in Fig. 2 is attributed to weak radiated sidebands at 50 Hz and harmonics thereof because of imperfect power supply filtering. The 7.1 MHz O and X signals in Runs V and VI have continuous frequency downshifted emissions, which are not well understood. A detailed study of the time evolution of the frequency spectrum (Supplementary Figs 7–9) shows that in Run V, in which O mode was used, the downshifted emissions are initiated only near the probe signal and increase in amplitude during the first 300 seconds after which the amplitude remains at the same level, while throughout Run VI, in which X mode was used, the downshifted emissions exist at an almost constant amplitude near both the probe and pump signals. The emissions could potentially result from induced scattering of the pump and probe waves by thermal ions in which the ion-acoustic beat-wave satisfies the Landau resonance where the ion velocities are close to the wave velocity. Another possibility is that the emissions result from the scattering of the probe and pump waves off velocity shear-modified IA waves[59,60] generated by inhomogeneous ion flow along the magnetic field lines. Ion flows may be produced by an increase of the electron pressure[61], by an increase in temperature during O mode heating, or potentially by an increase in plasma density during X mode heating[47]. In addition, there are weaker broadband emissions, manifest in the background signal about 0 dB in Fig. 2 (the blue background) when the pump is on. For O mode polarization, the EM wave reaches

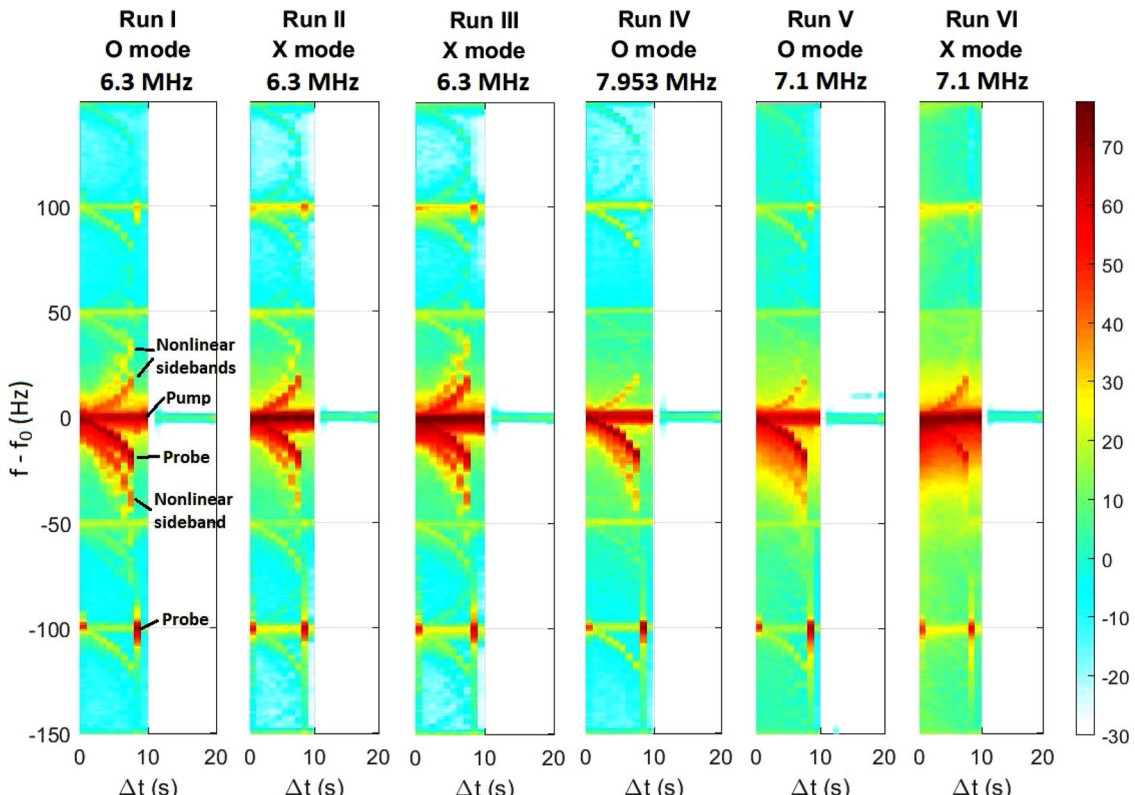

**Fig. 2 Spectrograms (dB) of recorded return signals in the experiment.** The spectrograms are averaged over the experimental cycles (Table 1 and Supplementary Figures 7–9). The pump signal is at $f - f_0 = 0$ and the probe is the first downshifted line at $f - f_0 = \Delta f$ given in Table 2. Visible are several up- and downshifted sidebands attributed to nonlinear wave mixing in the ionosphere producing SEE escaping the ionospheric plasma.

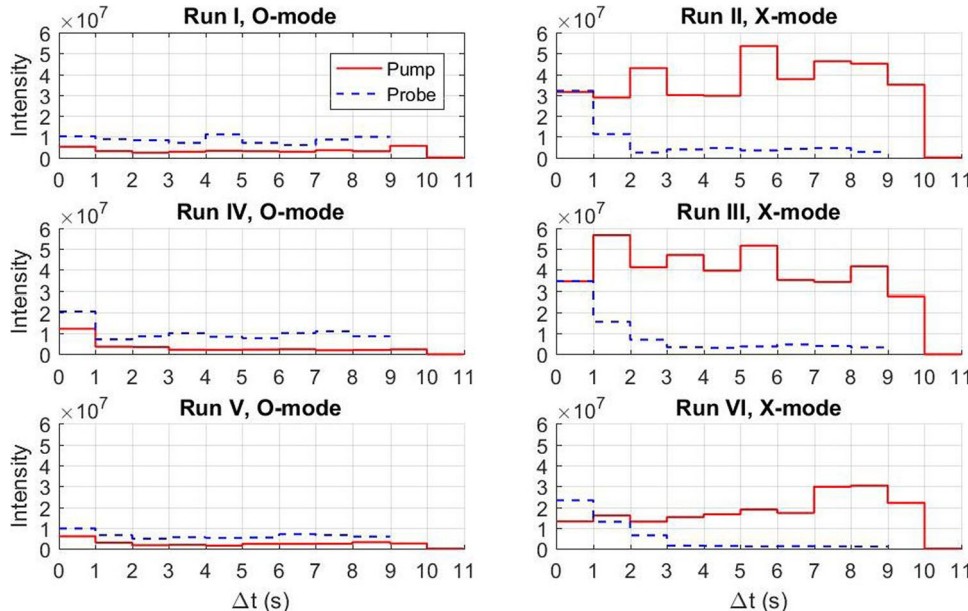

**Fig. 3 Pump and probe intensities.** Intensities (arb. units, linear scale) of the central spectral components of the returned pump and probe signals in Fig. 2 at different times, comparing O (left) and X (right) mode runs. Probe frequencies are given in Table 2.

the critical and upper hybrid layers (cf. Fig. 5 below) where it excites Langmuir and upper hybrid turbulence, which produce low amplitude, broadband emissions tens of kHz around the transmitted frequency[19]. With X mode polarization there is a possibility of 'leakage' of O modes by imperfect polarization by the emitting antenna, leading to Langmuir turbulence near the critical layer. These emissions may give rise to the blue background signal in Fig. 2 when the transmitter is on. The pump and upshifted sidebands have visibly lower amplitudes in the O mode Runs I, IV, and V compared to those of the X mode Runs II, III, and VI, indicating that the O mode has been significantly absorbed in the ionosphere.

The absorption of the returned O mode pump signals is emphasized in Fig. 3, which shows the intensities in the linear scale of the central spectral components of the returned pump and probe waves at different times. The returned O mode pump signals have an order of magnitude lower intensities than the X mode signals, which indicates almost complete conversion of the O mode to electrostatic waves in the ionosphere. Since the transmitted power is kept constant, the variations in time of the returned pump and probe signals are attributed to changing plasma conditions in the ionosphere. In the O mode runs, the pump amplitudes are consistently lower than the probe amplitudes, while the pump amplitudes are larger than the probe amplitudes in the X mode runs. The O mode pump and probe waves that reach the turbulent region are absorbed. However, while most of the well-collimated pump beam reaches the turbulent layer and is absorbed, the probe, being fed by two of the 12 rows in the antenna array, has a wider beam and can be partially reflected by the un-perturbed ionosphere outside the turbulent region.

**Numerical analysis of experimental results**. The experimental results are analysed numerically by means of ray-tracing and full-wave simulations using ionospheric parameters and profiles (Fig. 4) consistent with the experiment; both simulation models are described in the Methods. The strategy is to use ray-tracing simulations to calculate the local wave vector and corresponding wavelengths of the EM wave at different altitudes, which are used to predict the wave frequencies and Landau damping (described in the Supplementary Information) of ion-acoustic waves at

different altitudes in the full-wave simulations. Short (<1 ms) full-wave simulations in two-dimensional planes at different angles to the magnetic field are used in Fig. 5 to predict the general wave structure before any instability sets in and to compare the wave pattern with ray-tracing simulations showing the ray paths of the EM waves. Much longer simulations (10 s) are for computational efficiency carried out in one dimension, along the vertical axis to investigate the nonlinear coupling to electrostatic waves, first using only pump waves to demonstrate the general differences between O and X mode waves in Fig. 6 and then employing both the probe and pump waves in Fig. 7 to simulate the experiment in Fig. 2. Details of the numerical results are discussed below.

From ionogram data[62], it is estimated that vertically injected O mode waves in the frequency range 6–8 MHz are reflected at altitudes of 215–230 km. The geomagnetic field strength $B_0$ at Tromsø[63] is estimated to be $B_0 \approx 48.59\mu T$ at an altitude of 220 km, with the corresponding electron gyrofrequency $f_{ce} \approx$ 1.36 MHz. At Tromsø, the geomagnetic field is directed downwards with an angle to the vertical $\alpha \approx 11.8°$. In the simulations, the transmitted frequency $f_0 = 6.3MHz$ is used for the injected O and X mode waves. To represent the bottom-side ionosphere, the ionospheric density profile (Fig. 4a) is set to a Gaussian, $n_{i0}(z) = n_{0,max}\exp\left[-(z - z_{max})^2/L_{n0}^2\right]$, where $n_{0,max} = 1.048\times 10^{12}$ m$^{-3}$ is the number density at the F2 peak located at $z_{max} = 252.875km$, and $L_{n0} = 63.25km$ is the ionospheric length scale. The critical layer (reflection point) of the O mode wave, $X = 1$, with $X = f_{pe}^2/f_0^2$ and $f_{pe}$ being the electron plasma frequency, is at $z = 219.91km$, while the upper hybrid layer $X + Y^2 = 1$ with $Y = f_{ce}/f_0 = 0.216$ is at $z = 218.20km$. The critical layer of the X mode wave, $X + Y = 1$, is at $z = 211.68km$, and of the Z mode (slow X mode) wave, $X - Y = 1$, is at $z = 227.55km$. Figure 4b shows ray paths of vertically injected O and X mode waves of frequency 6.3 MHz, where the rays reach their respective critical layer and are descending along the same tracks as the ascending rays.

The profiles of the estimated electron temperature are shown in Fig. 4c for O and X mode polarization of the injected EM wave. The electron temperature is modified via Ohmic heating and via wave-particle interaction. It has been observed[64] that ionospheric

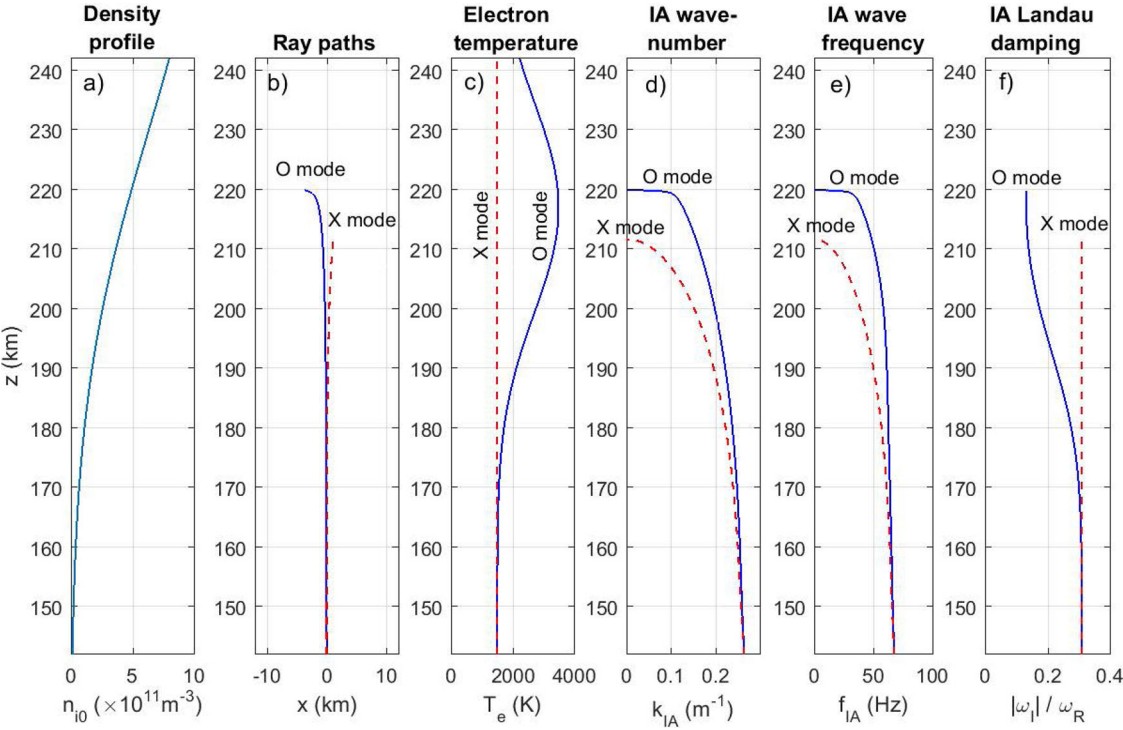

**Fig. 4 Ionospheric profiles. a** Ionospheric density profile, **b** ray paths in the magnetic meridian (*x-z*) plane of O and X mode waves of frequency 6.3 MHz injected vertically, **c** the profiles of the estimated electron temperatures during X and O mode heating (the ion temperature is $T_i = 1000$K), **d** the estimated resonant ion-acoustic (IA) wavenumber $k_{IA} = 2k_{EM,z}$, **e** the resonant IA wave frequency $f_{IA} = \omega_R/2\pi$, and **f** the relative ion Landau damping rate $|\omega_I/\omega_R|$.

heating experiments using vertically injected X mode waves lead only to a moderate temperature increase; here a constant electron temperature $T_e = 1500$K is used. On the other hand, during O mode pumping of the ionosphere[65] the electrons are subject to significant heating, with a temperature increase of a few thousand Kelvin in a localized region around the upper hybrid and critical layers where the O mode is absorbed. The electrons are assumed to have a temperature profile (in Kelvin) of the form

$$T_e = 1500 + 2000\exp\left[-\left(z - z_{Te,max}\right)^2/L_{Te}^2\right], \quad \text{where} \quad z_{Te,max} =$$

217km and $L_{Te} = 25$km. The ions are not significantly affected, and are kept at a constant temperature $T_i = 1000$K for all cases.

The ion-acoustic (IA) waves are for simplicity assumed to propagate vertically, with a resonant wavenumber twice the *z*-component of the EM wave vector, $k_{IA} = 2k_{EM,z}$ (Fig. 4d). To incorporate collisionless ion Landau damping effects due to resonant interaction between the IA wave and particles, we use a kinetic model based on linear Vlasov theory (described in the Supplementary Information) to obtain the damping rate for different values of the IA wavenumber. The resonant IA wave frequency at different altitudes is plotted in Fig. 4e. The IA frequency approximately follows fluid theory, $2\pi f_{IA} = \omega_{IA} \approx C_s k_{IA}$ (magnetic field effects on the IA waves have been neglected; see the Supplementary Information). The resonant interaction occurs at an altitude where $-\Delta f = f_{IA}$, where $f_{IA}$ (Fig. 4e) decreases continuously from about 65 Hz at low altitudes, approaching free space conditions, and tending to zero at the reflection heights of the O and X mode waves. For the O mode wave, $f_{IA}$ decreases sharply from about 30 Hz to zero close to the critical layer. At small values of $\Delta f$ the interaction region is pushed very close to the critical layer, and the interaction layer becomes thin because of the rapidly evolving phase speed of the EM wave with altitude.

The ionospheric profiles in Fig. 4 are used in full-wave simulations using a two-timescale Zakharov model, described in the Methods,

which governs the propagation and coupling of EM waves to electrostatic waves in spatially varying magnetized plasma. An effective radiated power (ERP) of 580 MW provided by the EISCAT array 1 (cf. Table 1) is consistent with a free space amplitude of $E = 0.6$V/m at the altitude $z = 220$ km, using ERP $= 4\pi z^2 I_{max}$ for a uniformly radiating source where $I_{max} = \epsilon_0 E^2 c$ is the intensity in the center of the beam and $E$ is the wave electric field amplitude for a circularly polarized wave. Solving for $E$ gives[66] $E(\text{V/m}) = (1/\sqrt{4\pi\epsilon_0 c})\sqrt{\text{ERP(MW)}}/z(\text{km}) \approx 5.5\sqrt{\text{ERP(MW)}}/z(\text{km})$.

Figure 5 shows the root mean square (RMS) amplitude of the electric field of Gaussian X and O mode beams of frequency 6.3 MHz and beam width 6.9° (cf. Table 1) corresponding to a waist radius of 126 m at $z = 0$. Solid lines show ray paths of EM waves injected from $x = y = z = 0$ with different angles to the vertical, in the magnetic meridian (*x-z*) plane (Fig. 5a, b) and in the *y-z* plane (Fig. 5c, d). In the *x-z* plane, O mode rays injected in the Spitze region[52,67] $-\theta_c < \theta < \theta_c$, where $\theta_c = \arcsin\left[\sqrt{Y/(Y+1)}\sin\alpha\right] = 4.94°$, reach the critical layer $X = 1$. (The German term 'Spitze' refers to the peaked shape of the ray paths in this region[68].) Near the critical layer, the O mode can excite Langmuir turbulence[52] in which small-scale Langmuir waves at frequencies near the plasma frequency are nonlinearly coupled to ion-acoustic waves. A small amount of O mode power will be converted to Z mode waves in a narrow cone around $\theta = +\theta_c$ and to electrostatic Langmuir waves near $\theta = -\theta_c$ in the magnetic meridian plane, as described by Eq. (18) in Ref. [67] (see a detailed discussion in the Supplementary Information). Most of the O mode beam reaches the critical layer in the *x-z* plane. However, in the *y-z* plane, only the vertical beam reaches the critical layer, while oblique beams are reflected at lower altitudes. The O mode RMS amplitude increases to more than 10 V/m near the reflection altitude, while the X mode in Fig. 5a, c is reflected at lower altitudes, below both the critical and upper hybrid layers with a maximum amplitude of about 4 V/m.

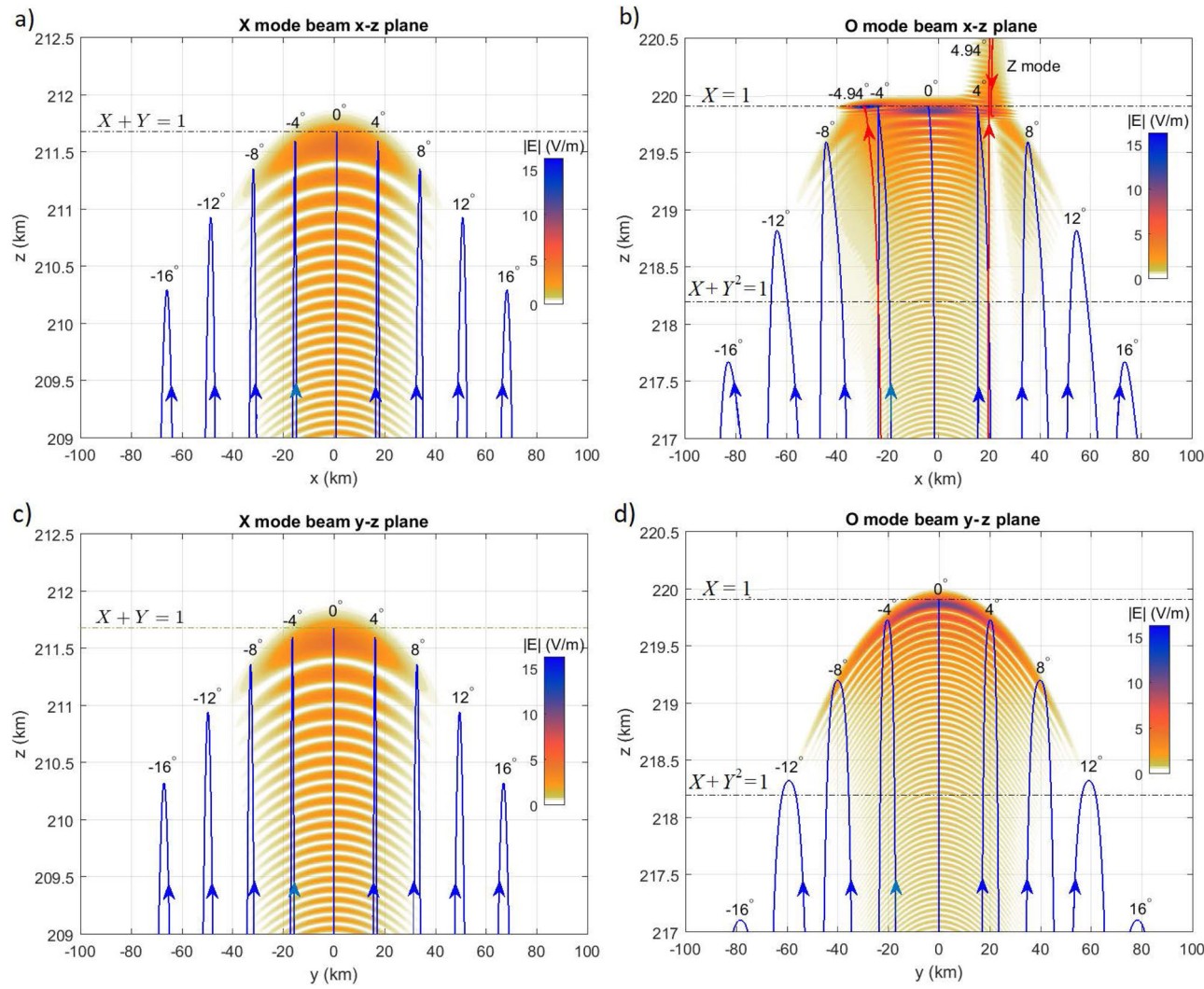

**Fig. 5 Wave structures near X and O mode reflection points.** Vertically injected **a, c** X mode and **b, d** O mode beams of frequency 6.3 MHz and width 6.9° (cf. Table 1), **a, b** in the magnetic meridian (x-z) plane including the magnetic field and **c, d** in the y-z plane, at t<1 ms, before the onset of parametric instabilities. Solid lines show ray paths at different incidence angles $\theta$. Red lines in panel (**b**) indicate rays injected at the Spitze angle +4.94° where the O mode is absorbed and converted to Z mode (slow X mode) waves and at −4.94° to electrostatic Langmuir waves. Horizontal dash-dotted lines show the altitudes of the X mode cutoff $X + Y = 1$ at z = 211.68km, the critical layer $X = 1$ at z = 219.91km, and the upper hybrid resonance $X + Y^2 = 1$ at z = 218.20km. The color scale shows the RMS magnitude |**E**| of the wave electric field. The vertical and horizontal axes are set to different scales for clarity.

Since the EM wave has a maximum amplitude in the central part of the beam (cf. Figure 5), it is anticipated that maximum nonlinear interaction takes place along the vertical axis. A set of one-dimensional simulations of vertically injected plane waves are carried out over much longer times to study the nonlinear coupling of the EM wave to the ions. Figure 6 shows the amplitudes at $t = 10$ s of X and O mode waves of frequency 6.3 MHz and that in free space would have an amplitude of $E = 0.6$V/m (i.e., the injected amplitude). The X mode (Fig. 6a) reaches its cutoff at $z = 211.68$km, and the O mode (Fig. 6b) reaches the altitude $z = 219.91$km. The X mode is reflected and sets up an oscillatory standing wave pattern in $E_x$ and $E_y$ below its cutoff. The ponderomotive force of the X mode standing wave (Fig. 6a) creates a grating in the ion density perturbation $n_{i1}$. The O mode (Fig. 6b) interacts via the oscillating two-stream instability (OTSI) and parametric decay instability (PDI)[52,53] near the critical altitude to excite small-scale electrostatic high-frequency Langmuir oscillations in $E_z$ and low-frequency ion density fluctuations $n_{i1}$. In the nonlinear phase of the instabilities, the ponderomotive force of the high-frequency Langmuir waves

pushes the electrons away from regions of large-amplitude electric fields, pulling the ions with them, resulting in small-scale (submeter) ion density cavities that trap Langmuir waves resulting in strong Langmuir turbulence. The Langmuir turbulence develops within a few milliseconds, and the generated IA fluctuations have typical frequencies in the tens of kHz range, much higher than the tens of Hz oscillations in SBS interactions. In this process, the up-going O mode is anomalously absorbed (see Supplementary Figure 6), resulting in a much weaker reflected wave and no visible oscillating standing wave pattern in $E_x$ and $E_y$ below the cutoff. The Z mode wave has an electrostatic resonance located slightly below the O mode cutoff[67,69] (see the Supplementary Information) where strong Langmuir turbulence takes place. An upward propagating electrostatic Langmuir wave becomes increasingly electromagnetically polarized as it passes the Z mode resonance, and can propagate into the denser plasma in the form of a Z mode. As seen in Fig. 6b, the lower amplitude Z mode wave propagates above the O mode cutoff up to the Z mode cutoff at $z = 227.55$ km where it is reflected, setting up an Airy-like standing wave pattern below its reflection point with swelling

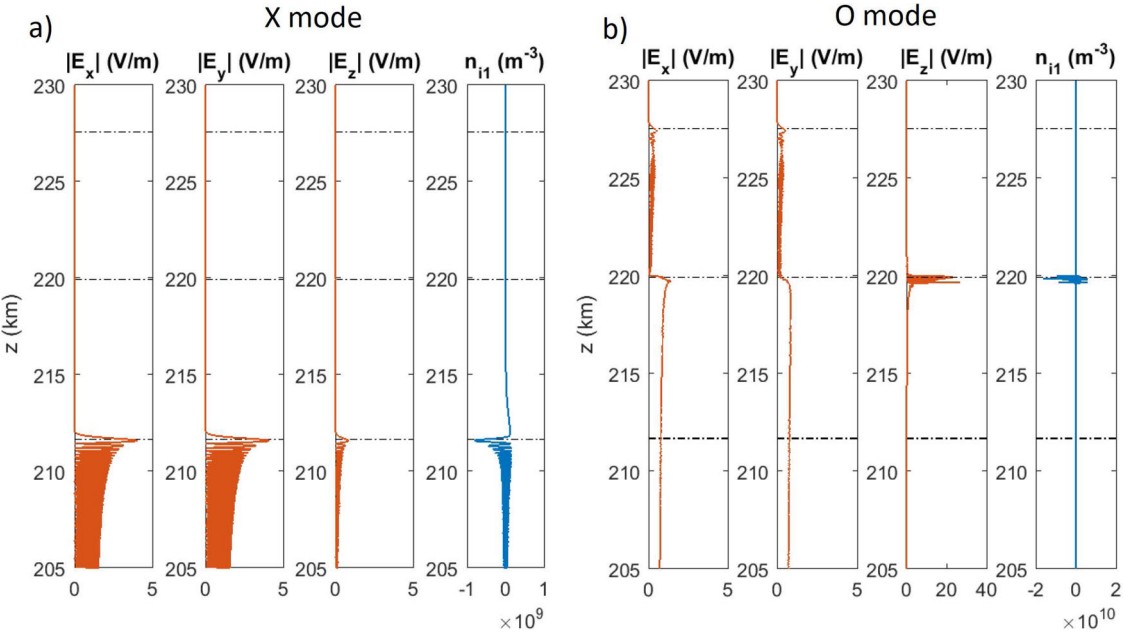

**Fig. 6 Structure of wave electric field and ion density fluctuations at $t = 10$ s.** One-dimensional simulation results at 10 s for transmitted waves having **a** X mode and **b** O mode polarization with frequency 6.3 MHz and injected amplitude 0.6 V/m at the bottom boundary. Shown are the amplitudes of the $x$, $y$ and $z$ components of the electric field, and the ion density fluctuations $n_{i1}$. Horizontal dash-dotted lines indicate the cutoff altitudes of the X, O, and Z mode waves at $z = 211.68$ km, 219.91 km and 227.55 km, respectively.

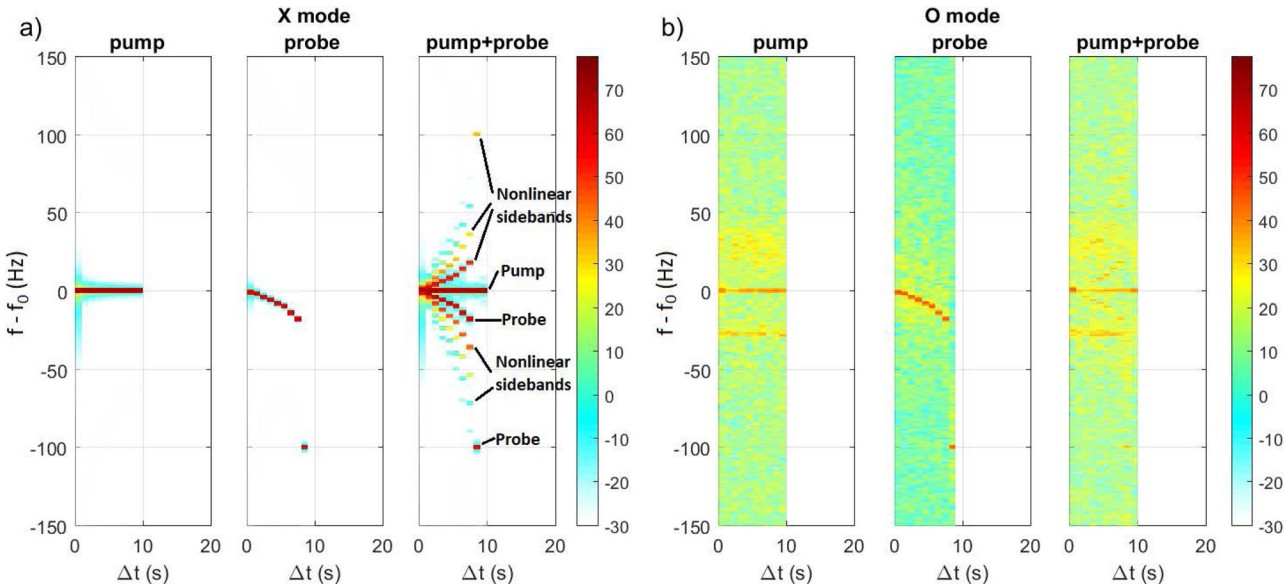

**Fig. 7 Spectrograms obtained in full-wave simulations.** One simulated cycle of **a** X mode and **b** O mode injected vertically. Each panel shows pump only with input amplitude 0.6 V/m and frequency $f_0 = 6.3$ MHz, probe only with input amplitude 0.2 V/m, and frequency offsets relative to the pump given in Table 2, and both probe and pump. The latter shows a rich spectrum of nonlinear sidebands that escape the plasma as SEE.

of the amplitude and increase of the wavelength. At lower altitudes, the wavelength decreases and the Z mode electric field becomes increasingly electrostatically polarized along the $z$ direction as it approaches its electrostatic resonance.

The second set of full-wave simulations are carried out to investigate beat-wave drive SBS interaction, with a vertically injected pump wave of amplitude 0.6 V/m and frequency of 6.3 MHz and a probe wave of amplitude 0.2 V/m with its frequency stepped down compared to that of the pump following the experimental scheme in Table 2. The returning wave electric field reaching the bottom boundary in the simulations is Fourier analysed in time (see Methods) to produce the spectrograms in

Fig. 7, to be compared to the experimental spectrograms in Fig. 2. The electric field of the injected, upward propagating wave is eliminated from the data.

For X mode polarization in Fig. 7a, the EM wave is reflected at $z = 211.68$ km (cf. Figure 5a, c), below the altitudes for resonant coupling to Langmuir and upper hybrid waves. The pump and probe waves individually do not give rise to instability, and hence there are no visible nonlinear sidebands in the corresponding spectrograms. However, for the case of superimposed pump and probe waves, there is a rich spectrum of nonlinear sidebands with several down- and upshifted lines, similar to those of the experimental spectrograms in Fig. 2. The probe and pump waves generate an IA wave with

frequencies and wave vectors obeying the matching conditions in Eq. (1a, b). The IA wave constitutes a time-oscillating density grating in the ionospheric plasma (see Supplementary Figure 5), which can scatter the probe and pump waves into EM sidebands with new frequencies. The newly created EM sideband can again be scattered by the IA wave, leading to a sequence of up- and downshifted nonlinear sidebands at different frequencies escaping the plasma as SEE as seen in Fig. 7a. For O mode polarisation in Fig. 7b, the EM wave reaches the critical layer at $z = 219.91$ km where Langmuir turbulence is excited, leading to a significant widening of the frequency spectrum seen in yellow and green in Fig. 7b, and to anomalous absorption of the probe and pump waves; more details on these processes are given in the Supplementary Information. The broad frequency spectrum of waves seen also during X mode polarization in the experiment (Fig. 2) may be due to the 'leaking' of O mode waves up to the critical layer due to imperfect polarization of the wave by the antenna. The simulation gives more control over the polarization, and when emitting a pure X mode the wave is reflected below the resonant layer for Langmuir turbulence, and therefore no broadband emission is seen in Fig. 7a. It should be noted that the threshold for exciting Langmuir turbulence is very low, and as seen in Fig. 7b (middle panel) the low-amplitude O mode probe wave alone is sufficient to excite Langmuir turbulence and broaden the frequency spectrum.

The returned O mode probe signal is notably stronger than the pump, due to the smaller amount of Langmuir turbulence generated by the weaker probe wave. The combined pump + probe in Fig. 7b are significantly absorbed, and only weak traces of returned pump and probe signals are seen in the spectrogram. For the cases involving the O mode pump wave, there is also a narrow downshifted line at $\Delta f \approx -25$ Hz, indicating that the amplitude of the pump exceeds the threshold for SBS generation, while the probe by itself is below the threshold for natural SBS. At an IA frequency of $f_{IA} = 25$ Hz the resonant interaction takes place very close to the O mode reflection altitude $z = 219.91$ km (cf. Figure 4e) where the wave electric field has a maximum amplitude (Figs. 5b and 6b) leading to the SBS instability discussed below Eq. (1a, b). The diffuse, upshifted lines at $20 - 40$ Hz above the transmitted frequency of the O mode pump may be interpreted as sidebands due to the scattering of the O mode wave off turbulence-generated ion-acoustic waves propagating downwards slightly below the critical layer producing a Doppler upshifted scattered signal. A modulational instability may also take place near the O mode critical layer[70]. The downshifted line is a signature of enhanced natural Brillouin interaction enabled by the locally decreased ion Landau damping. In an earlier experiment at EISCAT[35], natural SBS was observed in SEE spectra at $-8$ Hz and $+12$ Hz when injecting the O mode wave parallel to the magnetic field (magnetic zenith) with frequencies close to the third electron gyroharmonic, $3f_{ce} = 4.04$ MHz. The pumping near gyroharmonics suppresses the formation of small-scale field-aligned striations[71], allowing the O mode to propagate unattenuated to higher altitudes to exceed the threshold for SBS near the critical layer. The present experiment was conducted with frequencies away from the electron gyroharmonics, which may have resulted in the production of field-aligned striations, leading to anomalous absorption of the O mode wave at the upper hybrid layer[55–57]. The resulting decrease of the O mode pump amplitude below the threshold for SBS instability at higher altitudes near the critical layer may explain the absence of natural SBS lines in the experimental spectrograms in Fig. 2 for O mode polarization (Runs I, IV, and V). The 1D numerical model neglects the effect of field-aligned striations, and the unattenuated O mode pump wave is allowed to reach the critical layer where its amplitude exceeds the threshold for natural SBS generation, resulting in the lines in Fig. 7b.

The conversion of O mode waves to electrostatic upper hybrid and/or Langmuir waves may explain the relatively low amplitude of the returned O mode waves in Figs. 2, 3, and 7b. The returned O mode signal in Fig. 7b is visibly weaker compared to that in the experimental spectrograms for O mode polarization (Runs I, IV, and V) in Fig. 2. The difference between experiment and simulation for O mode polarization may be understood by considering that in the experiment the returned signal was recorded at Kroken, 13 km NNW of the transmitter at Ramfjordmoen (cf. Figure 1). As seen in Figs. 5b and 5d, O mode rays at oblique angles to the vertical can have their turning points at lower altitudes, below both the critical layer ($X = 1$) and the upper hybrid layer ($X + Y^2 = 1$). Consequently, these oblique waves are subject to less significant anomalous absorption, which may explain the stronger return signal of the O mode wave observed in the experiment compared to that of the simulation.

## Discussion

Beat-wave heating is an efficient means of injecting energy into plasma at low frequencies that would otherwise be out of reach for EM waves, and controlled beat-wave experiments may help to develop new and improved remote diagnostics of ionospheric plasma parameters. The present study shows that beat-wave driven Brillouin scattering was achieved at EISCAT using vertically injected electromagnetic (EM) waves with X and O mode polarization in the frequency range 6.3–7.9 MHz. The experiment and supporting simulations show clear signatures of nonlinear frequency sidebands attributed to the controlled excitation of low-frequency ion-acoustic waves with frequencies in the 10 Hz range by the nonlinear wave mixing of the pump and probe waves. The O mode is significantly absorbed by the ionosphere, attributed to conversion to electrostatic upper hybrid waves on magnetic field-aligned striations at the upper hybrid layer and to Langmuir waves on ion fluctuations at the critical layer (see Supplementary Information). An interesting generalization of the present investigation is to study beat-wave interaction at different injection angles of the transmitted beam to the ambient magnetic field, and at a wider range of frequency offsets of the probe wave relative to the pump. It would be of interest to compare beat-wave Brillouin interactions at mid and low latitudes with EISCAT and HAARP. An HF facility at the Jicamarca Radio Observatory, located at the geomagnetic equator, has been proposed in the past; experiments may be possible in the near future at a restored Arecibo HF facility in Puerto Rico, where the magnetic field is directed about 45° to the vertical[72]; and related experiments might also be done at the Sura HF facility in Russia, where the magnetic field is about 30° to the vertical[73,74]. The equatorial region has unique features that distinguish it from the midlatitude and auroral regions with a potential for interesting experiments involving Langmuir turbulence-driven suprathermal electrons and associated currents and artificial plasma layers along the magnetic field lines in the horizontal direction[53], and where the geometry may enable beat-wave Brillouin-driven ion Bernstein waves in the vertical direction. Observations of natural SBS are found to originate primarily from two regions: near the turning point of the O mode wave close to the critical layer and near the upper hybrid layer[33], denoted SBS-1 and SBS-2, respectively[34]. It has been found in experiments[36,40] that when the transmitter is tuned near an electron gyroharmonic, SBS-1 strengthens while SBS-2 is suppressed. Therefore it would be of interest to carry out a beat-wave experiment near an electron gyroharmonic to investigate whether significant power can be transferred from the pump to the probe through the SBS-1 process. The ion-acoustic wave is used as part of a diagnostic of the electron temperature in the ionosphere during ionospheric heating experiments[34,36,75], using the dependence of the dispersion of the

mode on the ion-acoustic speed. There are also observations of electromagnetic sidebands where the frequency separation was close to ion gyroharmonics[37–47] which gives information about the local magnetic field and ion composition. Beat-wave excitation of waves at the local electron plasma frequency[4] could also give information about the plasma density. The active control of the wave modes excited in the ionosphere has thus the potential of providing new and improved diagnostics. The ionosphere provides a magnetized plasma medium, which can offer insight into the physics of beat-wave interaction and other nonlinear effects, with relevance to laser-plasma interaction, laboratory plasmas, and inertial and magnetic confinement fusion.

## Methods

**Ray-tracing simulation model.** The ray-tracing model treats the wave as a set of optical beams propagating with the local group velocity of the wave in the spatially varying ionospheric plasma profile. Ray-tracing simulations are used to estimate the propagation paths of X and O mode waves, using the ray-tracing equations[53,76]

$$\frac{d\mathbf{k}}{dt} = -\nabla\omega \quad (3)$$

$$\frac{d\mathbf{r}}{dt} = \nabla_\mathbf{k}\omega \quad (4)$$

where the wave frequency $\omega(\mathbf{k}, z)$ is governed by the Appleton–Hartree dispersion relation based on a cold plasma model[58]

$$\frac{k^2 c^2}{\omega^2} = 1 - \frac{2X(1-X)}{2(1-X) - Y^2\sin^2\Theta \pm Y\Delta} \quad (5)$$

where $\Delta = \left[Y^2\sin^4\theta + 4(1-X)^2\cos^2\Theta\right]^{1/2}$, $X = \omega_{pe}^2/\omega^2$, $Y = \omega_{ce}/\omega$, and $\Theta$ is the angle between the wave vector $\mathbf{k}$ and magnetic field $\mathbf{B}_0$, obtained from the cosine formula $\cos\Theta = (\mathbf{k} \cdot \mathbf{B}_0)/(|\mathbf{k}||\mathbf{B}_0|)$. The upper/lower sign is for O/X mode polarization. Here $c$ is the speed of light in vacuum, $\omega_{pe} = \left(n_{i0}e^2/\epsilon_0 m_e\right)^{1/2}$ is the electron plasma frequency, $\omega_{ce} = eB_0/m_e$ is the electron gyrofrequency, $e$ is the unit charge, $\epsilon_0$ is the electric permittivity of vacuum, $m_e$ is the electron mass, and $n_{i0}(z)$ is the plasma density profile shown in Fig. 4a. The system (3) and (4) has the form of Hamilton's equations if $\omega$ is seen as the Hamiltonian. Equation (3) describes how the wave vector is changing as the wave propagates in the inhomogeneous ionospheric plasma profile, and Eq. (4) describes how a wave packet propagates in space with the local group velocity.

**Full-wave simulation model.** The full-wave model resolves the electromagnetic vector wavefield and plasma response in space and time and includes the nonlinear evolution of the system taking into account finite temperature effects, Landau damping (see the Supplementary Information), and electron-ion collisions. Full-wave simulations are carried out using a Zakharov model[77,78] governing the propagation and coupling of electromagnetic (EM) waves to electrostatic waves in spatially varying magnetized plasma. The Zakharov model is based on a two-time scale model where the high-frequency EM wave field and the fast electron dynamics are represented as $\mathbf{E} = \Re\left[\widetilde{\mathbf{E}}(z,t)e^{-i\omega_0 t}\right]$ and $\mathbf{v} = \Re\left[\widetilde{\mathbf{v}}(z,t)e^{-i\omega_0 t}\right]$ where $\widetilde{\mathbf{E}}$ and $\widetilde{\mathbf{v}}$ are slowly time-varying complex envelopes of the electric field and electron quiver velocity, and $\Re$ denotes the real part. The evolution of the electric field envelope is given by

$$-2i\omega_0\left(\frac{\partial}{\partial t} + \nu_e\right)\widetilde{\mathbf{E}} = -c^2\nabla\times(\nabla\times\widetilde{\mathbf{E}}) + 3\nu_{Te}^2\nabla(\nabla\cdot\widetilde{\mathbf{E}}) - \omega_0^2\left[X(\widetilde{\mathbf{E}} + \widetilde{\mathbf{v}}\times\mathbf{B}_0) - \widetilde{\mathbf{E}}\right] \quad (6)$$

where $X(z,t) = (n_{i0}(z) + n_{i1}(z,t))e^2/\epsilon_0 m_e \omega_0^2$ represents the normalized electron density, $n_{i1}(z,t)$ is the slowly time-varying ion (and electron) number density perturbation, and $\nu_e$ is the damping rate due to electron Landau damping and electron-ion collisions[79], $\nu_{Te} = \sqrt{k_B T_e/m_e}$ is the electron thermal speed, $T_e$ is the electron temperature, $m_e$ is the electron mass, and $k_B$ is Boltzmann's constant. The Lorentz force term is obtained approximately from the electron equation of motion as

$$\widetilde{\mathbf{v}}\times\mathbf{B}_0 = \frac{1}{1-Y^2}\hat{\mathbf{b}}_0\times\left(iY\widetilde{\mathbf{E}} - Y^2\hat{\mathbf{b}}_0\times\widetilde{\mathbf{E}}\right) \quad (7)$$

where $\hat{\mathbf{b}}_0 = \mathbf{B}_0/B_0$ is a unit vector parallel to the magnetic field. Roughly speaking, the first term on the right-hand side of Eq. (6) governs the propagation of EM waves and the second term governs the dynamics of high-frequency electrostatic Langmuir waves, which are linearly coupled through the magnetic field $\mathbf{B}_0$. Variations in the ion density $n_i$ can scatter the high-frequency waves and modify their propagation. To take into account the dynamics of the ions, the system is closed by the IA wave equation

$$\frac{\partial^2 n_{i1}}{\partial t^2} + 2\nu_L\frac{\partial n_{i1}}{\partial t} - C_s^2\frac{\partial^2 n_{i1}}{\partial z^2} = \frac{\omega_{pe}^2}{\omega_0^2}\frac{\epsilon_0}{4m_i}\frac{\partial^2 |\widetilde{\mathbf{E}}|^2}{\partial z^2} \quad (8)$$

where the right-hand side represents the ponderomotive pressure force of the high-frequency electric field, $\nu_L$ is the ion Landau damping operator, $C_s = \sqrt{k_B(T_e + 3T_i)/m_i}$ is the IA speed, $T_i$ is the ion temperature, and $m_i$ is the ion mass. The ponderomotive force pushes the electrons away from regions of large-amplitude oscillatory electric fields, pulling the ions with them via the slowly varying space-charge electric field. The resulting density fluctuations act as a grating to scatter the EM wave, further enforcing the field. Beyond a threshold amplitude, this can give rise to an SBS instability. In the present model, the envelope field is solved using an implicit Crank–Nicholson method, while the IA wave equation is solved with an explicit scheme, so that the time-stepping can be done directly on the slow ion timescale with time-steps exceeding the oscillation period of the high-frequency wave[80,81]. To resolve the small-scale electrostatic waves due to Langmuir turbulence near cutoff for the cases of injected O mode waves, the grid size is locally decreased to 4 cm in the region $z = 219 – 220.5$ km. A time-step of $\Delta t = 10^{-5}$ s is used, to resolve Langmuir turbulence with typical ion oscillation periods of $10^{-3} – 10^{-4}$ s. In Fig. 5, a Fourier method[66,82–84] is used for the horizontal $x$ and $y$ variables, resulting in one-dimensional systems along $z$ for each Fourier mode; the results in physical space are obtained via inverse Fourier transformation. In Fig. 5, the injected O mode wave is modeled as a Gaussian beam resolved by 1200 Fourier modes in either $k_x$ or $k_y$, covering a region $\pm 0.2k_0$ in wavenumber space and $\pm \pi/\Delta k \approx \pm 71$ km in space, where $\Delta k = 2 \times 0.2k_0/1200 \approx 4.4 \times 10^{-5}$ m$^{-1}$ is the wavenumber resolution. A numerical grid of 4 m grid size is used to resolve the solution in the vertical direction covering $142 \le z \le 242$ km. The angular width of the beam is taken to be $\varphi = 6.9° = 0.12$ rad, corresponding to a beam waist radius of $2/k_0\varphi = 126$ m at ground $z = 0$. To carry out 2D or 3D full-scale simulations of the EM wave propagation and Langmuir turbulence over 10 s is not computationally possible with the present simulation methods and resources. Therefore, the longer simulations in Figs. 6 and 7 are carried out in one dimension, where the solution is assumed to vary only in the vertical direction; hence $k_x = k_y = 0$. The ion Landau damping operator $\nu_L$ is approximated via a sum representation in the IA wave equation using the ion Landau damping rates shown in Fig. 4e, while the electron Landau damping in $\nu_e$ is approximated with a diffusion operator[79]. The EM waves are injected at the bottom boundary to propagate into the plasma, while downward propagating waves are absorbed at the bottom boundary via an outflow boundary condition.

**Calculation of spectrograms.** The experimental spectrograms in Fig. 2 (and Supplementary Figures 7–9) and simulated spectrograms in Fig. 7 are obtained by carrying out a Fourier transform of the wave electric field in time over the 1-second windows listed in Table 2 as the probe signal is stepped down in frequency with time. The received time-series signal in the experiment and the simulated signal in the simulation are multiplied by a Hanning (sine-squared) window in time before carrying out the Fourier transforms, in order to increase the dynamic range in the spectrograms. The 1-s time windows lead to a minimum frequency resolution of the spectra which are resolved by steps of 1 Hz, with a small degree of frequency smoothing of the spectrum introduced by the Hanning window.

## Data availability

Spectrogram data in Supplementary Figures 7–9, beam patterns, ionograms taken during the experiment, experimental logs, and data supporting the simulations have been deposited at the University of Strathclyde repository (https://doi.org/10.15129/d4ca75d2-a462-4940-9614-b1b3fe5a8e2e). Ionograms are available at the Tromsø Geophysical Observatory, University of Tromsø, Tromsø, Norway, http://geo.phys.uit.no/ionodata/ionosonde.html.

## Code availability

Analysis codes to analyze spectrogram data files are on the University of Strathclyde repository: https://doi.org/10.15129/d4ca75d2-a462-4940-9614-b1b3fe5a8e2e. The estimated ERPs in Table 1 and the antenna beam patterns plotted in Supplementary Figure 3 were obtained using the 'heatererp' software written by A. Senior with contributions by M. T. Rietveld, freely available at https://gitlab.com/andrewsenior/heatererp.

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

## Acknowledgements

The authors acknowledge support from the Engineering and Physical Sciences Research Council (EPSRC), UK, grants EP/M009386/1, EP/R004773/1, and EP/R034737/1. EIS-CAT is an international association supported by research organisations in China (CRIRP), Finland (SA), Japan (NIPR and ISEE), Norway (NFR), Sweden (VR), and the United Kingdom (UKRI).

## Author contributions

R. Bingham, A.S., M.R., I.M., R. Bamford, R.A.C., R.M.G.M.T. and J.T.M. planned the experiment. M.R., A.S., R. Bingham and I.M. were directly involved in the experimental campaign and collected the data. M.R., R. Bingham, I.M., A.D.R.P., K.R., D.C.S. and B.E. analysed the experimental results. B.E. carried out simulations and wrote the manuscript with input from all authors. R. Bingham supervised the project.

## Competing interests

The authors declare no competing interests.
