## [Peer Review File · Nature Communications]

Controlled beat-wave Brillouin scattering in the ionosphereREVIEWER COMMENTS

Reviewer #1 (Remarks to the Author):

-What are the noteworthy results? – Yes.

- Will the work be of significance to the field and related fields? How does it compare to the established literature? If the work is not original, please provide relevant references. – The authors donw a good job with citing current literature. The only one references missed from their list is a quite relevant review paper by Streltsov et al. [2018], Past, Present and Future of Active Radio Frequency Experiments, Space Sci. Rev. 214:118, doi: 10.1007/s11214-018-0549-7.

- Does the work support the conclusions and claims, or is additional evidence needed? – Yes.

- Are there any flaws in the data analysis, interpretation and conclusions? - Do these prohibit publication or require revision? – No.

- Is the methodology sound? Does the work meet the expected standards in your field? – Yes.

- Is there enough detail provided in the methods for the work to be reproduced? – Yes.

The paper present results from an ionospheric HF heating experiment conducted at EISCAT facility, Northern Norway, in which large amplitude radio waves with ordinary and extraordinary polarization are injected into the ionosphere. The waves reflect at some altitude in the ionosphere and the interactions between counterpropagating radio waves leads to a generation of ion-acoustic waves. The results from the experiment are confirmed by simulations of the full-wave model.

I think that this is a very solid, professional paper, which certainly can be published in some professional journal. I also think that it is a little bit “too professional” to be interesting to general readers of Nature Communication. Probably, it is more suitable for Geophysical Research Letters. Nevertheless, I recommend this paper for publication with only three minor corrections:

1. The list of references does not include all the recent papers related to the generation/usage of beat-waves in the ionospheric experiments. To resolve this issue, I recommend including the reference to Streltsov et al. [2018], Past, Present and Future of Active Radio Frequency Experiments, Space Sci. Rev. 214:118, doi: 10.1007/s11214-018-0549-7.

2. On p. 8 authors use a formula for the magnitude of the electric field in the ionosphere with a constant 6.7. To my best knowledge, in some previous paper regarding experiments at EISCAT this constant was 5.5. Please explain this difference.

3. In the last sentence of the Conclusion section authors claim that “The ionosphere provides a magnetised, nonlinear plasma medium ...”. I think that the word “nonlinear” mostly relates to waves or some interaction processes, but not to the ionosphere. So please remove this word or explain what you mean.

Reviewer #2 (Remarks to the Author):

This is an interesting and important contribution to the field.

- What are the noteworthy results?

As indicated below.

- Will the work be of significance to the field and related fields? How does it compare to the established literature?

Indicated below.

- Does the work support the conclusions and claims, or is additional evidence needed?

Mostly, as indicated below.

- Are there any flaws in the data analysis, interpretation and conclusions? Do these prohibit publication or require revision?

Basically sound, see comments below.

- Is the methodology sound? Does the work meet the expected standards in your field?

Yes.

- Is there enough detail provided in the methods for the work to be reproduced?

Yes.

This is a complex topic and the presentation in the manuscript should be improved.

The title

> Ionospheric beat-wave Brillouin scattering experiment

is very general. The title should better reflect the results, it should point to what is interesting and unique. For example something like

> Controlled beat-wave Brillouin scattering in the ionosphere

Similarly, reading the article I felt a bit lost. As I started reading, I was wondering, what is the point, what is the new thing, what is the strategy?

The authors correctly reference many past SBS results but don't point out clearly enough what is different about their result. Be clear and specific about what filled the role of the probe wave in past experiments and the difference in the experiment being reported here.

Also, what is the experimental set up? It is, finally, explained in the paper, but pump and probe should be put into better context at the outset. Science is like detective work, but this is not a detective story, it's a presentation of results, where clear presentation is key.

The simulation is a key part of the results being presented. What is meant by a full-wave simulation should be explained up front. I don't think it is ever explained in the text.

I would eliminate the current first sentence of the abstract. The second paragraph explains it, and the last sentence of the conclusion also covers it. The abstract could say something like:

> Stimulated Brillouin scattering in the ionospheric plasma has previously been reported under conditions where the EM pump wave causes an EM sideband wave to be emitted by the plasma. Here we report results of controlled, pump and probe beat-wave-driven Brillouin scattering in the ionosphere, in which the EM sideband wave is controlled by the probe wave via an intermediary ion-acoustic wave. In an experiment at EISCAT in northern Norway, a pump consisting of large amplitude radio waves with ordinary (O) and extraordinary (X) mode polarization was injected into the overhead ionosphere, along with a less powerful probe wave, and radio sideband emissions observed on the ground clearly show stimulated Brillouin emissions at frequencies agreeing with, and changing with, the pump and probe frequencies. The experiment was simulated using a numerical radio-plasma model, and clearly verifies the experimental results.

I might have it wrong, please make it right, above and also below.

The simulation is called full wave. What that means should be explained.

On page 3

> The three waves taking part in resonant SBS interaction ...

$$> \omega_1 = \omega_2 + \omega_3 \quad (1a)$$

$$> k_1 = k_2 + k_3 \quad (1b)$$

> where the subscript 1 refers to the pump EM wave, 2 to the probe EM wave and 3 to the stimulated IA wave.

Equations 1a and 1b are confusing. The pump and the probe are inputs to the system. Pedagogically, inputs might be summed on the left, and the output would appear on the right. What is really happening must be

$$\omega_1 - \omega_2 = \omega_3$$

$$k_1 - k_2 = k_3$$

But these equations are not complete. It is not the IA wave (wave 3) that is being observed. The IA wave is translated into an EM wave of the same frequency that shows up as an SEE line, for example in the spectra in figure 2 where the pump and probe are also seen. How the IA wave translates to an EM wave in that spectrum should be explained.

On page 3

> In previous Brillouin experiments [29-31], wave 2 occurred naturally in the plasma as a frequency down-shifted EM sideband that was driven unstable by the large amplitude pump EM wave.

Fu et al. (2015) write

> The physical process of SBS involves a nonlinear interaction in which an incident (pump) electromagnetic wave decays into an electrostatic IA wave and a scattered electromagnetic wave via the Brillouin instability. In such three-wave interaction processes, the wave matching conditions are satisfied: $\omega_0 = \omega_S + \omega_L$ and $k_0 = k_S + k_L$, where ω is the wave frequency; k is the wave propagation vector; and the subscripts 0, S and L denote the pump waves, the scattered waves and low-frequency waves, respectively.

This is clear, wave S (IA) is observed. In this paper this is, I think, called natural SBS. But it is not clear that wave S is the natural analog of the probe wave, since in the equations above S is an output and the probe is an input.

So for natural SBS what is happening is

$$\omega_{\text{pump}} = \omega_{\text{ia}} + \omega_{\text{see}} ; k_{\text{pump}} = k_{\text{ia}} + k_{\text{see}}$$

For controlled pump/probe SBS, maybe what is happening is

$$\omega_{\text{pump}} - \omega_{\text{probe}} = \omega_{\text{ia}} ; k_{\text{pump}} - k_{\text{probe}} = k_{\text{ia}}$$

and

$$\omega_{\text{pump}} \pm \omega_{\text{ia}} = \omega_{\text{see}} ; k_{\text{pump}} \pm k_{\text{ia}} = k_{\text{see}}$$

Natural SBS is mentioned on page 10 (comment below). Also, in figure 7, X mode, pump+probe panel, there is a second, outermost set, of nonlinear sidebands. They might barely appear in figure 2, but maybe not. I don't think those are explained, but they should be. Also, the image of the probe as a nonlinear sideband, in both figures, should be explained.

Basically, a clearer explanation of these processes, and how they relate to each other, is needed.

If these things do not stand out for me, who knows something about this field, it will also not stand out for many other potentially interested readers. All of this may be well understood by specialists and participants, but I think that publication in a Nature journal implies a wider audience, and a need for a more background explanation and clarity than might be needed in a more specialized journal such as JGR or Annales Geophysicae or Radio Science. But I'd probably make the same basic comment, i.e. improve the clarity, in any journal.

In figure 2, why does the background change from blue to white at when the pump is off? Does the pump create that much noise across the band, or is it possibly coming from the receiving system? Or due to the data processing? In figure 2, O and X appear nearly identical in this regard. In figure 7, the same effect occurs in the O mode simulation, but not in the X mode simulation. Why the difference in the simulation?

On page 5

> The 7.1 MHz O and X signals in Runs V and VI have continuous down-shifted emissions, which are not well understood.

Some further comment to explain why not, or what it could be, or how to find out, would be interesting.

On page 5

> While most of the O mode pump is absorbed, the probe, being fed by 2 of the 12 rows in the antenna array, has a wider beam and can be partially reflected by the un-perturbed ionosphere outside the turbulent region.

It should be clarified whether it is meant that the probe, as well as the pump, is absorbed within the turbulent region. It is stated on page 8, but no harm in clarifying it here.

On page 8

> At the Spitz angle 4.94° , the O mode is converted to a Z mode wave that is reflected at a higher altitude and then propagates down to its electrostatic resonance, and at the negative Spitz angle -4.94° the O mode is converted directly to electrostatic waves.

This could be clarified with a few additional words. At the positive Spitze, why does the Z mode not reach its ES resonance on the way up? A reference or references would be good. At the negative Spitze, what ES waves are the O mode converted to? Is there an appropriate reference?

On page 9

> A set of one-dimensional simulations of vertically injected plane waves are carried out over much longer times to study the nonlinear coupling of the EM wave to the ions.

and

> A second set of simulations are carried out to investigate beat-wave drive SBS interaction

It would be helpful, at the beginning of the discussion of the simulations, to explain the overall strategy. What are the goals, what different simulations are used to achieve those goals, in how many dimensions, what time steps and time span, etc. The term "full wave" could be explained here, and whether or not all of the simulations are full wave, and if not what the differences are and why.

A grammatical comment - for the simulations that were done, it makes more sense write "were carried out", instead of "are carried out", since it was done in the past.

On page 10

> For the cases involving the O mode pump wave, there are also frequency up-shifted ion lines at 30 Hz above the transmitted frequency, and a narrow down-shifted line at $Df \sim -25$ Hz, indicating that the amplitude of the pump exceeds the threshold for SBS generation, while the probe by itself is below the threshold for natural SBS.

Are the +30 and -25 both natural SBS? Why these frequencies? This should be made clearer (similarly to the comments above). Be clearer in explaining how the natural SBS works.

On page 10

> The difference between experiment and simulation for O mode polarization may be understood by considering that in the experiment the returned signal was recorded at Kroken, 13 km NNW of the transmitter at Ramfjordmoen (cf. Fig. 1). As seen in Fig. 5, O mode rays at oblique angles to the vertical have the turning point at lower altitudes, below the critical layer. In the y-z plane, only the vertical beam

reaches the critical layer, while oblique beams are reflected at lower altitudes. In this case, the anomalous absorption of the O mode wave may be less significant, leading to a stronger Brillouin interaction between the pump and probe.

Is this meant to say that this may be, or that it is, the reason?

Maybe I missed it, but why not model the true experimental situation to verify if this explanation is correct. I'm guessing the reason is in the paper already, but please add a few words here. Those words could refer to the summary explanation of the simulations and simulation strategy discussed above, and/or also be included there.

What is the predicted result if the measurements are repeated near a gyroharmonic?

In the supplement, page 2,

> Since the down-coming probe wave was not very weak, the modest antenna used by the SEE receiver was deemed adequate.

However, the point of the receiver was not just to receive the probe, but also the SEE, after which the receiver is named. The SEE emission should be mentioned.

Reviewer #3 (Remarks to the Author):

The manuscript describes an interesting experiment on nonlinear wave mixing in the ionosphere in which two EM waves are introduced to produce a beat wave at ion acoustic (IA) frequency, which is observed in the spectrogram. Generally, in ionospheric modification experiments a pump wave is introduced and scattered waves are observed. The authors describe a novel way to introduce two EM waves travelling in opposite directions to form a beat wave. The manuscript is clearly written and relevant to the area of nonlinear plasma physics.

While the novelty of the experiment is noteworthy and its explanation based on parametric processes is a possibility, it could be improved by considering and ruling out other possible mechanisms for the generation of the IA waves.

Normally, the IA waves do not spontaneously grow in ionospheric plasma with $T_i \sim T_e$ since they are heavily ion Landau damped. However, they are quite easily generated if there is an inhomogeneous ion flow along the magnetic field [Gavrishchaka et al., Phys Rev Lett., 80, 728, 1998], which has also been experimentally demonstrated in laboratory [Agrimson et al., Phys Rev Lett., 86, 5282, 2001]. The ponderomotive force generated by the EM waves in the experiment could locally drive an inhomogeneous ion flow along the magnetic field and could generate IA waves. The difference of the classical IA wave with the velocity shear-modified IA (SMIA) wave is that the SMIA waves are oblique and depends on the ratio of perpendicular to parallel wave vectors (k_{\perp}/k_{\parallel}). Without measurements of wave vectors along and across the magnetic field it is difficult to distinguish between the two. These waves would have the same signature in the spectrogram (Fig 2). The up and down shifted IA signature may be generated by different signs of (k_{\perp}/k_{\parallel}) [see dispersion relation given in Gavrishchaka et al.]. The authors should provide arguments and evidence to rule this possibility out.

Since the SMIA waves are kinetic, a PIC simulation with at least 2D is necessary for numerical simulation of SMIA. The simulation model used in the manuscript is 1D and presumably fluid, which will exclude this possibility. While the ponderomotive force of one EM wave may not be sufficient to generate above threshold shear the oppositely generated ion flows by counter propagating pump and probe waves may do the job.

In addition, the manuscript mentions, on page 5 line 9, "The 7.1 MHz O and X signals in Runs V and VI have continuous down-shifted emissions, which are not well understood." This signature could result from induced scattering of the pump wave by thermal ions in which the beat wave satisfies the Landau resonance. Fig. 4e,f show that the Landau resonance is satisfied by the beat wave. In the induced scattering process the plasmon number density is conserved but not in the decay process. The manuscript does not mention if this hypothesis was tested.

Reviewer #1 (Remarks to the Author):

-What are the noteworthy results? – Yes.

- Will the work be of significance to the field and related fields? How does it compare to the established literature? If the work is not original, please provide relevant references. – The authors donw a good job with citing current literature. The only one references missed from their list is a quite relevant review paper by Streltsov et al. [2018], Past, Present and Future of Active Radio Frequency Experiments, Space Sci. Rev. 214:118, doi: 10.1007/s11214-018-0549-7.

Response: We thank Reviewer #1 for pointing out this relevant review paper by Streltsov et al. [2018], which we now cite in the second paragraph of the Introduction as Ref. [18].

- Does the work support the conclusions and claims, or is additional evidence needed? – Yes.

- Are there any flaws in the data analysis, interpretation and conclusions? - Do these prohibit publication or require revision? – No.

- Is the methodology sound? Does the work meet the expected standards in your field? – Yes.

- Is there enough detail provided in the methods for the work to be reproduced? – Yes.

The paper present results from an ionospheric HF heating experiment conducted at EISCAT

facility, Northern Norway, in which large amplitude radio waves with ordinary and extraordinary polarization are injected into the ionosphere. The waves reflect at some altitude in the ionosphere and the interactions between counterpropagating radio waves leads to a generation of ion-acoustic waves. The results from the experiment are confirmed by simulations of the full-wave model.

I think that this is a very solid, professional paper, which certainly can be published in some professional journal. I also think that it is a little bit “too professional” to be interesting to general readers of Nature Communication. Probably, it is more suitable for Geophysical Research Letters. Nevertheless, I recommend this paper for publication with only three minor corrections:

Response: We thank Reviewer #1 for the positive comments and for recommending our manuscript for publication after minor corrections. Nature Communications takes papers from a broad area of science and is appropriate since the paper contains a new application of radio waves in the ionosphere that has not been done before. Nonlinear wave-wave interactions are of broad interest to many areas of science such as laser fusion, plasma accelerators and current-drive and heating of tokamaks using RF power.

1. The list of references does not include all the recent papers related to the generation/usage of beat-waves in the ionospheric experiments. To resolve this issue, I recommend including the reference to Streltsov et al. [2018], Past, Present and Future of Active Radio Frequency Experiments, Space Sci. Rev. 214:118, doi: 10.1007/s11214-018-0549-7.

Response: We have added the review paper by Streltsov et al. [2018] to the Reference list and cite it in the second paragraph of the Introduction (as Ref. [18]) discussing ionospheric heating experiments. Thank you for pointing out this relevant and interesting paper!

2. On p. 8 authors use a formula for the magnitude of the electric field in the ionosphere with a constant 6.7. To my best knowledge, in some previous paper regarding experiments at EISCAT this constant was 5.5. Please explain this difference.

Response: Yes, the numerical factor should be 5.5 (and not 6.7), and we have now corrected this on bottom of page 8, where we write:

“An Effective Radiated Power (ERP) of 580 MW provided by the EISCAT Array 1 (cf. Table 1) is consistent with a free space amplitude of $E = 0.6 \text{ V/m}$ at the altitude $h = 220 \text{ km}$, using that the $E_{max} = 4 \sqrt{P_{max}}$ for a uniformly radiating source where $P_{max} = \frac{1}{2} \epsilon_0 E^2$ is the intensity in the center of the beam and E is the wave electric field amplitude for a circularly polarized wave. Solving for E gives [57] $E (\text{V/m}) =$

$$\sqrt{\frac{4 \sqrt{P_{max}}}{\epsilon_0}} \approx 5.5 \sqrt{P_{max}} \text{ (MW)} \text{ } \vartheta \text{ (km).}”$$

We thank Reviewer #1 for pointing out this discrepancy.

Referece:

[57] Mishin, E. et al. Artificial ionospheric layers driven by high-frequency radiowaves: An assessment, *J. Geophys. Res. Space Phys.* **121**, 3497–3524 (2016).
<https://doi.org/10.1002/2015JA021823>

3. In the last sentence of the Conclusion section authors claim that “The ionosphere provides a magnetised, nonlinear plasma medium ...”. I think that the word “nonlinear” mostly relates to waves or some interaction processes, but not to the ionosphere. So please remove this word or explain what you mean.

Response: We agree with Reviewer #1, and have now reformulated this sentence at the end of the Discussion on page 13 as:

“The ionosphere provides a magnetised, plasma medium which can offer insight into the physics of beat-wave interaction and other nonlinear effects, ...”

Reviewer #2 (Remarks to the Author):

This is an interesting and important contribution to the field.

- What are the noteworthy results?

As indicated below.

- Will the work be of significance to the field and related fields? How does it compare to the established literature?

Indicated below.

- Does the work support the conclusions and claims, or is additional evidence needed? Mostly, as indicated below.

- Are there any flaws in the data analysis, interpretation and conclusions? Do these prohibit publication or require revision?

Basically sound, see comments below.

- Is the methodology sound? Does the work meet the expected standards in your field? Yes.

- Is there enough detail provided in the methods for the work to be reproduced? Yes.

This is a complex topic and the presentation in the manuscript should be improved.

Response: We thank Reviewer #2 for the helpful comments and suggestions which have helped to improve the manuscript significantly. To help the reading of our response below, we have enumerated the Reviewer's comments as (1), (2), and so on.

(1) The title

> Ionospheric beat-wave Brillouin scattering experiment

is very general. The title should better reflect the results, it should point to what is interesting and unique. For example something like

> Controlled beat-wave Brillouin scattering in the ionosphere

Response: We thank Reviewer #2 for this suggestion and have now changed the title of the manuscript accordingly to "Controlled beat-wave Brillouin scattering in the ionosphere".

(2) Similarly, reading the article I felt a bit lost. As I started reading, I was wondering, what is the point, what is the new thing, what is the strategy?

Response: The paper is about active control of ionospheric plasma modes such as the ion-acoustic wave. This wave is used as part of a diagnostic on EISCAT, where using the ion mode gives information about the temperature since the dispersion depends on the ion-acoustic speed. There are also observations of electromagnetic sidebands where the frequency separation was close to an ion gyro frequency which gives information about the magnetic field. The new idea of the paper is to use active control of the modes excited in the ionosphere, which could provide improved diagnostics. These points are now emphasized in the Abstract [see question (6) below] and Discussion. In the Discussion, we write:

“The ion-acoustic wave is used as part of a diagnostic of the electron temperature in the ionosphere during ionospheric heating experiments [32,34], using the dependence of the dispersion of the mode on the ion-acoustic speed. There are also observations of electromagnetic sidebands where the frequency separation was close to an ion gyro frequency [35-45] which gives information about the local magnetic field and ion composition. Beat wave excitation of waves at the local electron plasma frequency [4] could also give information about the plasma density. The active control of the wave modes excited in the ionosphere has thus the potential of providing new and improved diagnostics.”

We hope that the revised manuscript now makes the purpose clear; see in particular our responses to questions (5), (6), (8), (9) and (14) below.

(3) The authors correctly reference many past SBS results but don't point out clearly enough what is different about their result. Be clear and specific about what filled the role of the probe wave in past experiments and the difference in the experiment being reported here.

Response: In the paragraph below Eq. (1a,b) on page 3, we now describe more carefully the roles of the probe, pump and IA waves. In particular we write about past experiments that:

“In previous Brillouin experiments [31-45], only the pump wave 0 was transmitted while wave 1 occurred naturally as SEE escaping the ionosphere and was recorded on ground as a down-shifted sideband. The physical process of natural SBS involves a nonlinear interaction in which the incident (pump) EM wave 0 decays into an electrostatic IA wave 2 and a scattered EM wave 1 that grow from noise via the SBS instability [38]. In this three-wave resonant interaction processes, the wave matching conditions are again satisfied: $Co_0 = Co_1 + Co_2$ and $k_0 = k_1 + k_2$, where Co_1 and Co_2 become complex valued leading to exponential growth in time of the waves during the linear phase of the instability [19-21].”

(4) Also, what is the experimental set up? It is, finally, explained in the paper, but pump and probe should be put into better context at the outset. Science is like detective work, but this is not a detective story, it's a presentation of results, where clear presentation is key.

Response: The experimental set-up is described in last paragraph of the introduction on page 2 with the location of the experiment shown on the map in Fig. 1, and more detail about the set-up is given in the Section "Experimental set-up" on page 4. We now give more details on the role of the pump and probe waves in the text below Eq. (1a,b).

(5) The simulation is a key part of the results being presented. What is meant by a full-wave simulation should be explained up front. I don't think it is ever explained in the text.

Response: See our detailed response to question (14) below, where we discuss the full-wave and ray-tracing simulations (also discussed in the Methods).

(6) I would eliminate the current first sentence of the abstract. The second paragraph explains it, and the last sentence of the conclusion also covers it. The abstract could say something like:

> Stimulated Brillouin scattering in the ionospheric plasma has previously been reported under conditions where the EM pump wave causes an EM sideband wave to be emitted by the plasma. Here we report results of controlled, pump and probe beat-wave-driven Brillouin scattering in the ionosphere, in which the EM sideband wave is controlled by the probe wave via an intermediary ion-acoustic wave. In an experiment at EISCAT in northern Norway, a pump consisting of large amplitude radio waves with ordinary (O) and extraordinary (X) mode polarization was injected into the overhead ionosphere, along with a less powerful probe wave, and radio sideband emissions observed on the ground clearly show stimulated Brillouin emissions at frequencies agreeing with, and changing with, the pump and probe frequencies. The experiment was simulated using a numerical radio-plasma model, and clearly verifies the experimental results.

I might have it wrong, please make it right, above and also below.

Response: We are grateful to Reviewer #2 for suggesting an improved Abstract which reads very well. We have now adopted this Abstract using some of the wordings of Reviewer #2. We want to emphasize that the probe and pump waves are used to control the excitation of ionospheric modes such as the ion-acoustic wave:

"Stimulated Brillouin scattering experiments in the ionospheric plasma using a single electromagnetic pump wave have previously been observed to generate an electromagnetic sideband wave, emitted by the plasma, together with an ion-acoustic wave. Here we report results of a controlled, pump and probe beat-wave-driven Brillouin scattering experiment, in which an ion-acoustic wave generated by the beating of electromagnetic pump and probe waves, results in electromagnetic sideband waves

that are recorded on the ground. The experiment used the EISCAT facility in northern Norway that has several high power electromagnetic wave transmitters and receivers in the radio frequency range. An electromagnetic pump consisting of large amplitude radio waves with ordinary (O) and extraordinary (X) mode polarization was injected into the overhead ionosphere, along with a less powerful probe wave, and radio sideband emissions observed on the ground clearly show stimulated Brillouin emissions at frequencies agreeing with, and changing with, the pump and probe frequencies. The experiment was simulated using a numerical full-scale model which clearly supports the interpretation of the experimental results. Such controlled beat-wave experiments demonstrate a new way of remotely investigating the ionospheric plasma parameters.”

(7)The simulation is called full wave. What that means should be explained.

Response: See our detailed response to question (14) below.

(8)On page 3

> The three waves taking part in resonant SBS interaction ...

> $w_1 = w_2 + w_3$ (1a)

> $k_1 = k_2 + k_3$ (1b)

> where the subscript 1 refers to the pump EM wave, 2 to the probe EM wave and 3 to the stimulated IA wave.

Equations 1a and 1b are confusing. The pump and the probe are inputs to the system. Pedagogically, inputs might be summed on the left, and the output would appear on the right. What is really happening must be

$w_1 - w_2 = w_3$

$k_1 - k_2 = k_3$

But these equations are not complete. It is not the IA wave (wave 3) that is being observed. The IA wave is translated into an EM wave of the same frequency that shows up as an SEE line, for example in the spectra in figure 2 where the pump and probe are also seen. How the IA wave translates to an EM wave in that spectrum should be explained.

Response: We agree with these insightful comments, and have now improved the description of the waves involved in the SBS process. Please note that we re-numbered wave 1, 2, 3 as 0, 1, 2 for consistency with the rest of the manuscript.

Equations (1a,b) are now written as

$$0 - 1 = 2, \tag{1a}$$

$$k_0 - k_1 = k_2, \tag{1b}$$

emphasizing that wave 0 and 1 drive wave 2. Below Eq. (1a,b) we now more carefully describe the SBS interaction. In particular, we write

“The controlled IA wave sets up a time-oscillating density grating in the ionospheric plasma, on which the pump and probe can be scattered into each other and into EM waves with different frequencies and wave vectors leading to EM sidebands at sum and difference frequencies $\omega \pm \omega_a$ and $\omega \mp \omega_a$. The newly created sidebands can again be scattered by the IA wave to new frequencies $\omega \pm 2\omega_a$ and $\omega \mp 2\omega_a$, and so on, leading to a cascade of sidebands at different frequencies, which can escape the ionosphere as SEE.”

(9) On page 3

> In previous Brillouin experiments [29-31], wave 2 occurred naturally in the plasma as a frequency down-shifted EM sideband that was driven unstable by the large amplitude pump EM wave.

Fu et al. (2015) write

> The physical process of SBS involves a nonlinear interaction in which an incident (pump) electromagnetic wave decays into an electrostatic IA wave and a scattered electromagnetic wave via the Brillouin instability. In such three-wave interaction processes, the wave matching conditions are satisfied: $\omega_0 = \omega_S + \omega_L$ and $k_0 = k_S + k_L$, where ω is the wave frequency; k is the wave propagation vector; and the subscripts 0, S and L denote the pump waves, the scattered waves and low-frequency waves, respectively.

This is clear, wave S (IA) is observed. In this paper this is, I think, called natural SBS. But it is not clear that wave S is the natural analog of the probe wave, since in the equations above S is an output and the probe is an input.

So for natural SBS what is happening is

$$\omega_{\text{pump}} = \omega_{\text{ia}} + \omega_{\text{see}} ; k_{\text{pump}} = k_{\text{ia}} + k_{\text{see}}$$

For controlled pump/probe SBS, maybe what is happening is

$$\omega_{\text{pump}} - \omega_{\text{probe}} = \omega_{\text{ia}} ; k_{\text{pump}} - k_{\text{probe}} = k_{\text{ia}}$$

and

$$\omega_{\text{pump}} \pm \omega_{\text{ia}} = \omega_{\text{see}} ; k_{\text{pump}} \pm k_{\text{ia}} = k_{\text{see}}$$

Natural SBS is mentioned on page 10 (comment below). Also, in figure 7, X mode, pump+probe panel, there is a second, outermost set, of nonlinear sidebands. They might barely appear in figure 2, but maybe not. I don't think those are explained, but they should be. Also, the image of the probe as a nonlinear sideband, in both figures, should be explained.

Basically, a clearer explanation of these processes, and how they relate to each other, is needed.

If these things do not stand out for me, who knows something about this field, it will also not stand out for many other potentially interested readers. All of this may be well understood by specialists and participants, but I think that publication in a Nature journal implies a wider

audience, and a need for a more background explanation and clarity than might be needed in a more specialized journal such as JGR or Annales Geophysicae or Radio Science. But I'd probably make the same basic comment, i.e. improve the clarity, in any journal.

Response: We now discuss natural SBS more carefully below Eqs (1a,b), where we write

“In previous Brillouin experiments [31-45], only the pump wave 0 was transmitted while wave 1 occurred naturally as SEE escaping the ionosphere and was recorded on ground as a down-shifted sideband. The physical process of natural SBS involves a nonlinear interaction in which the incident (pump) EM wave 0 decays into an electrostatic IA wave 2 and a scattered EM wave 1 that grow from noise via the SBS instability [38]. In this three-wave resonant interaction processes, the wave matching conditions are again satisfied: $\omega_0 = \omega_1 + \omega_2$ and $k_0 = k_1 + k_2$, where ω and k become complex valued leading to exponential growth in time of the waves during the linear phase of the instability [19-21].”

References:

[19] Sjölund, A. & Stenflo, L. Parametric coupling between ion waves and electromagnetic waves, *Appl. Phys. Lett.* **10**, 201 (1967). [https://doi.org/10.1016/0031-8914\(67\)90128-0](https://doi.org/10.1016/0031-8914(67)90128-0)

[20] Drake, J. F. et al. Parametric instabilities of electromagnetic waves in plasmas, *Phys. Fluids* **17**, 778 (1974). <https://doi.org/10.1063/1.1694789>

[21] Dysthe, K. B., Leer, E., Trulsen, J. & Stenflo, L. Stimulated Brillouin scattering in the ionosphere, *J. Geophys. Res.* **82**(4), 717–718 (1977). <https://doi.org/10.1029/JA082i004p00717>

[38] Fu, H. Y. et al. Stimulated Brillouin scattering during electron gyro-harmonic heating at EISCAT, *Ann. Geophys.* **33**, 983–990 (2015). <https://doi.org/10.5194/angeo-33-983-2015>

(10) In figure 2, why does the background change from blue to white at when the pump is off? Does the pump create that much noise across the band, or is it possibly coming from the receiving system? Or due to the data processing? In figure 2, O and X appear nearly identical in this regard. In figure 7, the same effect occurs in the O mode simulation, but not in the X mode simulation. Why the difference in the simulation?

Response: In the experiment using O mode polarization, the EM wave reaches the critical and upper hybrid layers (cf. Fig. 5) where it excites Langmuir and upper hybrid turbulence, which produce low amplitude, broadband emissions tens of kHz around the transmitted frequency. With X mode polarization there is a possibility of ‘leakage’ of O modes by imperfect polarization by the emitting antenna, leading to Langmuir turbulence near the critical layer. These emissions may give rise to the weak but visible

frequency components in Fig 2 when the transmitter is on. In the experiment, there is also repetition of the frequency spectra at multiples of 50 Hz, attributed to weak radiated sidebands at 50 Hz and harmonics thereof because of imperfect power supply filtering, as described in the first paragraph of the section ‘Experimental observations of SBS interaction.’ The simulation gives more control over the polarization, and when emitting a pure X mode the wave is reflected at the X mode cut-off near 212 km, and there is no leaking up to the critical layer at \approx 220 km, as seen in Fig. 6a, while in the numerical simulation using O mode, the wave reaches the critical layer where it excites Langmuir turbulence, giving rise to the broad band emissions in Fig. 7. It should be noted that the threshold for exciting Langmuir turbulence is very low, and as seen in Fig. 7b the low-amplitude probe wave (middle panel) is sufficient to excite Langmuir turbulence and broaden the spectrum.

On page 6, we now write:

“In addition there are weaker broad band emissions, manifest in the background signal about 0 dB in Fig. 2 (the blue background) when the pump is on. For O mode polarization, the EM wave reaches the critical and upper hybrid layers (cf. Fig. 5 below) where it excites Langmuir and upper hybrid turbulence, which produce low amplitude, broadband emissions tens of kHz around the transmitted frequency [17]. With X mode polarization there is a possibility of ‘leakage’ of O modes by imperfect polarization by the emitting antenna, leading to Langmuir turbulence near the critical layer. These emissions may give rise to the blue background signal in Fig 2 when the transmitter is on.”

Starting the end of page 11 we write:

“The broad frequency spectrum of waves seen also during X mode polarization in experiment (Fig. 2) may be due to the ‘leaking’ of O mode waves up to the critical layer due to imperfect polarization of the wave by the antenna. The simulation gives more control over the polarization, and when emitting a pure X mode the wave is reflected below the resonant layer for Langmuir turbulence, and therefore no broadband emission is seen in Fig. 7a. It should be noted that the threshold for exciting Langmuir turbulence is very low, and as seen in Fig. 7b (middle panel) the low-amplitude O mode probe wave alone is sufficient to excite Langmuir turbulence and broaden the frequency spectrum.”

We also introduced some smaller improvements in the text on pages 11 and 12, marked in yellow.

References:

[17] Leyser, T. B. Stimulated electromagnetic emissions by high-frequency electromagnetic pumping of the ionospheric plasma, *Space Sci. Rev.* **98**(3–4), 223–328 (2001). <https://doi.org/10.1023/A:1013875603938>

(11) On page 5

> The 7.1 MHz O and X signals in Runs V and VI have continuous down-shifted emissions, which are not well understood.

Some further comment to explain why not, or what it could be, or how to find out, would be interesting.

Response: We have now expanded on this topic, including a mechanism suggested by Reviewer #3 below. Started near bottom of page 5, we write some more detail:

“A detailed study of the time evolution of the frequency spectrum (Extended data) shows that in Run V the down-shifted emissions are initiated only near the probe signal and increase in amplitude during the first 300 seconds after which the amplitude remains at the same level, while throughout Run VI the down-shifted emissions exist at an almost constant amplitude near both the probe and pump signals. The emissions could potentially result from induced scattering of the pump and probe waves by thermal ions in which the ion-acoustic beat wave satisfies the Landau resonance where the ion velocities are close to the wave velocity. Another possibility is that the emissions result from the scattering of the probe and pump waves off velocity shear-modified IA waves [50,51] generated by inhomogeneous ion flow along the magnetic field lines. Ion flows may be produced by an increase of the electron pressure [52] by an increase in temperature during O mode heating, or potentially by an increase in plasma density during X mode heating [45].”

References:

[45] Blagoveshchenskaya, N. F. Borisova, T. D., Yeoman, T. K., Häggström, I. & Kalishin, A. K. Modification of the high latitude ionosphere F region by X-mode powerful HF radiowaves: Experimental results from multi-instrument diagnostics, *J. Atmos. Solar-Terr. Phys.* **135**, 50–63 (2015). <https://doi.org/10.1016/j.jastp.2015.10.009>

[50] Gavrishchaka, V. V., Ganguli, S. B. & Ganguli, G. I. Origin of Low-Frequency Oscillations in the Ionosphere, *Phys. Rev. Lett.* **80**, 728–731 (1998). <https://doi.org/10.1103/PhysRevLett.80.728>

[51] Agrimson, E., D’Angelo, N. & Merlino, R. L. Excitation of ion-acoustic-like waves by subcritical currents in a plasma having equal electron and ion temperatures, *Phys. Rev. Lett.* **86**, 5282–5285 (2001). <https://doi.org/10.1103/PhysRevLett.86.5282>

[52] Kosch, M. J., Ogawa, Y., Rietveld, M. T., Nozawa, S. & Fujii, R. An analysis of pump-induced artificial ionospheric ion upwelling at EISCAT, *J. Geophys. Res.* **115**, A12317 (2010). <https://doi.org/10.1029/2010JA015854>

(12) On page 5

> While most of the O mode pump is absorbed, the probe, being fed by 2 of the 12 rows in

the antenna array, has a wider beam and can be partially reflected by the unperturbed ionosphere outside the turbulent region.

It should be clarified whether it is meant that the probe, as well as the pump, is absorbed within the turbulent region. It is stated on page 8, but no harm in clarifying it here.

Response: We have now clarified this statement near bottom of page 6 as:

“The O mode pump and probe waves that reach the turbulent region are absorbed. However, while most of the well collimated pump beam reaches the turbulent layer and is absorbed, the probe, being fed by 2 of the 12 rows in the antenna array, has a wider beam and can be partially reflected by the unperturbed ionosphere outside the turbulent region.”

(13) On page 8

> At the Spitze angle 4.94° , the O mode is converted to a Z mode wave that is reflected at a higher altitude and then propagates down to its electrostatic resonance, and at the negative Spitze angle -4.94° the O mode is converted directly to electrostatic waves.

This could be clarified with a few additional words. At the positive Spitze, why does the Z mode not reach its ES resonance on the way up? A reference or references would be good. At the negative Spitze, what ES waves are the O mode converted to? Is there an appropriate reference?

Response: The main reference is to the review paper by Mjølhus (1990) which describes the two conversion processes in some detail. We also cite Mjølhus et al. (1993) who gives further details about Langmuir turbulence excited by EM waves at different angles to the magnetic field. For (historical) reference we also cite the paper by Poverlein (1948) where to our knowledge the German term ‘Spitze’ originates. To conform with Mjølhus (1990) the angle between an injected ray and the vertical is denoted by θ and the angle between the geomagnetic field lines and the vertical is denoted by α .

On page 9, we write:

“In the α -plane, O mode rays injected in the Spitze region [58,59] $\alpha_{\text{min}} < \alpha < \alpha_{\text{max}}$, where $\alpha_{\text{min}} = \arcsin \left(\frac{v_{\text{ph}}}{v_{\text{te}}} + 1 \right) \sin \theta = 4.94^\circ$, reach the critical layer $\mu = 1$. (The German term ‘Spitze’ refers to the peaked shape of the ray paths in this region [60].) Near the critical layer, the O mode can excite Langmuir turbulence [59] in which small-scale Langmuir waves at frequencies near the plasma frequency are nonlinearly coupled to ion-acoustic waves. A small amount of O mode power will be converted to Z mode waves in a narrow cone around $\theta = +\theta_{\text{min}}$ and to electrostatic Langmuir waves near $\theta = -\theta_{\text{min}}$ in the magnetic meridian plane, as described by Eq. (18) in Ref. [58] (See a detailed discussion in the Supplemental material). Most of the O mode beam reaches the critical layer in the α -plane.”

We have given a more detailed and technical discussion in the Supplemental material. Starting the third paragraph on page 2 of the Supplemental material, we write:

“The vertically directed heater beam has a typical width of about 6.5° - 7.0° ; hence, most of the injected power is injected at incidence angles smaller than the Spitz angle [5], $|\theta| \leq \theta_0$, where $\theta_0 = \arcsin \frac{v_{\text{ph}}}{(1 - \epsilon_z)^{1/2}} \sin \theta_{\text{d}}$ (with $v_{\text{ph}} = \omega / k$) is between 4.5° and 5° for a transmitted frequency $\omega = 6 - 8$ MHz. At certain injection angles, the O mode wave is linearly converted to plasma waves that remain trapped in the plasma (cf. Section 4 of Ref. [5]). In a narrow cone around the Spitz angle $\theta = \theta_0$, the O mode is converted to an upward propagating Z mode wave that is reflected at a higher altitude and then propagates down toward its electrostatic resonance during which its wavelength decreases and the Z mode is gradually converted to an electrostatic Langmuir wave. The electrostatic resonance is located about 65 meters below the $\omega = 1$ level in Fig 5b of the main article, where the dielectric tensor component $\epsilon_{zz} = 0$, with $\epsilon_{zz} = 1 - (1 - \epsilon_z \cos^2 \theta) / (1 - \epsilon_z)$, giving $\theta \approx 0.9990$. The direct coupling between the upward propagating O mode and downward propagating Langmuir wave is very weak since the two waves have vastly different wavelengths and different polarizations; the O mode has transverse polarization with the electric field perpendicular to the wave vector while the Langmuir wave is longitudinal with the electric field parallel to the wave vector [6]. At the negative Spitz angle $\theta = -\theta_0$, the O mode reaches a turning point slightly above the $\omega = 1$ level after which it propagates down to the electrostatic resonance where it is gradually converted to a Langmuir wave.”

References in the main article:

[58] Mjølhus, E. On linear conversion in a magnetized plasma, *Radio Science* **25**(6), 1321–1339 (1990). <https://doi.org/10.1029/RS025i006p01321>

[59] Mjølhus, E., Helmersen E. & DuBois D. F. Geometric aspects of HF driven Langmuir turbulence in the ionosphere, *Nonlin. Proc. Geophys.* **10**, 151–177 (2003). <https://doi.org/10.5194/npg-10-151-2003>

[60] Pöeberlein, H. Strahlwege von Radiowellen in der Ionosphäre, *S. B. Bayer. Akad. Wiss., Math-nat. Klasse*, 175–201 (1948). Available electronically at the Bayerische Akademie der Wissenschaften, <https://publikationen.badw.de/en/016510405>

References in the Supplemental material:

[5] Mjølhus, E. On linear conversion in a magnetized plasma, *Radio Sci.* **25**(6), 1321–1339 (1990). <https://doi.org/10.1029/RS025i006p01321>

[6] Mjølhus, E., Helmersen E. & DuBois D. F. Geometric aspects of HF driven Langmuir turbulence in the ionosphere, *Nonlin. Proc. Geophys.* **10**, 151–177 (2003). <https://doi.org/10.5194/npg-10-151-2003>

(14) On page 9

> A set of one-dimensional simulations of vertically injected plane waves are carried out over much longer times to study the nonlinear coupling of the EM wave to the ions.

and

> A second set of simulations are carried out to investigate beat-wave drive SBS interaction

It would be helpful, at the beginning of the discussion of the simulations, to explain the overall strategy. What are the goals, what different simulations are used to achieve those goals, in how many dimensions, what time steps and time span, etc. The term "full wave" could be explained here, and whether or not all of the simulations are full wave, and if not what the differences are and why.

Response: We have now added a paragraph in the numerical section to describe the over-all strategy of the numerical work, as well the difference between ray-tracing and full-wave simulations. The first paragraph of the 'Numerical analysis of experimental results' section starting on page 7 now reads:

"The experimental results are analysed numerically by means of ray-tracing and full-wave simulations using ionospheric parameters and profiles (Fig. 4) consistent with the experiment; both simulation models are described in the Methods. The strategy is to use ray-tracing simulations to calculate the local wave vector and corresponding wavelengths of the EM wave at different altitudes, which are used to predict the wave frequencies and Landau damping (described in the Supplemental material) of ion acoustic waves at different altitudes in the full-wave simulations. Short (< 1 ms) full-wave simulations in two-dimensional planes at different angles to the magnetic field are used in Fig. 5 to predict the general wave structure before any instability sets in, and to compare the wave pattern with ray-tracing simulations showing the ray paths of the EM waves. Much longer simulations (10 s) are for computational efficiency carried out in one dimension, along the vertical axis to investigate the non-linear coupling to electrostatic waves, first using only pump waves to demonstrate the general differences between O and X mode waves in Fig. 6 and then employing both probe and pump waves in Fig. 7 to simulate the experiment in Fig. 2. Details of the numerical results are discussed below."

In the Methods, we have added a paragraph describing the ray-tracing model, to complement the description of the full-wave simulation model. We have added more details about simulation parameters (e.g. the time-step) and a discussion about dimensionality in the Methods.

Reference:

[70] Eliasson, B. Full-scale simulations of ionospheric Langmuir turbulence, *Mod. Phys. Lett. B* **27**, 1330005 (2013). <https://doi.org/10.1142/S0217984913300056>

(15) A grammatical comment - for the simulations that were done, it makes more sense write "were carried out", instead of "are carried out", since it was done in the past.

Response: Yes, we agree and have now changed "are carried out" to "were carried out" for the experiment throughout in the manuscript.

(16) On page 10

> For the cases involving the O mode pump wave, there are also frequency up-shifted ion lines at 30 Hz above the transmitted frequency, and a narrow down-shifted line at $\Delta f \sim -25$ Hz, indicating that the amplitude of the pump exceeds the threshold for SBS generation, while the probe by itself is below the threshold for natural SBS.

Are the +30 and -25 both natural SBS? Why these frequencies? This should be made clearer (similarly to the comments above). Be clearer in explaining how the natural SBS works.

Response: On closer inspection, the downshifted peak at $\Delta f \sim -25$ Hz is quite narrow but the upshifted peak is broader and more diffuse. In the second paragraph on page 12, we have clarified the discussion as:

“For the cases involving the O mode pump wave, there is also a narrow down-shifted line at $\Delta f \sim -25$ Hz, indicating that the amplitude of the pump exceeds the threshold for SBS generation, while the probe by itself is below the threshold for natural SBS. At an IA frequency of $\omega_{IA} = 25$ Hz the resonant interaction takes place very close to the O mode reflection altitude = 219.91 km (cf. Fig. 4e) where the wave electric field has a maximum amplitude (Figs. 5b and 6b) leading to the SBS instability discussed below Eqs. (1a,b). The diffuse, up-shifted lines at 20 - 40 Hz above the transmitted frequency of the O mode pump may be interpreted as sidebands due to the scattering of the O mode wave off turbulence-generated ion-acoustic waves propagating downwards slightly below the critical layer leading to a Doppler up-shifted scattered signal.”

(17) On page 10

> The difference between experiment and simulation for O mode polarization may be understood by considering that in the experiment the returned signal was recorded at Kroken, 13 km NNW of the transmitter at Ramfjordmoen (cf. Fig. 1). As seen in Fig. 5, O mode rays at oblique angles to the vertical have the turning point at lower altitudes, below the critical layer. In the y-z plane, only the vertical beam reaches the critical layer, while oblique beams are reflected at lower altitudes. In this case, the anomalous absorption of the O mode wave may be less significant, leading to a stronger Brillouin interaction between the pump and probe.

Is this meant to say that this may be, or that it is, the reason?

Response: Yes, we clarify this in the last sentence quoted by the Reviewer #2, which is modified as (bottom of page 12):

“... while oblique beams are reflected at lower altitudes and therefore do not excite Langmuir turbulence. Consequently, these oblique waves are subject to less significant anomalous absorption, which may explain the stronger return signal of the O mode wave observed in the experiment compared to that of the simulation.”

(18) Maybe I missed it, but why not model the true experimental situation to verify if this explanation is correct. I'm guessing the reason is in the paper already, but please add a few words here. Those words could refer to the summary explanation of the simulations and simulation strategy discussed above, and/or also be included there.

Response: Yes, ideally one should do full-scale 2D or 3D simulations of EM wave propagation and Langmuir turbulence, but that is not computationally possible with the present simulation methods and resources: The short 2D simulation (~ 1 millisecond) in Fig 5 took about 5-10 hours to carry out, while much longer simulations over 10 seconds in 2D would take about 10 000 times longer. Therefore the longer simulations over 10 s were carried out in 1D, which still took more than 2 weeks due to the need to resolve small-scale Langmuir turbulence in both space and time. We hope to increase the computational efficiency to be able to run multiscale turbulence simulations in more than 1D in the future.

In the first paragraph on page 7 dealing with simulation strategy, we mention that:

“Simulations of the coupling of EM waves to small-scale Langmuir turbulence is challenging due to the disparate scales in space and time [70]. Short (< 1 ms) full-wave simulations in two-dimensional planes at different angles to the magnetic field are used in Fig. 5 to predict the general wave structure before any instability sets in, and to compare the wave pattern with ray-tracing simulations showing the ray paths of the EM waves. Much longer simulations (10 s) are for computational efficiency carried out in one dimension, along the vertical axis to investigate the non-linear coupling to electrostatic waves, ...”

In the methods, we mention

“A time-step of $\Delta t \approx 10^{-9}$ s is used, to resolve Langmuir turbulence with typical ion oscillation periods of $10^{-8} - 10^{-9}$ s.”

and

“To carry out 2D or 3D full-scale simulations of the EM wave propagation and Langmuir turbulence over 10s is not computationally possible with the present simulation methods and resources. Therefore, the longer simulations in Figs. 6 and 7 are carried out in one dimension, where the solution is assumed to vary only in the vertical direction”

Reference:

[70] Eliasson, B. Full-scale simulations of ionospheric Langmuir turbulence, *Mod. Phys. Lett. B* **27**, 1330005 (2013). <https://doi.org/10.1142/S0217984913300056>

(19) What is the predicted result if the measurements are repeated near a gyroharmonic?

Response: If the transmitter is tuned near an electron gyroharmonic it has been found in experiments that natural SBS originating near the critical layer (denoted SBS-1) strengthens [Fu et al., 2015] while SBS originating from the upper hybrid layer (SBS-2) is suppressed [Fu et al., 2020]. Therefore it would be very interesting to repeat the beat-wave experiment near a cyclotron harmonic to see if significant power can be transferred from the pump to the probe via the SBS-1 process. We have added a short discussion about this in the Discussion on page 13:

“It has been found in experiments [34,38] that when the transmitter is tuned near an electron gyroharmonic, natural SBS originating near the critical layer (denoted SBS-1) strengthens, while SBS originating from the upper hybrid layer (SBS-2) is suppressed. Therefore it would be of interest to carry out a beat-wave experiment near a gyroharmonic to investigate whether significant power can be transferred from the pump to the probe through the SBS-1 process.”

In the Discussion, we have also added suggestions how different wave modes (ion-acoustic, ion-cyclotron, plasma frequency) could be probed by beat waves for new and improved diagnostics of temperature, magnetic field and plasma density.

References:

[34] Fu, H. Y., et al. Electron temperature inversion by stimulated Brillouin scattering during electron gyroharmonic heating at EISCAT. *Geophys. Res. Lett.* **47**, e2020GL089747 (2020). <https://doi.org/10.1029/2020GL089747>

[38] Fu, H. Y. et al. Stimulated Brillouin scattering during electron gyro-harmonic heating at EISCAT, *Ann. Geophys.* **33**, 983–990 (2015). <https://doi.org/10.5194/angeo-33-983-2015>

(20) In the supplement, page 2,

> Since the down-coming probe wave was not very weak, the modest antenna used by the SEE receiver was deemed adequate.

However, the point of the receiver was not just to receive the probe, but also the SEE, after which the receiver is named. The SEE emission should be mentioned.

Response: We have clarified this and changed the sentence to:

“Since the down-coming probe wave and SEE sidebands were not very weak, ...”

We also introduced some smaller improvement in the text (marked in yellow) with reference to Fu et al. [2015,2020] (Refs. [3,4]) for the SEE receiver, and the discussion of the O mode beam propagation within the Spitze region.

Reviewer #3 (Remarks to the Author):

The manuscript describes an interesting experiment on nonlinear wave mixing in the ionosphere in which two EM waves are introduced to produce a beat wave at ion acoustic (IA) frequency, which is observed in the spectrogram. Generally, in ionospheric modification experiments a pump wave is introduced and scattered waves are observed. The authors describe a novel way to introduce two EM waves travelling in opposite directions to form a beat wave. The manuscript is clearly written and relevant to the area of nonlinear plasma physics.

While the novelty of the experiment is noteworthy and its explanation based on parametric processes is a possibility, it could be improved by considering and ruling out other possible mechanisms for the generation of the IA waves.

Normally, the IA waves do not spontaneously grow in ionospheric plasma with $T_i \sim T_e$ since they are heavily ion Landau damped. However, they are quite easily generated if there is an inhomogeneous ion flow along the magnetic field [Gavrishchaka et al., Phys Rev Lett., 80, 728, 1998], which has also been experimentally demonstrated in laboratory [Agrimson et al., Phys Rev Lett., 86, 5282, 2001]. The ponderomotive force generated by the EM waves in the experiment could locally drive an inhomogeneous ion flow along the magnetic field and could generate IA waves. The difference of the classical IA wave with the velocity shear-modified IA (SMIA) wave is that the SMIA waves are oblique and depends on the ratio of perpendicular to parallel wave vectors (k_{\perp}/k_{\parallel}). Without measurements of wave vectors along and across the magnetic field it is difficult to distinguish between the two. These waves would have the same signature in the spectrogram (Fig 2). The up and down shifted IA signature may be generated by different signs of (k_{\perp}/k_{\parallel}) [see dispersion relation given in Gavrishchaka et al.]. The authors should provide arguments and evidence to rule this possibility out.

Since the SMIA waves are kinetic, a PIC simulation with at least 2D is necessary for numerical simulation of SMIA. The simulation model used in the manuscript is 1D and presumably fluid, which will exclude this possibility. While the ponderomotive force of one EM wave may not be sufficient to generate above threshold shear the oppositely generated ion flows by counter propagating pump and probe waves may do the job.

Response: We thank Reviewer #3 for these insightful comments. We believe that in our case the beat-wave drives the ion-acoustic wave, since the SEE signals change according to the difference between the pump and probe signals. Since the IA wave is driven by the beating of the probe and pump waves, it is much less sensitive to the ion Landau damping than natural SBS relying on the SBS instability where the pump wave decays into an ion-acoustic wave and a down-shifted sideband. In fact, the driven IA wave can in principle be a quasi-mode, not following the un-driven IA dispersion relation.

On the other hand, we would like to propose that the shear-flow driven ion-acoustic modes could be involved in the generation of the down-shifted features discussed below.

In addition, the manuscript mentions, on page 5 line 9, “The 7.1 MHz O and X signals in Runs V and VI have continuous down-shifted emissions, which are not well understood.” This signature could result from induced scattering of the pump wave by thermal ions in which the beat wave satisfies the Landau resonance. Fig. 4e,f show that the Landau resonance is satisfied by the beat wave. In the induced scattering process the plasmon number density is conserved but not in the decay process. The manuscript does not mention if this hypothesis was tested.

Response: We thank Reviewer #3 for this suggestion that the down-shifted emissions could result from the scattering by thermal ions occurring at the interaction altitude of the 7.1 MHz O and X signals. We also would like to propose that the EM wave could be scattered off the velocity shear-modified IA (SMIA) waves mentioned above by Reviewer #3. We have added more details about the observations and write, starting near bottom of page 5:

“A detailed study of the time evolution of the frequency spectrum (Extended data) shows that in Run V the down-shifted emissions are initiated only near the probe signal and increase in amplitude during the first 300 seconds after which the amplitude remains at the same level, while throughout Run VI the down-shifted emissions exist at an almost constant amplitude near both the probe and pump signals. The emissions could potentially result from induced scattering of the pump and probe waves by thermal ions in which the ion-acoustic beat wave satisfies the Landau resonance where the ion velocities are close to the wave velocity. Another possibility is that the emissions result from the scattering of the probe and pump waves off velocity shear-modified IA waves [50,51] generated by inhomogeneous ion flow along the magnetic field lines. Ion flows may be produced by an increase of the electron pressure [52] by an increase in temperature during O mode heating, or potentially by an increase in plasma density during X mode heating [45].”

References:

[45] Blagoveshchenskaya, N. F. Borisova, T. D., Yeoman, T. K., Häggström, I. & Kalishin, A. K. Modification of the high latitude ionosphere F region by X-mode powerful HF radiowaves: Experimental results from multi-instrument diagnostics, *J. Atmos. Solar-Terr. Phys.* **135**, 50–63 (2015). <https://doi.org/10.1016/j.jastp.2015.10.009>

[50] Gavrishchaka, V. V., Ganguli, S. B. & Ganguli, G. I. Origin of Low-Frequency Oscillations in the Ionosphere, *Phys. Rev. Lett.* **80**, 728–731 (1998). <https://doi.org/10.1103/PhysRevLett.80.728>

[51] Agrimson, E., D'Angelo, N. & Merlino, R. L. Excitation of ion-acoustic-like waves by subcritical currents in a plasma having equal electron and ion temperatures, *Phys. Rev. Lett.* **86**, 5282–5285 (2001). <https://doi.org/10.1103/PhysRevLett.86.5282>

[52] Kosch, M. J., Ogawa, Y., Rietveld, M. T., Nozawa, S. & Fujii, R. An analysis of pump- induced artificial ionospheric ion upwelling at EISCAT, *J. Geophys. Res.* **115**, A12317 (2010). <https://doi.org/10.1029/2010JA015854>

We again thank all three Reviewers for their positive and constructive comments and suggestions that have helped to improve our manuscript significantly.

In addition to changes requested by the Reviewers, we have slightly changed the layout to conform with the style of Nature Communications. We hope that the revised manuscript is now suitable for publication in Nature Communications.

Yours sincerely

Dr. Bengt Eliasson

For and on behalf of all authors

REVIEWER COMMENTS

Reviewer #1 (Remarks to the Author):

I am satisfied with the author's response to my previous comments and I recommend this manuscript for the publication in the present form.

Reviewer #2 (Remarks to the Author):

The authors have done a really impressive job in improving the paper, with very clear and insightful explanations. I have learned a few interesting things and I thank the authors for that. This paper includes interesting comments, and brief but nice summaries of a broad swath of overlapping research.

- What are the noteworthy results?

As indicated in the first review.

- Will the work be of significance to the field and related fields? How does it compare to the established literature?

Indicated in the first review.

- Does the work support the conclusions and claims, or is additional evidence needed?

Yes.

- Are there any flaws in the data analysis, interpretation and conclusions? Do these prohibit publication or require revision?

No.

- Is the methodology sound? Does the work meet the expected standards in your field?

Yes.

- Is there enough detail provided in the methods for the work to be reproduced?

Yes.

Other comments and suggestions:

24> with ordinary (O) and extraordinary (X) mode

=> with ordinary (O) or extraordinary (X) mode

(to make it clear that it is one at a time)

(I realize that I had put "and" in my previous suggestion... :)

Something that might be made clear at the start of the paper is the relation of the frequencies of the measurements presented in the paper to the gyroharmonics. My understanding is that these results are presented as representing the situation far enough from the gyroharmonics that gyroharmonic effects are not a factor for these results. It is discussed in the Discussion but perhaps it should be stated in the introduction.

153> The probe and pump spectra are partially overlapping during the first three seconds due to the fundamental restriction $\Delta t |\Delta f| > 1$ with $\Delta t = 1$ s.

What Δf is meant? $\Delta t = 1$ s which is the step period both in Table 2 and Figure 2. Table 2 has a probe $|\Delta f|$ which is always 1 or greater so $\Delta t |\Delta f| > 1$ fails only for $t = 0$ to 1 s. So Δf is maybe the frequency width shown in figure 2 which then must have a minimum of 1 Hz.

In that case, from 0 to 1 s pump and probe are 1 Hz apart and so their linewidths touch, at 1 to 2 s then are 2 Hz apart and their linewidths are 1 Hz apart, and at 2 to 3 s they are 3 Hz apart. But the lines are

not rectangular and they overlap. Why is 3 s taken as the end of the overlap? Some explanation and clarification could be interesting and helpful.

161> A detailed study of the time evolution of the frequency spectrum (Extended data) shows that in Run V the down-shifted emissions are initiated only near the probe signal and increase in amplitude during the first 300 seconds after which the amplitude remains at the same level, while throughout Run VI the down-shifted emissions exist at an almost constant amplitude near both the probe and pump signals.

The explanation is

165> The emissions could potentially result from induced scattering of the pump and probe waves by thermal ions in which the ion-acoustic beat-wave satisfies the Landau resonance where the ion velocities are close to the wave velocity. Another possibility is that the emissions result from the scattering of the probe and pump waves off velocity shear-modified IA waves [50,51] generated by inhomogeneous ion flow along the magnetic field lines. Ion flows may be produced by an increase of the electron pressure [52] by an increase in temperature during O mode heating, or potentially by an increase in plasma density during X mode heating [45].

The different transmission polarizations in the two runs are not mentioned, but the down shifted emissions in the two runs are different. Perhaps this should be pointed out, for example:

=> A detailed study of the time evolution of the frequency spectrum (Extended data) shows that in Run V, in which O mode was used, the down-shifted emissions are initiated only near the probe signal and increase in amplitude during the first 300 seconds after which the amplitude remains at the same level, while throughout Run VI, in which X mode was used, the down-shifted emissions exist at an almost constant amplitude near both the probe and pump signals.

409> diagnostic of the electron temperature in the ionosphere during ionospheric heating experiments [32,34], using the dependence of the dispersion of the mode on the ion-acoustic speed

Here another relevant reference, from another HF facility:

Mahmoudian et al. 2019

NSEE Yielding Electron Temperature Measurements at the Arecibo Observatory

Journal of Geophysical Research: Space Physics, 124, 5, 3699-3708

<https://doi.org/10.1029/2019JA026594>

<https://agupubs.onlinelibrary.wiley.com/doi/abs/10.1029/2019JA026594>

Supplementary Information

end of page 2 > and the Z mode is gradually converted to an electrostatic Langmuir wave

top of page 3 > it propagates down to the electrostatic resonance where it is gradually converted to a Langmuir wave

Maybe Mjølhus explain it, but what is meant by gradually converted?

Supplementary Information

end of page 5 > The resonant layer progressively moves to lower altitudes for larger Δf , and the interaction region becomes wider.

In figure S5 I can't really tell how the width of the region is changing. The most distinctive thing is that the density fluctuation becomes less, but the width, if I had to say, looks mostly constant, if anything, maybe getting a bit narrower. Some additional explanation would be good.

end of page 5 > For small values of Δf , the interaction takes place in a thin layer very close to the critical surface $z = 211.68$ km, where the plane wave matching condition (1b) breaks down and the character of the interaction changes.

A few words describing the breakdown and the change in the interaction would be good, with a reference or two.

top of page 6 > leading to non-resonant interaction

Unless I missed it, I think there is no measurable non-resonant interaction in this data. Maybe that should be mentioned.

Formatting details:

I suggest that the authors and the journal use a system that reliably reproduces the symbols that are meant to be used.

For example, in the supplementary information, at the top of page 3, the "1-Y2" equation does not display correctly. The exponent "2" of "Y2" is okay, but there are inverted ? marks:

> 1-Y2

> (i)

> $\epsilon_{zz} = 1 - X (1 - Y^2 \cos^2 \alpha) / i$

There are similar problems in the response to the reviewers. For example, there are some bad symbols near the top of page 11:

> $|\theta|$, where $\theta = \arcsin(c/v) \sin \theta_0$ (with $v = c/n$) is between 4.5° and 5° for a transmitted frequency = 6 – 8 MHz.

and

> The electrostatic resonance is located about 65 meters below the $\epsilon = 1$ level in Fig 5b of the main article, where the dielectric tensor component $\epsilon_{xx} = 0$, with $\epsilon_{xx} = 1 - (1 - 2 \cos^2 \theta)/(1 - 2)$, giving 0.9990.

A pdf file should present correctly on all computer systems, so I suggest that doc files not be used for sharing these documents.

Grammar and typos:

22 northern Norway that has several

=> northern Norway, which has several

44 un-magnetized

=> unmagnetized

53 Beat-waves are

=> Beat waves are

(vs 61 beat-wave experiment - that is correct - check grammar rules)

170 electron pressure [52] by an increase in

=> electron pressure [52], by an increase in

253 Magnetic

=> magnetic

262 An Effective Radiated Power (ERP) of

=> An effective radiated power (ERP) of

(see capitalization rules for common and proper nouns)

264 using that the ERP =

=> using ERP =

289 (See a detailed

=> (see a detailed

308 waves of amplitude $E = 0.6 \text{ V/m}$ and frequency 6.3 MHz.

=> waves of frequency 6.3 MHz and that in free space would have an amplitude of $E = 0.6 \text{ V/m}$ at 220 km (i.e. the injected amplitude).

422 model treat the wave

=> model treats the wave

711 and lose-ups

=> and close-ups

Supplementary p2 fig S2 > The time-variation

=> The time variation

Perhaps the authors and the journal can check for additional grammatical and typo errors.

Reviewer #3 (Remarks to the Author):

The authors have adequately addressed the issues that I raised. The manuscript is now ready for publication.

Reviewer #2 (Remarks to the Author):

The authors have done a really impressive job in improving the paper, with very clear and insightful explanations. I have learned a few interesting things and I thank the authors for that. This paper includes interesting comments, and brief but nice summaries of a broad swath of overlapping research.

Response: We thank again Reviewer #2 for the comments both in the first report and in the second report which has helped us to improve the manuscript. We have gone through the present comments and carefully introduced corrections as indicated below.

- What are the noteworthy results?

As indicated in the first review.

- Will the work be of significance to the field and related fields? How does it compare to the established literature?

Indicated in the first review.

- Does the work support the conclusions and claims, or is additional evidence needed?

Yes.

- Are there any flaws in the data analysis, interpretation and conclusions? Do these prohibit publication or require revision?

No.

- Is the methodology sound? Does the work meet the expected standards in your field?

Yes.

- Is there enough detail provided in the methods for the work to be reproduced?

Yes.

Other comments and suggestions:

24> with ordinary (O) and extraordinary (X) mode

=> with ordinary (O) or extraordinary (X) mode

(to make it clear that it is one at a time)

(I realize that I had put "and" in my previous suggestion... :)

Response: We agree and have changed “and” to “or” in this sentence.

Something that might be made clear at the start of the paper is the relation of the frequencies of the measurements presented in the paper to the gyroharmonics. My understanding is that these results are presented as representing the situation far enough from the gyroharmonics that gyroharmonic effects are not a factor for these results. It is discussed in the Discussion but perhaps it should be stated in the introduction.

Response: We agree with this, and now mention in the Introduction on page 2 that:

“The frequencies are chosen far enough from the electron gyroharmonics (cf. Fig. S2 in the Supplementary Information) such that gyroharmonic effects can be neglected.”

153> The probe and pump spectra are partially overlapping during the first three seconds due to the fundamental restriction $\Delta t |\Delta f| > 1$ with $\Delta t = 1$ s.

What Δf is meant? $\Delta t = 1$ s which is the step period both in Table 2 and Figure 2. Table 2 has a probe $|\Delta f|$ which is always 1 or greater so $\Delta t |\Delta f| > 1$ fails only for $t = 0$ to 1 s. So Δf is maybe the frequency width shown in figure 2 which then must have a minimum of 1 Hz.

In that case, from 0 to 1 s pump and probe are 1 Hz apart and so their linewidths touch, at 1 to 2 s then are 2 Hz apart and their linewidths are 1 Hz apart, and at 2 to 3 s they are 3 Hz apart. But the lines are not rectangular and they overlap. Why is 3 s taken as the end of the overlap? Some explanation and clarification could be interesting and helpful.

Response: Yes, the exact statements are slightly too detailed and not fully consistent. We have changed the sentence to:

“The probe and pump spectra are partially overlapping during the first few seconds.”

The fundamental restriction $\Delta t |\Delta f| > 1$ is for a pulse and its Fourier transform and reaches equality for a Gaussian pulse (with exact definitions of Δt and Δf). In our case the pulse is a 1s long square pulse in time but which has been windowed by a Hanning window. But we omit such detailed discussion here.

161> A detailed study of the time evolution of the frequency spectrum (Extended data) shows that in Run V the down-shifted emissions are initiated only near the probe signal and increase in amplitude during the first 300 seconds after which the amplitude remains at the same level, while throughout Run VI the down-shifted emissions exist at an almost constant amplitude near both the probe and pump signals.

The explanation is

165> The emissions could potentially result from induced scattering of the pump and probe waves by thermal ions in which the ion-acoustic beat-wave satisfies the Landau resonance where the ion velocities are close to the wave velocity. Another possibility is that the emissions result from the scattering of the probe and pump waves off velocity shear-modified IA waves [50,51] generated by inhomogeneous ion flow along the magnetic field lines. Ion flows may be produced by an increase of the electron pressure [52] by an increase in temperature during O mode heating, or potentially by an increase in plasma density during X mode heating [45].

The different transmission polarizations in the two runs are not mentioned, but the down shifted emissions in the two runs are different. Perhaps this should be pointed out, for example:

=> A detailed study of the time evolution of the frequency spectrum (Extended data) shows that in Run V, in which O mode was used, the down-shifted emissions are initiated only near the probe signal and increase in amplitude during the first 300 seconds after which the amplitude remains at the same level, while throughout Run VI, in which X mode was used, the down-shifted emissions exist at an almost constant amplitude near both the probe and pump signals.

Response: We thank Reviewer #2 for this suggestion, and have now inserted the statements “in which O mode was used” and “in which X mode was used” to clarify the wave polarizations in the respective case.

409> diagnostic of the electron temperature in the ionosphere during ionospheric heating experiments [32,34], using the dependence of the dispersion of the mode on the ion-acoustic speed

Here another relevant reference, from another HF facility:

Mahmoudian et al. 2019
NSEE Yielding Electron Temperature Measurements at the Arecibo Observatory
Journal of Geophysical Research: Space Physics, 124, 5, 3699-3708
<https://doi.org/10.1029/2019JA026594>
<https://agupubs.onlinelibrary.wiley.com/doi/abs/10.1029/2019JA026594>

Response: Thank you for this relevant reference which we now cite as Ref. [66].

Supplementary Information

end of page 2 > and the Z mode is gradually converted to an electrostatic Langmuir wave
top of page 3 > it propagates down to the electrostatic resonance where it is gradually converted to a Langmuir wave

Maybe Mjølhus explain it, but what is meant by gradually converted?

Response: Yes, Mjølhus (1991) has a better wording which we have partially adopted:

“... and then propagates down toward its electrostatic resonance during which its wavelength decreases and it becomes increasingly electrostatically polarized as it is transformed to an electrostatic Langmuir wave.”

Supplementary Information

end of page 5 > The resonant layer progressively moves to lower altitudes for larger Δf , and the interaction region becomes wider.

In figure S5 I can't really tell how the width of the region is changing. The most distinctive thing is that the density fluctuation becomes less, but the width, if I had to say, looks mostly constant, if anything, maybe getting a bit narrower. Some additional explanation would be good.

Response: Yes we agree with this observation, and now write

“The resonant layer progressively moves to lower altitudes for larger Δf , and the ion fluctuations become weaker.”

end of page 5 > For small values of Δf , the interaction takes place in a thin layer very close to the critical surface $z = 211.68$ km, where the plane wave matching condition (1b) breaks down and the character of the interaction changes.

A few words describing the breakdown and the change in the interaction would be good, with a reference or two.

Response: We have expanded on the description of waves near the interaction region, and write write on page 5:

“For small values of Δf , the interaction takes place in a thin layer very close to the reflection point $z = 211.68$ km, where the spatial profile of the electromagnetic wave is best described by Airy functions [6,10] and not as a sinusoidal wave. Therefore the wave vector matching condition (1b) breaks down and the character of the interaction may change from a three-wave interaction to a modulational interaction [11].”

We cited Mjølhus et al. (2003) and the new reference Lundborg and Thidé (1985) for description of the standing wave pattern near reflection point in magnetized plasma. Bingham & Lashmore-Davies (1976) discusses nonlinear interaction near a reflection point.

References:

[6] Mjølhus, E., Helmersen E. & DuBois D. F. Geometric aspects of HF driven Langmuir turbulence in the ionosphere, *Nonlin. Proc. Geophys.* 10, 151–177 (2003).

<https://doi.org/10.5194/npg-10-151-2003>

[10] Lundborg, B. and Thidé, B. Standing wave pattern of HF radio waves in the ionospheric reflection region: 1. General Formulas, *Radio Sci.*, 20(4), 947–958 (1985).

<https://doi.org/10.1029/RS020i004p00947>

[11] Bingham, R. and Lashmore-Davies C. N. Self-modulation and filamentation of electromagnetic waves in a plasma, *Nuclear Fusion* 16(1), 67–72 (1976).

<https://doi.org/10.1088/0029-5515/16/1/007>

top of page 6 > leading to non-resonant interaction

Unless I missed it, I think there is no measurable non-resonant interaction in this data. Maybe that should be mentioned.

Response: Thank you for pointing this out. Sidebands at multiples of 50 Hz are to some degree masked by the repetition of the frequency spectra at harmonics of 50 Hz due to imperfect power supply filtering as mentioned on page 6 of the main article, but there are a few cases in Fig. 2 which indicate non-resonant interaction for $\Delta f = -100$ Hz, and

these are supported by the simulation in Fig.7. We have added a note on this end of page 5 in the Supplementary information:

“...leading to non-resonant interaction. In Fig. 5 of the main article, Runs II and III have clearly visible upshifted sidebands at +100 Hz for $\Delta f = -100$ Hz indicating non-resonant interaction, while this sideband is weaker for other runs. Such an up-shifted sideband is also visible in the simulated spectrogram in Fig. 7a for the pump+probe case, supporting the experimental observations.”

Formatting details:

I suggest that the authors and the journal use a system that reliably reproduces the symbols that are meant to be used.

For example, in the supplementary information, at the top of page 3, the "1-Y2" equation does not display correctly. The exponent "2" of "Y2" is okay, but there are inverted ? marks:

> 1-Y2
> (i)
> $\epsilon_{zz} = 1 - X(1 - Y2 \cos 2\alpha) / \epsilon$

There are similar problems in the response to the reviewers. For example, there are some bad symbols near the top of page 11:

> $|\delta| \approx \delta$, where $\delta = \arcsin(\epsilon / (1 - \epsilon)) \sin \theta$ (with $\theta = \theta$) is between 4.5° and 5° for a transmitted frequency = 6 – 8 MHz.

and

> The electrostatic resonance is located about 65 meters below the $\epsilon = 1$ level in Fig 5b of the main article, where the dielectric tensor component $\epsilon_{zz} = 0$, with $\epsilon_{zz} = 1 - (1 - 2 \cos^2 \theta) / (1 - 2)$, giving 0.9990.

A pdf file should present correctly on all computer systems, so I suggest that doc files not be used for sharing these documents.

Response: We are sorry for the problems with formatting which may be due to the conversion of submitted Word files. Hopefully such problems will not appear in the revised manuscript and response letter.

Grammar and typos:

22 northern Norway that has several
=> northern Norway, which has several

Response: Corrected.

44 un-magnetized
=> unmagnetized

Response: Corrected.

53 Beat-waves are
=> Beat waves are
(vs 61 beat-wave experiment - that is correct - check grammar rules)

Response: Corrected.

170 electron pressure [52] by an increase in
=> electron pressure [52], by an increase in

Response: Corrected.

253 Magnetic
=> magnetic

Response: Corrected.

262 An Effective Radiated Power (ERP) of
=> An effective radiated power (ERP) of
(see capitalization rules for common and proper nouns)

Response: Corrected.

264 using that the ERP =
=> using ERP =

Response: Corrected.

289 (See a detailed
=> (see a detailed

Response: Corrected.

308 waves of amplitude $E = 0.6$ V/m and frequency 6.3 MHz.
=> waves of frequency 6.3 MHz and that in free space would have an amplitude of $E = 0.6$ V/m at 220 km (i.e. the injected amplitude).

Response: Corrected.

422 model treat the wave
=> model treats the wave

Response: Corrected.

711 and lose-ups
=> and close-ups

Response: Corrected.

Supplementary p2 fig S2 > The time-variation
=> The time variation

Response: Corrected. Thank you for finding these typos.

We again thank Reviewer #2 for the positive and helpful comments and suggestions that have improved our manuscript and eliminated several typos. We hope that the revised manuscript is now suitable for publication in Nature Communications.

REVIEWER COMMENTS

Reviewer #2 (Remarks to the Author):

I thank the authors for their responses and for their patience with my replies. Mjølhus is a particularly insightful author, and this article's clarity in physical insight promises to follow in that tradition. That will be a benefit for Nature Communications readers, and a resource for those starting out or for anyone interested in this area.

Below I mention a few other places where clarity might be improved or additional insights included. Please excuse some disorganization and possible repetition. I hope these comments are helpful. Nice paper.

--- natural SBS

In the discussion of natural SBS, the manuscript says:

102 In previous Brillouin experiments [31-45], only the pump wave 0 was transmitted while wave 1 occurred naturally as SEE escaping the ionosphere and was recorded on ground as a down-shifted sideband.

104 The physical process of natural SBS involves a nonlinear interaction in which the incident (pump) EM wave 0 decays into an electrostatic IA wave 2 and a scattered EM wave 1 that grow from noise via the SBS instability [38].

This seems to say that both the IA wave and the scattered EM wave grow from noise. It might be more accurate to say that the IA waves grow from noise, thus allowing the pump to start the instability and producing the scattered EM wave. So perhaps something like this is more accurate:

=> The physical process of natural SBS involves a nonlinear interaction in which the incident (pump) EM wave 0 decays into an electrostatic IA wave 2 and a scattered EM wave 1 that grow via the SBS instability [38], where electrostatic IA wave 2 is initially excited by noise-like fluctuations in the plasma.

See for example Garmire 2018 (<https://doi.org/10.1155/2018/2459501>, might be good to add to the reference list). At the start of her section 3, she writes:

> SBS occurs in any medium (liquid, solids, gases, plasmas) by interaction of laser light and a coherent density grating caused primarily (but not exclusively) by electrostriction.

> The laser reflects off the grating, which moves with velocity v_{ac} ; it will be Doppler-shifted, and scattered light will experience a wavelength-shift in a coherent inelastic scattering process.

> As with any parametric amplifier, the gain builds from whatever input signal is present. In most cases there is no input signal, except for noise; spontaneous Stokes waves occur by scattering from thermally excited acoustic vibrations.

Of course for us the laser is the HF pump wave and the acoustic vibrations are the IA wave.

This suggestion is again with the aim of giving the best possible physical interpretation and feeling to the explanations and discussions.

That was also my aim in asking what is meant by "gradually converted" in the Supplementary Information, i.e. to try to give some feeling of how that works, and if need be the limits of what we know about it.

--- polarization and SBS

For the experiment presented here, with pump/probe frequencies not near the gyroharmonics, X mode produces more SBS, and O mode less SBS, because O mode is absorbed. At the gyroharmonics O mode is not expected to be absorbed. In that case:

- Would O and X be expected to produce equally strong SBS?

- Is there any reason the E field direction and reflection height would be expected to cause different excitations of SBS IA or MSBS EIC waves, or different SIBS (see comments below), for X vs O mode pumping, assuming both experience equal absorption?

Please see further related comments below.

--- non-resonant excitation and ion gyroharmonics

In the supplementary information:

127 clearly visible upshifted sidebands at +100 Hz for $Df = -100$ Hz

It is unfortunate that -100 Hz was chosen for the largest probe offset, not only because of the 50 Hz power line harmonics but also the O+ (~50 Hz) and NO+ (~25 Hz) gyroharmonics.

Perhaps not only should power line harmonics be mentioned, but also the ion harmonics be pointed out. Could they be somehow involved in the non-resonant enhancements?

--- Airy pattern

In Figure 6:

303 The X mode is reflected and sets up a standing wave pattern, while the O mode is absorbed resulting in an up-going wave with weak altitude-dependence of the amplitude below the critical layer.

An oscillating Airy pattern is clearly seen for X mode. Are similar oscillations not seen for O mode simply because the reflected wave is weaker?

My impression is that altitude dependence of the amplitude is due to the change in phase velocity near reflection, how is that affected by absorption (which is also mostly near O mode reflection)?

In the figure, oscillations for O mode do seem to appear near Z mode reflection, is that right? Is it possible to explain why are oscillations visible at Z mode reflection and not at O mode reflection?

Some details are hard to see on the figure even when zoomed -- perhaps the digital resolution of the figure should be improved, even if the size of the figure on the page is not changed. (Zooming is an advantage we have with digital media so why not make use of it.)

--- wave processes

Another direction for additional insight is pointed to by Bernhardt et al. 2009, who write:

> Laser induced parametric decay processes that have been detected in an unmagnetized plasma include the decay of the light wave into

> (1) an ion acoustic and an electron plasma wave

This is the PDI, parametric decay instability, which leads to cavitating Langmuir turbulence and enhanced ion and plasma lines.

In the manuscript:

76 While O mode waves reach the upper hybrid and critical layers where they can be converted to high-frequency electrostatic upper hybrid and Langmuir waves (electron plasma oscillations), the X mode waves are reflected below these resonant layers, leading to significant differences with respect to wave absorption, ionospheric turbulence, and electron heating.

Perhaps the PDI and cavitating Langmuir turbulence should be mentioned here, with a reference or two.

For clarity for non-experts, explain what anomalous absorption means as a subset of absorption.

Some related excerpts in the manuscript:

191 The returned O mode pump signals have an order of magnitude lower intensities than the X mode signals, which indicates almost complete conversion of the O mode to electrostatic waves in the ionosphere.

244 with a temperature increase of a few thousand Kelvin in a localized region around the upper hybrid and critical layers where the O mode is absorbed.

How much of the temperature increase is due to anomalous absorption i.e. the PDI+cavitation, and how much is due to non-anomalous absorption i.e. collisional heating?

274 Red lines in panel (b) indicate rays injected at the Spitze angle $+4.94^\circ$ where the O mode is absorbed and converted to Z mode (slow X mode) waves and at -4.94° to electrostatic Langmuir waves.

303 The X mode is reflected and sets up a standing wave pattern, while the O mode is absorbed resulting in an up-going wave with weak altitude-dependence of the amplitude below the critical layer.

345 For O mode polarisation in Fig. 7b, the EM wave reaches the critical layer at $z = 219.91$ km where Langmuir turbulence is excited, leading to a significant widening of the frequency spectrum and to anomalous absorption of the probe and pump waves.

Please explain if the widening is the yellow in figure 7b, or something else.

356 The combined pump + probe in Fig. 7b are significantly absorbed, and only weak traces of returned pump and probe signals are seen in the spectrogram.

371 The pumping near gyroharmonics suppresses the formation of small-scale field aligned striations [62] on which the pump wave can be anomalously absorbed near the upper hybrid layer [63-65]. The present experiment was conducted with frequencies away from the electron gyroharmonics, which may have resulted in the production of field-aligned striations, leading to the absorption of the O mode wave. The resulting decrease of the O mode pump amplitude below the threshold for SBS instability may explain the absence of natural SBS lines in the experimental spectrograms in Fig. 2 for O mode polarization (Runs I, IV and V). The 1D numerical model neglects the effect of field-aligned striations, and the unattenuated O mode pump wave is allowed to reach the critical layer where its amplitude exceeds the threshold for natural SBS generation, resulting in the lines in Fig. 7b.

Is "on which the pump wave can be anomalously absorbed near the upper hybrid layer" the same situation as the PDI and Langmuir turbulence and other absorption-related statements in the manuscript? The specificity of this statement raises potential confusion about that which should be clarified.

389 oblique waves are subject to less significant anomalous absorption, which may explain the stronger return signal of the O mode wave observed in the experiment compared to that of the simulation.

401 The O mode is significantly absorbed by the ionosphere, attributed to conversion to electrostatic waves at the upper hybrid and critical layers.

Here both upper hybrid and critical layers are mentioned in regard to O mode absorption, yet on line 371+ only the upper hybrid layer, with striations, is mentioned. Clarification is needed here also.

SEE is associated with striations at the upper hybrid layer. Are you saying that SEE is also originating at the critical layer, and/or that striations at the critical layer are also present and perhaps equally important for SEE and PDI/LT? Do lack of striations at the gyroharmonics mean that the PDI/LT is not occurring at the gyroharmonics, or perhaps at the critical level but not at the UH level? This should be part of the clarification.

Bernhardt et al. 2009 continue:

> (2) an ion acoustic and an electromagnetic wave

This corresponds to SBS and, with a magnetic field, MSBS/MIBS processes.

Bernhardt et al. 2011 discuss stimulated ion Bernstein scattering (SIBS) at the second gyroharmonic, and Samimi et al. 2012 goes farther, discussing also upper hybrid/electron Bernstein and neutralized ion Bernstein (IB) waves.

A brief mention that this occurs at the second gyroharmonic, i.e. that SBS/MSBS is suppressed, and that SIBS appears, and perhaps a brief insight as to why, would be a helpful contribution to clarity and completeness.

> (3) an electron plasma and an electromagnetic wave

That is SRS, stimulated Raman scattering.

> (4) two electron plasma waves

That is the TPI, two-plasmon instability.

I think 3 and 4 have not been seen in the ionosphere. It would be interesting to comment on the prospects for seeing them, and what experimental geometry and conditions would be needed to do so.

--- MSBS and SIBS

Related to my previous comments about the gyroharmonics:

In the Supplementary Information you write

102 The dispersion relation shows that the IA wave has a resonance ($k_{IA} \rightarrow \infty$) at $\omega_{IA} = \omega_{ci} \cos\theta$, a cut-off ($k_{IA} \rightarrow 0$) at $\omega_{IA} = \omega_{ci}$, and a forbidden frequency band between the resonance and cut-off.

104 The branch at $\omega_{IA} < \omega_{ci}$ is the magnetized IA wave, while the higher frequency branch $\omega_{IA} > \omega_{ci}$ is the electrostatic ion cyclotron wave.

106 for IA waves propagating at small angles to the magnetic field, the effect of the magnetic field on the IA wave may be neglected except very close to the cyclotron resonance

Close to the cyclotron resonance, or at larger angles to the magnetic field, the magnetic field becomes important and SRS becomes magnetized stimulated Brillouin scatter (MSBS).

Several papers discuss magnetized SBS (MSBS). For example, Bernhardt et al. 2010, in the abstract, write "Either an ion-acoustic wave with a frequency less than the ion cyclotron frequency (f_{CI}) or an electrostatic ion cyclotron (EIC) wave just above f_{CI} can be produced." and calls it MSBS, and Fu et al. 2020, in the introduction, discusses "IA and EIC emissions due to MSBS". (Both of those papers you have already cited.)

Other papers discuss SBS near the gyroharmonics but might not call it MSBS, for example that might be the case for Bernhardt et al. 2009, perhaps because it was an earlier paper?

Samimi et al. 2012 note that MSBS is suppressed at both the second and third gyroharmonics, and discuss upper hybrid/electron Bernstein and neutralized ion Bernstein (IB) waves, but do not call it SIBS.

Perhaps you could assist in clarifying what is meant by MSBS and SIBS e.g. in what cases they apply, which appears to be near the gyroharmonics and at oblique angles to the magnetic field. As explained in the manuscript, this is not the case for your paper but a clarifying comment would assist interested readers.

Mahmoudian et al. 2019 (<https://doi.org/10.1029/2019JA026594>) discuss MSBS for the case at Arecibo, with about 45 degrees to the magnetic field, and not too near a gyroharmonic. Mahmoudian et al. appear to see an MSBS ion acoustic line but not an ion cyclotron line.

A comment on what might be done at Arecibo in regard to SBS, MSBS, SIBS, and beat-wave SBS, MSBS, and SIBS, would be interesting, especially since plans are being discussed to rebuild the Arecibo high-power HF facility, perhaps within the next two years. The initial restoration would likely be at 5.1 MHz and perhaps include the 5th harmonic, but a proposal for a new Arecibo heating facility might include heating down to the second harmonic (about 2 MHz at Arecibo) -- here you have an opportunity to assist in that effort, and it is a current and interesting topic.

--- experimental setup

In regard to my question about previous line 153

>> The probe and pump spectra are partially overlapping during the first three seconds due to the fundamental restriction $\Delta t |\Delta f| > 1$ with $\Delta t = 1$ s.

In the response you wrote:

> The fundamental restriction $\Delta t |\Delta f| > 1$ is for a pulse and its Fourier transform and reaches equality for a Gaussian pulse (with exact definitions of Δt and Δf). In our case the pulse is a 1s long square pulse in time but which has been windowed by a Hanning window. But we omit such detailed discussion here.

This seems the type of detail useful for explaining and perhaps sometime repeating the experiment, which perhaps should be included in the methods section or in supplementary information.

--- typos/grammar/references

In the supplementary information:

127 In Fig. 5 of the main article, Runs II and III have clearly

=> I think Fig 2 is meant.

In the manuscript:

Transmitter and run and array are pretty generic things, and common nouns are not capitalized in English (if not at the start of a sentence) (check on usage for common vs proper nouns).

Some examples here from the manuscript:

127 while the Transmitters 3–12 were used for the pump

=> while transmitters 3–12 were used for the pump

125 EISCAT's antenna array 1

That is correct.

265 EISCAT Array 1

Should be "array".

128 divided into 6 Runs

divided into 6 runs

131 Each Run

=> each run

137 heater Array 1, where transmitters 3 – 12 were used for the pump and Transmitters 1 and 2

=> heater array 1 ... transmitters 3 – 12 ... transmitters 1 and 2

140 Run I

=> run 1 (note digit 1 vs Roman numeral I)

152 Runs

=> runs

and so on.

144 Experimental observations of SBS interaction

Figure 2 is between the title and the text but I guess this will be fixed by the journal before publication.

186 arb. Units, linear scale

=> arb. units, linear scale

407 natural SBS originating near the critical layer (denoted SBS-1) strengthens, while SBS originating from the upper hybrid layer (SBS-2) is suppressed.

Was this notation (SBS-1 and SBS-2) originated by [31] Bernhardt et al 2009, or by whom? It is specific enough that it might be good to reference that.

470 Langmuir

Extra I.

564 [31]

The doi link to Norin et al. 2009 should be

<https://doi.org/10.1103/PhysRevLett.102.065003>

In regard to my comment about explaining the meaning of anomalous absorption, you could explain it from the historical point of view, as absorption that was above that which was expected from collisional heating, and cite one of the early publications, in a similar way that you did in your explanation of the Spitzer angle.

13 June, 2021

We thank Reviewer #2 for the insightful comments and suggestions which have helped to improve our manuscript significantly. Below we list our point-by-point response to Reviewer #2's comments and our changes to the manuscript. All changes are indicated in yellow in the manuscript:

Reviewer #2 (Remarks to the Author):

I thank the authors for their responses and for their patience with my replies. Mjølhus is a particularly insightful author, and this article's clarity in physical insight promises to follow in that tradition. That will be a benefit for Nature Communications readers, and a resource for those starting out or for anyone interested in this area.

Below I mention a few other places where clarity might be improved or addition insights included. Please excuse some disorganization and possible repetition. I hope these comments are helpful. Nice paper.

Response: We are grateful to Reviewer #2 for the positive comments. For convenience, we have enumerated the Reviewer's comments as **1.**, **2.** and so on below:

1. --- natural SBS

In the discussion of natural SBS, the manuscript says:

102 In previous Brillouin experiments [31-45], only the pump wave 0 was transmitted while wave 1 occurred naturally as SEE escaping the ionosphere and was recorded on ground as a down-shifted sideband.

104 The physical process of natural SBS involves a nonlinear interaction in which the incident (pump) EM wave 0 decays into an electrostatic IA wave 2 and a scattered EM wave 1 that grow from noise via the SBS instability [38].

This seems to say that both the IA wave and the scattered EM wave grow from noise. It might be more accurate to say that the IA waves grows from noise, thus allowing the pump to start the instability and producing the scattered EM wave. So perhaps something like this is more accurate:

=> The physical process of natural SBS involves a nonlinear interaction in which the incident (pump) EM wave 0 decays into an electrostatic IA wave 2 and a scattered EM wave 1 that grow via the SBS instability [38], where electrostatic IA wave 2 is initially excited by noise-like fluctuations in the plasma.

See for example Garmire 2018 (<https://doi.org/10.1155/2018/2459501>, might be good to add to the reference list). At the start of her section 3, she writes:

> SBS occurs in any medium (liquid, solids, gases, plasmas) by interaction of laser light and a coherent density grating caused primarily (but not exclusively) by electrostriction.

> The laser reflects off the grating, which moves with velocity v_{ac} ; it will be Doppler-shifted, and scattered light will experience a wavelength-shift in a coherent inelastic scattering process.

> As with any parametric amplifier, the gain builds from whatever input signal is present. In most cases there is no input signal, except for noise; spontaneous Stokes waves occur by scattering from thermally excited acoustic vibrations.

Of course for us the laser is the HF pump wave and the acoustic vibrations are the IA wave.

This suggestion is again with the aim of giving the best possible physical interpretation and feeling to the explanations and discussions.

That was also my aim in asking what is meant by "gradually converted" in the Supplementary Information, i.e. to try to give some feeling of how that works, and if need be the limits of what we know about it.

Response 1.: We thank Reviewer #2 for this suggestion, and for pointing out the excellent review article by Garmire (2018) which we now cite as Ref. [14] in the Introduction and above Eq. (2). Near bottom of page 3, we have now adopted the formulation (again citing Ref. [14])

"The physical process of natural SBS involves a nonlinear interaction in which the incident (pump) EM wave 0 decays into an electrostatic IA wave 2 and a scattered EM wave 1 that grow via the SBS instability [14,40], where the electrostatic IA wave 2 is initially excited by noise-like fluctuations in the plasma."

Thank you again for your comment in your previous report on our formulation in the Supplementary Information about "gradually converted" Z mode which we reformulated in a clearer way.

2. --- polarization and SBS

For the experiment presented here, with pump/probe frequencies not near the gyroharmonics, X mode produces more SBS, and O mode less SBS, because O mode is absorbed. At the gyroharmonics O mode is not expected to be absorbed. In that case:

- Would O and X be expected to produce equally strong SBS?

Response 2: In the experiment the O mode is probably significantly absorbed at the upper hybrid layer due to mode conversion and absorption on magnetic field aligned striations. If transmitted near electron gyroharmonics, the O mode could propagate to higher altitudes to the critical layer. However, also there there might be significant absorption on the Langmuir turbulence which again would decrease the amplitude of the reflected pump wave although maybe to a lower degree which would increase the SBS interaction. There could potentially also be a transfer of energy and amplification of the probe signal. In the Discussion on page 14 we suggest such an experiment, writing:

“It has been found in experiments [36,40] that when the transmitter is tuned near an electron gyroharmonic, SBS-1 strengthens while SBS-2 is suppressed. Therefore it would be of interest to carry out a beat-wave experiment near an electron gyroharmonic to investigate whether significant power can be transferred from the pump to the probe through the SBS-1 process.”

In the Supplementary Information we also discuss the different absorption processes in the new section “Collisional and anomalous absorption of EM waves”.

3. - Is there any reason the E field direction and reflection height would be expected to cause different excitations of SBS IA or MSBS EIC waves, or different SIBS (see comments below), for X vs O mode pumping, assuming both experience equal absorption?

Please see further related comments below.

Response 3: The O mode could excite SBS IA when the beam propagates nearly parallel to the magnetic field, and MSBS EIC at higher altitudes near its turning point where it propagates at large angles to the magnetic field, since the SBS process drives ion waves primarily in the same direction as the EM wave. It could also drive the SIBS process near the upper hybrid layer, when the EM wave has electric field components perpendicular to the magnetic field and decays into an upper hybrid and an ion cyclotron wave propagating at large angles to the magnetic field. For X mode polarization the SBS IA and MS IAC could be triggered in a similar way as the O mode. The SIBS process is probably inefficient for X mode polarization since the X mode is reflected below the upper hybrid layer. Please see our comments and addition to the Supplementary Information discussed in Comment/Response **17** below.

4. --- non-resonant excitation and ion gyroharmonics

In the supplementary information:

127 clearly visible upshifted sidebands at +100 Hz for $D_f = -100$ Hz

It is unfortunate that -100 Hz was chosen for the largest probe offset, not only because of the 50 Hz power line harmonics but also the O+ (~50 Hz) and NO+ (~25 Hz) gyroharmonics.

Perhaps not only should power line harmonics be mentioned, but also the ion harmonics be pointed out. Could they be somehow involved in the non-resonant enhancements?

Response 4: We agree with Reviewer #2, and have modified the statement to mention both the line harmonics and ion cyclotron harmonics, in the first paragraph on page 6 in the Supplementary Information, as:

“In Fig. 2 of the main article, Runs II and III have clearly visible upshifted sidebands at +100 Hz for $\Delta f = -100$ Hz indicating non-resonant interaction, while this sideband is weaker for other runs. Oxygen (O^+) and nitric oxygen (NO^+) ions have their gyrofrequencies at 46.6Hz and 24.8Hz, respectively, and their harmonics could potentially be involved in the enhancement at +100 Hz. However, such an up-shifted sideband is also visible in the simulated spectrogram in Fig. 7a for the pump+probe case, using un-magnetized ions, supporting the experimental observations without ion cyclotron harmonic effects. As mentioned in the main article, repetition of the frequency spectra at multiples of 50 Hz in Fig 2 is attributed to weak radiated sidebands at 50 Hz and harmonics thereof because of imperfect power supply filtering.”

5. --- Airy pattern

In Figure 6:

303 The X mode is reflected and sets up a standing wave pattern, while the O mode is absorbed resulting in an up-going wave with weak altitude-dependence of the amplitude below the critical layer.

An oscillating Airy pattern is clearly seen for X mode. Are similar oscillations not seen for O mode simply because the reflected wave is weaker?

Response 5: Yes, the reflected O mode is very weak and therefore no standing wave pattern is set up. For clarity we have moved the text describing the wave from the figure caption to the paragraph below Figure 6, where we write (see text marked in yellow)

“The X mode is reflected and sets up an oscillatory standing wave pattern in E_x and E_y below its cut-off.”

and

“The O mode (Fig. 6b) interacts via the oscillating two-stream instability (OTSI) and parametric decay instability (PDI) [52,53] near the critical altitude to excite small-scale electrostatic high-frequency Langmuir oscillations in E_z and low-frequency ion density fluctuations n_{i1} .”

and

“In this process, the up-going O mode is anomalously absorbed (see Fig. S6 in the Supplementary Information), resulting in a much weaker reflected wave and no visible oscillating standing wave pattern in E_x and E_y below the cut-off.”

References:

[52] Mjølhus, E., Helmersen E. & DuBois D. F. Geometric aspects of HF driven Langmuir turbulence in the ionosphere, *Nonlin. Proc. Geophys.* **10**, 151–177 (2003). <https://doi.org/10.5194/npg-10-151-2003>

[53] Eliasson, B. & Papadopoulos, K. HF wave propagation and induced ionospheric turbulence in the magnetic equatorial region, *J. Geophys. Res. Space Physics* **121**, 2727–2742 (2016). <https://doi.org/10.1002/2015JA022323>

6. My impression is that altitude dependence of the amplitude is due to the change in phase velocity near reflection, how is that affected by absorption (which is also mostly near O mode reflection)?

Response 6: The O mode increases in amplitude due to a decrease in the group velocity as it approaches the cut-off. Once the O mode reaches the region with small-scale Langmuir turbulence, then the EM wave is efficiently converted to Langmuir waves on the small-scale density fluctuations. This works as anomalous resistivity (e.g. Kruer & Dawson, 1972) on the O mode wave which rapidly decreases in amplitude with altitude as it propagates to higher altitudes. We now discuss this in some detail in the new section “Collisional and anomalous absorption of EM waves” in the Supplementary Information.

7. In the figure, oscillations for O mode do seem to appear near Z mode reflection, is that right? Is it possible to explain why are oscillations visible at Z mode reflection and not at O mode reflection?

Response 7: Some of the O mode is converted to Z mode waves which propagate to the Z mode cut-off where they are reflected and set up a standing wave pattern. We write, near bottom of page 11:

“The interaction between the O mode and the Langmuir turbulence also generates a lower amplitude Z mode wave that propagates above the O mode cut-off up to the Z mode cut-off where it is reflected, setting up a standing wave pattern below its reflection point.”

8. Some details are hard to see on the figure even when zoomed -- perhaps the digital resolution of the figure should be improved, even if the size of the figure on the page is not changed. (Zooming is an advantage we have with digital media so why not make use of it.)

Response 8: We will do our best to submit high-resolution figures for the published article. In the Supplementary Information, we have now produced the new Fig. S6 with a close-up near the critical layer of the O mode and the electrostatic Langmuir turbulence.

9. --- wave processes

Another direction for additional insight is pointed to by Bernhardt et al. 2009, who write:

> Laser induced parametric decay processes that have been detected in an unmagnetized plasma include the decay of the light wave into

> (1) an ion acoustic and an electron plasma wave

This is the PDI, parametric decay instability, which leads to cavitating Langmuir turbulence and enhanced ion and plasma lines.

In the manuscript:

76 While O mode waves reach the upper hybrid and critical layers where they can be converted to high-frequency electrostatic upper hybrid and Langmuir waves (electron plasma oscillations), the X mode waves are reflected below these resonant layers, leading to significant differences with respect to wave absorption, ionospheric turbulence, and electron heating.

Perhaps the PDI and cavitating Langmuir turbulence should be mentioned here, with a reference or two.

For clarity for non-experts, explain what anomalous absorption means as a subset of absorption.

Response 9: Thank you for pointing this out. The 3-wave PDI gives rise to propagating IA waves, but the more 'severe' instability is the oscillating two-stream instability (OTSI) which is a 4-wave process in which the EM wave decays into two counter-propagating Langmuir waves and a purely growing (non-propagating) ion fluctuation which leads to strong, cavitating turbulence.

Sketch of OTSI and PDI

The above pictures (from Eliasson & Papadopoulos, 2016) show a cartoon sketch of these two instabilities (top box) and a solution of the dispersion relation showing the growth-rate and oscillation frequency of the instabilities (bottom box). The resulting ion fluctuations work as a density grating on which the EM wave is efficiently converted to Langmuir waves, effectively leading to anomalous resistivity (e.g. Krueer & Dawson, 1972) and absorption of the EM wave. In the Supplementary Information, we have written the new section “Collisional and anomalous absorption of EM waves” where we discuss these processes in some detail. At the critical layer the anomalous absorption is at least about 3 orders of magnitude higher than collisional absorption, and similar in magnitude is the absorption on striations near the upper hybrid layer.

We now write near top of page 3:

“However, a large amplitude O mode wave can be anomalously absorbed at the F2 layer [51], even though the plasma is only weakly collisional there (see Supplementary Information). Within a few milliseconds, there is a drop of about 10 dB in the reflected power. This is attributed to the fact that the O mode wave first excites short wavelength electrostatic high-frequency Langmuir waves (electron plasma oscillations) and low-

frequency ion density fluctuations via the oscillating two-stream instability (OTSI) and parametric decay instability (PDI) [52,53] near the critical layer where the transmitted frequency equals the electron plasma frequency. The ion fluctuations work as a grating on which the O mode is continuously converted to Langmuir waves, leading to an anomalous resistivity [54] that absorbs the O mode. On a longer time-scale of about a second, there is a further drop of 10-15 dB in the reflected power. This absorption is due to magnetic field aligned striations (density cavities) formed via a thermal instability, on which the O mode is converted to high-frequency upper hybrid waves [55-57] at the upper hybrid layer, a few km below the critical layer. The electrostatic waves remain in the plasma and dissipate their energy by heating the electrons and to lesser extent the ions. In contrast, the X mode wave is reflected by the ionosphere below these resonant layers and is therefore not absorbed, leading to significant differences with respect to wave absorption, ionospheric turbulence, and electron heating.”

References:

[51] Fejer, J. A. and Kopka, H. The Effect of Plasma instabilities on the ionospherically reflected wave from a high-power transmitter, *J. Geophys. Res.* **86**, 5746–5750 (1981).

<https://doi.org/10.1029/JA086iA07p05746>

[52] Mjølhus, E., Helmersen E. & DuBois D. F. Geometric aspects of HF driven Langmuir turbulence in the ionosphere, *Nonlin. Proc. Geophys.* **10**, 151–177 (2003). <https://doi.org/10.5194/npg-10-151-2003>

[53] Eliasson, B. & Papadopoulos, K. HF wave propagation and induced ionospheric turbulence in the magnetic equatorial region, *J. Geophys. Res. Space Physics* **121**, 2727–2742 (2016).

<https://doi.org/10.1002/2015JA022323>

[54] Kruer, W. L. & Dawson, J. M. Anomalous high-frequency resistivity of a plasma, *Phys. Fluids* **15**(3), 446–453 (1972). <https://doi.org/10.1063/1.1693927>

[55] Mjølhus, E. Anomalous absorption and reflection in ionospheric radio modification experiments, *J. Geophys. Res.* **90**(A5), 4269–4279 (1985). <https://doi.org/10.1029/JA090iA05p04269>

[56] Mjølhus, E. Theoretical model for long time stimulated electromagnetic emission generation in ionospheric radio modification experiments, *J. Geophys. Res.* **103**(A7), 14711–14729 (1998).

<https://doi.org/10.1029/98JA00927>

[57] Eliasson, B. & Papadopoulos, K. Numerical study of anomalous absorption of O mode waves on magnetic field-aligned striations, *Geophys. Res. Lett.* **42**, 2603–2611 (2015).

<https://doi.org/10.1002/2015GL063751>

10. Some related excerpts in the manuscript:

191 The returned O mode pump signals have an order of magnitude lower intensities than the X mode signals, which indicates almost complete conversion of the O mode to electrostatic waves in the ionosphere.

244 with a temperature increase of a few thousand Kelvin in a localized region around the upper hybrid and critical layers where the O mode is absorbed.

How much of the temperature increase is due to anomalous absorption i.e. the PDI+cavitation, and how much is due to non-anomalous absorption i.e. collisional heating?

Response 10: Anomalous absorption is much more significant than collisional heating in the F2 region, while collisional absorption can be significant at lower altitudes in the D region during daytime conditions, as discussed in the Supplementary Information. An order-of-magnitude estimate shows that anomalous absorption (and hence heating) is about 3 orders of magnitude higher than collisional absorption near the critical layer. A similar degree of anomalous absorption takes place near the upper hybrid layer once magnetic field aligned striations have been formed.

11. 274 Red lines in panel (b) indicate rays injected at the Spitzze angle $+4.94^\circ$ where the O mode is absorbed and converted to Z mode (slow X mode) waves and at -4.94° to electrostatic Langmuir waves.

303 The X mode is reflected and sets up a standing wave pattern, while the O mode is absorbed resulting in an up-going wave with weak altitude-dependence of the amplitude below the critical layer.

345 For O mode polarisation in Fig. 7b, the EM wave reaches the critical layer at $z = 219.91$ km where Langmuir turbulence is excited, leading to a significant widening of the frequency spectrum and to anomalous absorption of the probe and pump waves.

Please explain if the widening is the yellow in figure 7b, or something else.

Response 11: We have now clarified this, where we write near bottom of page 12:

“... leading to a significant widening of the frequency spectrum seen in yellow and green in Fig. 7b, and to anomalous absorption (see the Supplementary Information) of the probe and pump waves.”

12. 356 The combined pump + probe in Fig. 7b are significantly absorbed, and only weak traces of returned pump and probe signals are seen in the spectrogram.

371 The pumping near gyroharmonics suppresses the formation of small-scale field aligned striations [62] on which the pump wave can be anomalously absorbed near the upper hybrid layer [63-65]. The present experiment was conducted with frequencies away from the electron gyroharmonics, which may have resulted in the production of field-aligned striations, leading to the absorption of the O mode wave. The resulting decrease of the O mode pump amplitude below the threshold for SBS instability may explain the absence of natural SBS lines in the experimental spectrograms in Fig. 2 for O mode polarization (Runs I, IV and V). The 1D numerical model neglects the effect of field-aligned striations, and the unattenuated O mode pump wave is allowed to reach the critical layer where its amplitude exceeds the threshold for natural SBS generation, resulting in the lines in Fig. 7b.

Is "on which the pump wave can be anomalously absorbed near the upper hybrid layer" the same situation as the PDI and Langmuir turbulence and other absorption-related statements in the manuscript? The specificity of this statement raises potential confusion about that which should be clarified.

Response 12: We have slightly modified and clarified the statements near end of the second paragraph on page 13 as:

"The pumping near gyroharmonics suppresses the formation of small-scale field aligned striations [70] on which the pump wave can be anomalously absorbed (see Supplementary Information) near the upper hybrid layer [54-56]. The present experiment was conducted with frequencies away from the electron gyroharmonics, which may have resulted in the production of field-aligned striations, leading to anomalous absorption of the O mode wave at the upper hybrid layer. The resulting decrease of the O mode pump amplitude below the threshold for SBS instability at higher altitudes near the critical layer may explain the absence of natural SBS lines in the experimental spectrograms in Fig. 2 for O mode polarization (Runs I, IV and V). The 1D numerical model neglects the effect of field-aligned striations, and the unattenuated O mode pump wave is allowed to reach the critical layer where its amplitude exceeds the threshold for natural SBS generation, resulting in the lines in Fig. 7b."

We hope that this clarifies the absorption at the upper hybrid layer. Details of the absorption processes are discussed in the Supplementary Information.

13. 389 oblique waves are subject to less significant anomalous absorption, which may explain the stronger return signal of the O mode wave observed in the experiment compared to that of the simulation.

401 The O mode is significantly absorbed by the ionosphere, attributed to conversion to electrostatic waves at the upper hybrid and critical layers.

Here both upper hybrid and critical layers are mentioned in regard to O mode absorption, yet on line 371+ only the upper hybrid layer, with striations, is mentioned. Clarification is needed here also.

Response 13: For clarity, we have modified the above statements near end of page 13 as

"As seen in Figs. 5b and 5d, O mode rays at oblique angles to the vertical can have their turning points at lower altitudes, below both the critical layer ($X = 1$) and the upper hybrid layer ($X + Y^2 = 1$). Consequently, these oblique waves are subject to less significant anomalous absorption, which may explain the stronger return signal of the O mode wave observed in the experiment compared to that of the simulation."

and first paragraph on page 14:

“The O mode is significantly absorbed by the ionosphere, attributed to conversion to electrostatic upper hybrid waves on magnetic field aligned striations at the upper hybrid layer and to Langmuir waves on ion fluctuations at the critical layer (see Supplementary Information).”

The anomalous absorption at the upper hybrid and critical layers is discussed in the text (see question 9 above the next question 14 below) and in the Supplementary Information. We hope in the revised text the reader will be able to distinguish these absorption processes.

14. SEE is associated with striations at the upper hybrid layer. Are you saying that SEE is also originating at the critical layer, and/or that striations at the critical layer are also present and perhaps equally important for SEE and PDI/LT? Do lack of striations at the gyroharmonics mean that the PDI/LT is not occurring at the gyroharmonics, or perhaps at the critical level but not at the UH level? This should be part of the clarification.

Response 14: Different types of SEE are generated at the critical layer and upper hybrid layer. In the new section “Stimulated electromagnetic emissions (SEE) in ionospheric heating experiments” in the Supplementary Information we briefly discuss the most common features of the SEE generated at these layers and give references to several review articles on the subject. Much work has been done on SEE in the past, and we can only give a very short summary here, but we hope that the cited review papers can guide the reader on this interesting subject.

In the main article we refer to the Supplementary Information near bottom of page 12:

“For O mode polarisation in Fig. 7b, the EM wave reaches the critical layer at $z = 219.91$ km where Langmuir turbulence is excited, leading to a significant widening of the frequency spectrum seen in yellow and green in Fig. 7b, and to anomalous absorption of the probe and pump waves; more details on these processes are given in the Supplementary Information.”

15. Bernhardt et al. 2009 continue:

> (2) an ion acoustic and an electromagnetic wave

This corresponds to SBS and, with a magnetic field, MSBS/MIBS processes.

Bernhardt et al. 2011 discuss stimulated ion Bernstein scattering (SIBS) at the second gyroharmonic, and Samimi et al. 2012 goes farther, discussing also upper hybrid/electron Bernstein and neutralized ion Bernstein (IB) waves.

A brief mention that this occurs at the second gyroharmonic, i.e. that SBS/MSBS is suppressed, and that SIBS appears, and perhaps a brief insight as to why, would be a helpful contribution to clarity and completeness.

Response 15: Thank you for pointing out the papers by Bernhardt et al. 2011 and Samimi et al. 2012 who discuss parametric decay of an EM wave into electrostatic upper hybrid/electron Bernstein waves and neutralized ion Bernstein waves, which distinctly different from the Brillouin process. Near the second gyroharmonic the production of field aligned striations is probably reduced, leading to a decreased electron heating and increase of ion Landau damping. This could damp out the SBS/MSBS near the upper hybrid layer and instead trigger the SIBS in which the unattenuated electromagnetic wave propagating through the upper hybrid layer decays into an upper hybrid wave and a neutralized ion cyclotron or ion Bernstein wave propagating across the magnetic field lines. One can also imagine that the formation of magnetic field aligned striations could work as a plasma grating to help the MSBS process when the transmitter is not too near electron gyroharmonics. Please see our Response **17** below where we discuss the MSBS and SIBS processes and our addition to the Supplementary Information.

16. > (3) an electron plasma and an electromagnetic wave

That is SRS, stimulated Raman scattering.

> (4) two electron plasma waves

That is the TPI, two-plasmon instability.

I think 3 and 4 have not been seen in the ionosphere. It would be interesting to comment on the prospects for seeing them, and what experimental geometry and conditions would be needed to do so.

Response 16: There were observations at half frequency $f_0/2$ (and second harmonic $2f_0$) by Derblom et al. (1989) which were attributed to SRS and/or TPI. Even though cause of these observations may be unclear, the paper contains a useful discussion about the possibilities of these two instabilities: The plasma inhomogeneity makes convective losses significant, and therefore the instability is thought to take place at an altitude where $\omega_{pe} = \omega_0/2$, i.e. where the plasma density is the quarter critical density and the Raman and two-plasmon instabilities would be absolute (non-convective) which could produce the emissions at $f_0/2$. We are not aware of any attempts to repeat these experiments with higher power. We briefly discuss this and other observations of SEE in the new section "Stimulated electromagnetic emissions (SEE) in ionospheric heating experiments" in the Supplementary Information.

Reference:

Derblom, H., Thidé, B., Leyser, T. B., Nordling, J. A., Hedberg, Å., Stubbe, P., Kopka, H., & Rietveld, M. (1989), Tromsø Heating Experiments: Stimulated emission at HF pump harmonic and subharmonic frequencies, *J. Geophys. Res.*, 94(A8), 10111– 10120, <https://doi.org/10.1029/JA094iA08p10111>

17. --- MSBS and SIBS

Related to my previous comments about the gyroharmonics:

In the Supplementary Information you write

102 The dispersion relation shows that the IA wave has a resonance ($k_{IA} \rightarrow \infty$) at $\omega_{IA} = \omega_{ci} \cos \theta$, a cut-off ($k_{IA} \rightarrow 0$) at $\omega_{IA} = \omega_{ci}$, and a forbidden frequency band between the resonance and cut-off.

104 The branch at $\omega_{IA} < \omega_{ci}$ is the magnetized IA wave, while the higher frequency branch $\omega_{IA} > \omega_{ci}$ is the electrostatic ion cyclotron wave.

106 for IA waves propagating at small angles to the magnetic field, the effect of the magnetic field on the IA wave may be neglected except very close to the cyclotron resonance

Close to the cyclotron resonance, or at larger angles to the magnetic field, the magnetic field becomes important and SBS becomes magnetized stimulated Brillouin scatter (MSBS).

Several papers discuss magnetized SBS (MSBS). For example, Bernhardt et al. 2010, in the abstract, write "Either an ion-acoustic wave with a frequency less than the ion cyclotron frequency (fCI) or an electrostatic ion cyclotron (EIC) wave just above fCI can be produced." and calls it MSBS, and Fu et al. 2020, in the introduction, discusses "IA and EIC emissions due to MSBS". (Both of those papers you have already cited.)

Other papers discuss SBS near the gyroharmonics but might not call it MSBS, for example that might be the case for Bernhardt et al. 2009, perhaps because it was an earlier paper?

Samimi et al. 2012 note that MSBS is suppressed at both the second and third gyroharmonics, and discuss upper hybrid/electron Bernstein and neutralized ion Bernstein (IB) waves, but do not call it SIBS.

Perhaps you could assist in clarifying what is meant by MSBS and SIBS e.g. in what cases they apply, which appears to be near the gyroharmonics and at oblique angles to the magnetic field. As explained in the manuscript, this is not the case for your paper but a clarifying comment would assist interested readers.

Mahmoudian et al. 2019 (<https://doi.org/10.1029/2019JA026594>) discuss MSBS for the case at Arecibo, with about 45 degrees to the magnetic field, and not too near a gyroharmonic. Mahmoudian et al. appear to see an MSBS ion acoustic line but not an ion cyclotron line.

Response 17: Thank you for clarifying the nomenclature of MSBS and SIBS. In the Supplementary Information, we have now added the following text about MSBS and SIBS in the paragraph above Fig. S5 on page 5:

“The scattering of the EM wave off these wave modes and off ion Bernstein waves are referred to as magnetized stimulated Bernstein scattering (MSBS), and occurs primarily where the EM wave propagates at oblique angles to the magnetic field and the transmitted frequency is not too close to electron cyclotron harmonics so that magnetic field aligned striations can be formed [4,9-11]. In stimulated ion Bernstein scattering (SIBS), which appears when the O mode frequency is tuned near an electron cyclotron harmonic [12,13], the EM wave is converted into a high-frequency upper hybrid wave on a low-frequency ion Bernstein wave propagating at large angles to the magnetic field, in a process similar to the parametric decay instability (PDI) discussed below.”

References:

- [4] Fu, H. Y., et al. Electron temperature inversion by stimulated Brillouin scattering during electron gyroharmonic heating at EISCAT. *Geophys. Res. Lett.* **47**, e2020GL089747 (2020). <https://doi.org/10.1029/2020GL089747>
- [9] Bernhardt, P. A., Selcher, C. A., Lehmborg, R. H., Rodriguez, S., Thomason, J., McCarrick, M., and Frazer, G. Determination of the electron temperature in the modified ionosphere over HAARP using the HF pumped Stimulated Brillouin Scatter (SBS) emission lines, *Ann. Geophys.* **27**, 4409–4427 (2009). <https://doi.org/10.5194/angeo-27-4409-2009>
- [10] Bernhardt, P. A. et al. Stimulated Brillouin scatter in a magnetized ionospheric plasma, *Phys. Rev. Lett.* **104**, 165004 (2010). <https://doi.org/10.1103/PhysRevLett.104.165004>
- [11] Mahmoudian, A., Nossa, E., Isham, B., Bernhardt, P. A., Briczinski, S. J., & Sulzer, M. NSEE yielding electron temperature measurements at the Arecibo Observatory. *J. Geophys. Res. Space Physics*, **124**, 3699–3708 (2019). <https://doi.org/10.1029/2019JA026594>
- [12] Bernhardt, P. A., Selcher, C. A. & Kowtha, S. Electron and ion Bernstein waves excited in the ionosphere by high power EM waves at the second harmonic of the electron cyclotron frequency, *Geophys. Res. Lett.* **38**, L19107 (2011). <https://doi.org/10.1029/2011GL049390>
- [13] Samimi, A. et al. On ion gyro-harmonic structuring in the stimulated electromagnetic emission spectrum during second electron gyro-harmonic heating, *Ann. Geophys.* **30**, 1584-1594 (2012). <https://doi.org/10.5194/angeo-30-1587-2012>

18. A comment on what might be done at Arecibo in regard to SBS, MSBS, SIBS, and beat-wave SBS, MSBS, and SIBS, would be interesting, especially since plans are being discussed to rebuild the Arecibo high-power HF facility, perhaps within the next two years. The initial restoration would likely be at 5.1 MHz and perhaps include the 5th harmonic, but a proposal for a new Arecibo heating facility might include heating down to the second harmonic (about 2 MHz at Arecibo) -- here you have an opportunity to assist in that effort, and it is a current and interesting topic.

Response 18: We are very excited to hear that there are plans to rebuild the Arecibo HF Facility potentially including new frequencies down to 2nd electron gyroharmonic; this would

indeed be a very interesting development. The unique conditions at mid-latitude with the oblique angle of the magnetic field to the vertical and enhanced levels of photoelectrons, etc., would complement the high-latitude facilities EISCAT and HAARP. We hope that the plans to rebuild the facility will be successful, and that it will have the capability of beat-wave transmissions, which would increase the possibilities of interesting experiments. Since the plans to rebuild the Arecibo HF facility are not yet official (?) we do not want to write too many details, but in the Discussion we now write:

“It would be of interest to compare beat-wave Brillouin interactions at the mid-latitude Arecibo HF Facility in Puerto Rico where the magnetic field is directed about 45° to the vertical, with EISCAT and other high-latitude facilities.”

19. --- experimental setup

In regard to my question about previous line 153

>> The probe and pump spectra are partially overlapping during the first three seconds due to the fundamental restriction $\Delta t |\Delta f| > 1$ with $\Delta t = 1$ s.

In the response you wrote:

> The fundamental restriction $\Delta t |\Delta f| > 1$ is for a pulse and its Fourier transform and reaches equality for a Gaussian pulse (with exact definitions of Δt and Δf). In our case the pulse is a 1s long square pulse in time but which has been windowed by a Hanning window. But we omit such detailed discussion here.

This seems the type of detail useful for explaining and perhaps sometime repeating the experiment, which perhaps should be included in the methods section or in supplementary information.

Response 19: Thank you for pointing this out. We have now added a short paragraph in the Methods, on page 16, discussing the windowing technique and fundamental limit on frequency resolution:

“Calculation of Spectrograms

The experimental spectrograms in Fig. 2 (and extended data Figs. 1–3) and simulated spectrograms in Fig. 7 are obtained by carrying out a Fourier transform of the wave electric field in time over the 1-second windows listed in Table 2 as the probe signal is stepped down in frequency with time. The received time series signal in the experiment and the simulated signal in the simulation are multiplied by a Hanning (sine-squared) window in time before carrying out the Fourier transforms, in order to increase the dynamic range in the spectrograms. The 1-second time windows lead to a minimum frequency resolution of the spectra which are resolved by steps of 1 Hz, with a small degree of frequency smoothing of the spectrum introduced by the Hanning window.”

We refer to the Methods in the text when discussing the spectrograms in Figs 2 and 7.

20. --- typos/grammar/references

In the supplementary information:

127 In Fig. 5 of the main article, Runs II and III have clearly

=> I think Fig 2 is meant.

Response 20: Yes, thank you for spotting this typo which we have corrected.

21. In the manuscript:

Transmitter and run and array are pretty generic things, and common nouns are not capitalized in English (if not at the start of a sentence) (check on usage for common vs proper nouns).

Some examples here from the manuscript:

127 while the Transmitters 3–12 were used for the pump

=> while transmitters 3–12 were used for the pump

125 EISCAT's antenna array 1

That is correct.

265 EISCAT Array 1

Should be "array".

Response 21: Thank you. We have changed 'Array' and 'Transmitter' to 'array' and 'transmitter' throughout the manuscript and Supplementary Information where applicable, but have kept Run I, Run II and so on (with Roman numerals) to denote the experimental runs.

22. 128 divided into 6 Runs

divided into 6 runs

131 Each Run

=> each run

Response 22: Thank you. We have changed these.

23. 137 heater Array 1, where transmitters 3 – 12 were used for the pump and Transmitters 1 and 2

=> heater array 1 ... transmitters 3 – 12 ... transmitters 1 and 2

Response 23: We have changed 'Array' and 'Transmitter' to 'array' and 'transmitter' throughout the manuscript and Supplementary Information where applicable.

24. 140 Run I

=> run 1 (note digit 1 vs Roman numeral I)

Response 24: We would prefer to keep Run I, Run II and so on with Roman numerals to denote the experimental runs, to be consistent with the tables and figures.

25. 152 Runs

=> runs

and so on.

144 Experimental observations of SBS interaction

Figure 2 is between the title and the text but I guess this will be fixed by the journal before publication.

186 arb. Units, linear scale

=> arb. units, linear scale

Response 25: Thank you. We have introduced these corrections.

26. 407 natural SBS originating near the critical layer (denoted SBS-1) strengthens, while SBS originating from the upper hybrid layer (SBS-2) is suppressed.

Was this notation (SBS-1 and SBS-2) originated by [31] Bernhardt et al 2009, or by whom? It is specific enough that it might be good to reference that.

Response 26: Yes, Reviewer #2 is correct that the notation SBS-1 and SBS-2 were originally introduced by Bernhardt et al 2009. The two processes were also discussed by Norin et al. (2009) but they did not use the notation SBS-1 and SBS-2. We have now clarified the notation by writing in the first paragraph on page 14 that

“Observations of natural SBS are found to originate primarily from two regions: near the turning point of the O mode wave close to the critical layer and near the upper hybrid layer [33], denoted SBS-1 and SBS-2, respectively [34]. It has been found in experiments [36,40] that when the transmitter is tuned near an electron gyroharmonic, SBS-1 strengthens while SBS-2 is suppressed.”

References:

[33] Norin, L., Leyser, T. B., Nordblad, E., Thidé, B. & McCarrick, M. Unprecedentedly strong and narrow electromagnetic emissions stimulated by high-frequency radio waves in the ionosphere, *Phys. Rev. Lett.* **102**, 065003 (2009). <https://doi.org/10.1103/PhysRevLett.102.065003>

[34] Bernhardt, P. A. et al. Determination of the electron temperature in the modified ionosphere over HAARP using the HF pumped Stimulated Brillouin Scatter (SBS) emission lines, *Ann. Geophys.* **27**, 4409-4427 (2009). <https://doi.org/10.5194/angeo-27-4409-2009>.

[36] Fu, H. Y., et al. Electron temperature inversion by stimulated Brillouin scattering during electron gyroharmonic heating at EISCAT. *Geophys. Res. Lett.* **47**, e2020GL089747 (2020). <https://doi.org/10.1029/2020GL089747>

[40] Fu, H. Y. et al. Stimulated Brillouin scattering during electron gyro-harmonic heating at EISCAT, *Ann. Geophys.* **33**, 983–990 (2015). <https://doi.org/10.5194/angeo-33-983-2015>

27. 470 Langmuir

Extra I.

Response 27: Thank you. We have corrected this.

28. 564 [31]

The doi link to Norin et al. 2009 should be

<https://doi.org/10.1103/PhysRevLett.102.065003>

Response 28: Thank you. We have corrected this.

29. ---

In regard to my comment about explaining the meaning of anomalous absorption, you could explain it from the historical point of view, as absorption that was above that which was expected from collisional heating, and cite one of the early publications, in a similar way that you did in your explanation of the Spitz angle.

Response 29: Thank you for this suggestion. We have now added the new section “Collisional and anomalous absorption of EM waves” in the Supplementary Information where we discuss collisional processes and collisional absorption with references to early works on magneto-ionic theory, as well as anomalous absorption due to nonlinearity near the critical layer and due to magnetic field aligned striations at the upper hybrid layer. We hope this will be informative and helpful for the readers.

We again thank Reviewer #2 for the positive and helpful comments and suggestions that have improved our manuscript significantly and have eliminated a number of typos. We hope that the revised manuscript is now suitable for publication in Nature Communications.

Yours sincerely

Bengt Eliasson

For and on behalf of all authors

REVIEWER COMMENTS

Reviewer #2 (Remarks to the Author):

I thank the authors for the additional helpful explanations and clarifications in the manuscript.

The authors have written an excellent paper. I don't want to unfairly put too much burden on them, but I mention a few things related mostly to the changes and additions, which, since those are new, I had not seen before.

The things I mention, especially in a paper such as this in a journal with more general readership, and given the authors' depth of experience in these questions, are an important opportunity to help clarify some common confusions in the field.

They may be my own confusions as much as broader confusions in the field, but my defense is that if I am confused, so must be many other colleagues, and in particular this journal is aimed at a broader audience of colleagues who may be less familiar with the field than typical readers of this topic.

This article has the potential to be a fundamental reference for this interesting and important area of plasma physics, and consideration of these details now could help students and postdocs, and anyone new to the field, get off to a good start.

My best regards to the authors.

In their reply to point 18:

433 > It would be of interest to compare beat-wave Brillouin interactions at the mid-latitude Arecibo HF Facility in Puerto Rico where the magnetic field is directed about 45° to the vertical, with EISCAT and other high-latitude facilities.

Rather than single out Arecibo, and since there are only a few HF facilities, I suggest including the others, plus Jicamarca at the magnetic equator, where an HF facility has been proposed in the past.

HAARP was already mentioned in the article, and the Sura facility in Russia could also be mentioned, which would then account for all current and near-future HF facilities and Jicamarca, which could host a unique and interesting HF facility.

Also it should be noted that the Arecibo HF facility is not currently operational. The current magnetic field angle at the Arecibo HF facility is currently 47.5 degrees to vertical and rapidly increasing, so the version below is perhaps more complete and a bit more accurate:

> 433 => It would be of interest to compare beat-wave Brillouin interactions at mid and low latitudes with EISCAT and HAARP. An HF facility at the Jicamarca Radio Observatory, located at the geomagnetic equator, has been proposed in the past; experiments may be possible in the near future at a restored Arecibo HF facility in Puerto Rico, where the magnetic field is directed nearly 50° to the vertical; and related experiments might also be done at the Sura HF facility in Russia, where the magnetic field is about 30° to vertical [Kagan et al. 2006, Grach et al. 2007].

It would be particularly interesting and perhaps motivating to point out any especially illuminating comparisons that might be done, e.g. are there particular features that could be investigated during HF pumping at Sura, Arecibo, and Jicamarca that might be of special interest for experiments with natural and beat-wave SBS, MSBS, and SIBS, as well as for the Raman and two-plasmon instabilities, that might be mentioned? [21] in the Supplementary Information ([53] in the main article) may already explain some of this, but a comment in the text would be interesting and helpful.

Kagan et al. specify the field angle at Sura as 29 degrees but that was in 2006, I'm not sure how it might be changing:

<https://ui.adsabs.harvard.edu/abs/2006RRPRA..11..221K/>

Grach et al. 2007 also mention the field angle at Sura:

<https://angeo.copernicus.org/articles/25/689/2007/>

One detail, note "Sura" is a place name, like Arecibo and Jicamarca, not an acronym like EISCAT and HAARP, so "Sura" is correct, and "SURA" is incorrect. I suspect that in Russian all caps might be normally used in the case of a name like Sura, and some have used SURA in English also, but that is not correct.

In their reply to points 5 and 7 the authors write:

337 > In this process, the up-going O mode is anomalously absorbed (see Fig. S6 in the Supplementary Information), resulting in a much weaker reflected wave and no visible oscillating standing wave pattern in E_x and E_y below the cut-off. The interaction between the O mode and the Langmuir turbulence also generates a lower amplitude Z mode wave that propagates above the O mode cut-off up to the Z mode cut-off where it is reflected, setting up a standing wave pattern below its reflection point.

The power reaching the Z mode cutoff is likely less than at the O mode cutoff. Could the Airy amplification be enough greater at the Z cutoff to make up for the field being so weak at the O mode cutoff, and to make up for an loss before reaching the Z mode cutoff, to explain why the Airy pattern is not visible at O cutoff and is visible at Z cutoff?

In the Supplementary Information, near end of page 6, in regard to explaining the origin of the term anomalous absorption, I suggest an additional sentence, something like replacing this sentence:

> However, experiments show that a large amplitude O mode wave can be anomalously absorbed in the F2 region [18,19].

Replacing with these two, as a way of taking the opportunity to explain the term anomalous absorption:

=> However, experiments show that a large amplitude O mode wave can undergo absorption in the F2 region that is much greater than collisional absorption [18,19,29,57]. When first discovered, this additional absorption was called anomalous absorption, a term which is now typically understood to mean absorption through instability processes.

Maybe include [29] and [57] as references (as added above).

Continuing in the Supplementary Information, near end of page 6, immediately following the above (also at line 81 in the main article):

> Within a few milliseconds after switch-on of the transmitter, there is a drop of about 10 dB in the reflected power.

This sounds like the relaxation of what has been called the overshoot. A reference is needed, or maybe references.

Also, Duncan and Sheerin 1985

<https://doi.org/10.1029/JA090iA09p08371>

write

> These measurements resolve the miniovershoot (2-10 ms after HF on) and main overshoot (20-40 ms after HF on) plasma wave excitation features.

> The main overshoot ... enhancements of a factor of 10 greater than the steady state ...

> in one case, a small plasma line enhancement is detected at very early times (< 1 ms) distinct from the miniovershoot

> the interaction layer is observed to broaden by several hundred meters during the main overshoot relaxation.

How many ms is the "few milliseconds" that you mention? Would that be after the miniovershoot, or after the main overshoot? I guess it would be after the main overshoot, so it is a few tens of ms? Is the distinction between the pre-mini, mini, and main overshoot real? Or is there just one overshoot, as your explanation implies? Perhaps overshoot is an incompletely understood phenomenon.

Continuing in the Supplementary Information, near end of page 6:

> becomes tilted

Maybe

=> becomes refracted

A reference would be good, perhaps Leyser's paper (maybe Leyser et al.) showing the E field components and Airy pattern. I don't remember for sure the reference to it, and don't have access to a copy right now, but maybe

<https://doi.org/10.1103/PhysRevLett.63.1145>

or

<https://doi.org/10.1029/JA094iA08p10111>

or referenced in one of those.

Supplementary Information, near end of page 6:

> the oscillating two-stream instability (OTSI) is ...

This could be written as

=> the OTSI is ...

(in parallel with the "PDI" abbreviation earlier in the sentence).

Continuing:

> ... OTSI ... is a 4-wave process in which the EM wave decays into two counter-propagating Langmuir waves and a purely growing (non-propagating) ion fluctuation. In the non-linear stage, large amplitude Langmuir waves with group velocities smaller than the ion-acoustic velocity may also undergo a 4-wave modulational instability [20] which, together with the OTSI, leads to strong, cavitating Langmuir turbulence [21].

This is confusing. What 4-wave modulational instability is being referred to in addition to the OTSI, which is also 4-wave modulational instability...?

[20] might explain it, but a qualitative explanation of what the second 4-wave modulational instability is, equivalent to what is given for the OTSI in the previous sentence, would be worthwhile, and would put your MI comment on an equal footing in the text with your OTSI comment. And a note about whether both the new MI, along with the OTSI, are both required in the evolution to cavitating LT.

I think the OTSI is an MI but you might be saying that they are distinct processes. An explanation here would be valuable in that regard.

I suggest references to Fejer 1988 and Robinson 1997:

Robinson 1997

<https://doi.org/10.1103/RevModPhys.69.507>

is very serious and concerning in writing:

> For historical reasons, some modulational instabilities have been termed oscillating two-stream instabilities (OTSI) in much of the older literature, and even in some recent work. This terminology has proved to be highly confusing and should be rigorously avoided.

Fejer 1988

[https://doi.org/10.1016/0273-1177\(88\)90371-7](https://doi.org/10.1016/0273-1177(88)90371-7)

agrees that OTSI is a particular case of MIs:

> In the OTSI a spatially periodic but temporally aperiodic plasma density perturbation scatters the electromagnetic pump wave into a standing Langmuir wave. The ponderomotive force due the interference pattern of the pump wave with the standing Langmuir wave generates the temporally aperiodic density perturbation and thus completes the feedback loop. Such parametric instabilities in which temporally aperiodic density perturbations scatter the pump wave and thus spatially modulate the pump field, are usually called modulational instabilities.

This may be a simple matter, but Robinson and Fejer should be considered in regard to being clear about OTSI vs MI. This is an opportunity to clear up a old point of confusion.

In the Supplementary Information, near bottom of page 7:

> On a much longer time-scale of about a second, there attributed to the development of magnetic field aligned density striations through a thermal instability at the upper hybrid layer, a few km below the critical layer.

Is this the same as discussed by Fejer 1988, who writes:

> Short-scale field aligned irregularities are produced by a thermal parametric instability of the modulational type in which, however, the irregularities scatter the pump wave into Langmuir waves, rather than into electromagnetic waves as in the case of thermal self-focusing. The theory of these instabilities was developed in /33,34,35,36,37,38,39/.

In the Supplementary Information, near top of page 8:

> However, the formation of striations is suppressed when the transmitted frequency is close to electron cyclotron harmonics [32,33]

A brief reason why would be helpful to interested readers.

In the Supplementary Information, page 8, SEE section:

> The PDI/OTSI and resulting Langmuir turbulence near the critical layer gives rise to the narrow downshifted continuum a few kHz below the pump frequency

A reference or two here would be good -- are there references specific to this issue?

> the downshifted maximum and broad upshifted maximum originate from the upper hybrid layer once magnetic field aligned striations have developed. The down-shifted maximum is about 10 kHz (the lower hybrid frequency) below the pump frequency and is attributed to upper hybrid and lower hybrid turbulence. The broad upshifted maximum is several tens of kHz above the pump frequency and is generated at the upper hybrid layer due to the coupling to electron Bernstein waves at frequencies between electron

gyroharmonics.

Also a specific reference or two for these important cases would be good, more specific than the general references of the reviews.

Typos:

I noticed a few small things.

Reference 20 is missing the year.

Supplementary Information, page 7, the equation should be S7.

Missing d in field:

523 => wave electric field in time

No "-" in "downshifted" or "upshifted": various places, main article lines 68, 119, 165, 166, 4 on page 6, 366, 381, 390, and 173, 360, 386, 389, also Supplementary Information, page 8,

=> The downshifted maximum

Note a few lines early "downshifted" is used:

> the narrow downshifted continuum

and "upshifted" is also used in the main article and in the Supplementary Information.

No "-" in timescale:

87 > longer timescale of about

or

87 > longer time scale of about

(according to the dictionary they look to me to be the same meaning)

No "-" in daytime:

Supplementary Information mid page 5

> during daytime conditions

This next thing may be a question of journal style, but I'd guess no () needed around equation numbers; () are part of the equation format, not part of the equation number:

77 => Eq. S5

107 => Eq. M3

307 => by Eq. 18 in Ref. [67]

362 => in Eqs. 1a and 1b.

386 => below Eqs. 1a and 1b.

467 => Equation M1 describes

469 => and Eq. M2

487 => of Eq. M4

Supplementary Information mid page 5

=> [cf. Eq. M3 in the main article]

Note parentheses () are not used around figure numbers or table numbers e.g.:

117 > Figure S5

288 > (cf. Table 1)

Somewhat similar thought for the reference numbers, [] are not part of the reference number but are shorthand for "reference", so "Ref. [67]" is redundant.

307 => in Ref. 67

or

307 => in [67]

Supp Info bottom of page 2:

=> (cf. Section 4 of Ref. 5)

Supp Info bottom of page 3:

=> Ref. 7

Supp Info bottom of page 3, since "Sect." is used just once I suggest no abbreviation:

Supp Info bottom of page 3:

=> Section 6.6 of

I'm not sure if equation, eq., reference, ref., section, figure, and table should be capitalized or small. Based only on grammar it seems to me they should not be capitalized but that's a journal style question I guess.

3 August, 2021

We thank Reviewer #2 for the insightful comments and suggestions which have helped to improve our manuscript significantly. Below we list our point-by-point response to Reviewer #2's comments and our changes to the manuscript. All changes are indicated in yellow in the manuscript:

Reviewer #2 (Remarks to the Author):

I thank the authors for the additional helpful explanations and clarifications in the manuscript.

The authors have written an excellent paper. I don't want to unfairly put too much burden on them, but I mention a few things related mostly to the changes and additions, which, since those are new, I had not seen before.

The things I mention, especially in a paper such as this in a journal with more general readership, and given the authors' depth of experience in these questions, are an important opportunity to help clarify some common confusions in the field.

They may be my own confusions as much as broader confusions in the field, but my defense is that if I am confused, so must be many other colleagues, and in particular this journal is aimed at a broader audience of colleagues who may be less familiar with the field than typical readers of this topic.

This article has the potential to be a fundamental reference for this interesting and important area of plasma physics, and consideration of these details now could help students and postdocs, and anyone new to the field, get off to a good start.

My best regards to the authors.

Response: We thank Reviewer #2 again for the insightful and constructive comments. Below we address the remaining questions, mainly about the new material added in the previous revision, and hope that the revised manuscript will now be suitable for publication.

In their reply to point 18:

433 > It would be of interest to compare beat-wave Brillouin interactions at the mid-latitude Arecibo HF Facility in Puerto Rico where the magnetic field is directed about 45° to the vertical, with EISCAT and other high-latitude facilities.

Rather than single out Arecibo, and since there are only a few HF facilities, I suggest including the others, plus Jicamarca at the magnetic equator, where an HF facility has been proposed in the past.

HAARP was already mentioned in the article, and the Sura facility in Russia could also be mentioned, which would then account for all current and near-future HF facilities and Jicamarca, which could host a unique and interesting HF facility.

Also it should be noted that the Arecibo HF facility is not currently operational. The current magnetic field angle at the Arecibo HF facility is currently 47.5 degrees to vertical and rapidly increasing, so the version below is perhaps more complete and a bit more accurate:

> 433 => It would be of interest to compare beat-wave Brillouin interactions at mid and low latitudes with EISCAT and HAARP. An HF facility at the Jicamarca Radio Observatory, located at the geomagnetic equator, has been proposed in the past; experiments may be possible in the near future at a restored Arecibo HF facility in Puerto Rico, where the magnetic field is directed nearly 50° to the vertical; and related experiments might also be done at the Sura HF facility in Russia, where the magnetic field is about 30° to vertical [Kagan et al. 2006, Grach et al. 2007].

It would be particularly interesting and perhaps motivating to point out any especially illuminating comparisons that might be done, e.g. are there particular features that could be investigated during HF pumping at Sura, Arecibo, and Jicamarca that might be of special interest for experiments with natural and beat-wave SBS, MSBS, and SIBS, as well as for the Raman and two-plasmon instabilities, that might be mentioned? [21] in the Supplementary Information ([53] in the main article) may already explain some of this, but a comment in the text would be interesting and helpful.

Kagen et al. specify the field angle at Sura as 29 degrees but that was in 2006, I'm not sure how it might be changing:

<https://ui.adsabs.harvard.edu/abs/2006RRPRA..11..221K/>

Grach et al. 2007 also mention the field angle at Sura:

<https://angeo.copernicus.org/articles/25/689/2007/>

Response: Thank you for this suggestion which we have now introduced and extended in the Discussion, as

“It would be of interest to compare beat-wave Brillouin interactions at mid and low latitudes with EISCAT and HAARP. An HF facility at the Jicamarca Radio Observatory, located at the geomagnetic equator, has been proposed in the past; experiments may be possible in the near future at a restored Arecibo HF facility in Puerto Rico, where the magnetic field is directed nearly 50° to the vertical; and related experiments might also be done at the Sura HF facility in Russia, where the magnetic field is about 30° to the vertical [71,72]. The equatorial region has unique features that distinguishes it from the midlatitude and auroral regions with a potential for interesting experiments involving Langmuir turbulence-driven suprathermal electrons and associated currents and artificial plasma layers along the magnetic field lines in the horizontal direction [53], and where the geometry may enable beatwave Brillouin-driven ion Bernstein waves in the vertical direction.”

References:

[53] Eliasson, B. & Papadopoulos, K. HF wave propagation and induced ionospheric turbulence in the magnetic equatorial region, *J. Geophys. Res. Space Physics* **121**, 2727–2742 (2016). <https://doi.org/10.1002/2015JA022323>

[71] Kagan, L. M. *et al.* Optical and radio frequency diagnostics of the ionosphere over the Sura facility: Review of results, *Radio Phys. Radio Astron.* **11**(3), 221–241 (2006). <http://www.rpra-journal.org.ua/index.php/ra/article/view/641>

[72] Grach, S. M., Kosch, M. J., Yashnov, V. A., Sergeev, E. N., Atroshenko, M. A., and Kotov, P. V. On the location and structure of the artificial 630-nm airglow patch over Sura facility, *Ann. Geophys.* **25**, 689–700 (2007). <https://doi.org/10.5194/angeo-25-689-2007>

One detail, note "Sura" is a place name, like Arecibo and Jicamarca, not an acronym like EISCAT and HAARP, so "Sura" is correct, and "SURA" is incorrect. I suspect that in Russian all caps might be normally used in the case of a name like Sura, and some have used SURA in English also, but that is not correct.

Response: Thank you for pointing this out; yes we are aware that Sura is a place name and therefore should be written "Sura" and not "SURA".

In their reply to points 5 and 7 the authors write:

337 > In this process, the up-going O mode is anomalously absorbed (see Fig. S6 in the Supplementary Information), resulting in a much weaker reflected wave and no visible oscillating standing wave pattern in E_x and E_y below the cut-off. The interaction between the O mode and the Langmuir turbulence also generates a lower amplitude Z mode wave that propagates above the O mode cut-off up to the Z mode cut-off where it is reflected, setting up a standing wave pattern below its reflection point.

The power reaching the Z mode cutoff is likely less than at the O mode cutoff. Could the Airy amplification be enough greater at the Z cutoff to make up for the field being so weak at the O mode cutoff, and to make up for an loss before reaching the Z mode cutoff, to explain why the Airy pattern is not visible at O cutoff and is visible at Z cutoff?

Response: Yes, in the below zoomed-in figure one can see some degree of swelling of the Z mode near its cut-off at $z=227.55$ km. The mode converted Z mode wave is not monochromatic in time but has a frequency spectrum several kHz wide due to the Langmuir turbulence, so the figure below shows only a snapshot in time of the Z mode electric field. At lower altitudes, away from the Z mode cut-off, the wavelength becomes shorter and the Z mode becomes increasingly electrostatic with a gradual increase in the amplitude of the longitudinal electric field E_z as the Z mode approaches its electrostatic resonance. The Airy pattern is visible only near the Z mode cut-off due to i) swelling of the wave due to a decrease of the group velocity and ii) an increase of the wavelength near the cut-off.

We have now written in the text on lines 339-345 that

“The interaction between the O mode and the Langmuir turbulence also generates a lower amplitude Z mode wave that propagates above the O mode cut-off up to the Z mode cut-off at $z = 227.55$ km where it is reflected, setting up an Airy-like standing wave pattern below its reflection point with swelling of the amplitude and increase of the wavelength. At lower altitudes the wavelength decreases and the Z mode electric field becomes increasingly electrostatically polarized along the z direction as it approaches its electrostatic resonance just below the O mode cut-off (see the Supplementary Information).”

In the Supplementary Information, near end of page 6, in regard to explaining the origin of the term anomalous absorption, I suggest an additional sentence, something like replacing this sentence:

> However, experiments show that a large amplitude O mode wave can be anomalously absorbed in the F2 region [18,19].

Replacing with these two, as a way of taking the opportunity to explain the term anomalous absorption:

=> However, experiments show that a large amplitude O mode wave can undergo absorption in the F2 region that is much greater than collisional absorption [18,19,29,57]. When first discovered, this additional absorption was called anomalous

absorption, a term which is now typically understood to mean absorption through instability processes.

Maybe include [29] and [57] as references (as added above).

Response: Thank you. On lines 161-164, we have adopted the formulation suggested by Reviewer #2 as:

“However, experiments show that a large amplitude O mode wave can undergo absorption in the F2 region that is much greater than collisional absorption [18-22]. When first discovered, this additional absorption was called anomalous absorption, a term which is now typically understood to mean absorption through parametric instability processes.”

References:

[18] Leyser, T. B. Stimulated electromagnetic emissions by high-frequency electromagnetic pumping of the ionospheric plasma, *Space Sci. Rev.* **98**(3–4), 223–328 (2001).

<https://doi.org/10.1023/A:1013875603938>

[19] Fejer, J. A. & Kopka, H. The Effect of Plasma instabilities on the ionospherically reflected wave from a high-power transmitter, *J. Geophys. Res.* **86**, 5746–5750 (1981).

<https://doi.org/10.1029/JA086iA07p05746>

[20] Mjølhus, E. Anomalous absorption and reflection in ionospheric radio modification experiments, *J. Geophys. Res.* **90**(A5), 4269–4279 (1985).

<https://doi.org/10.1029/JA090iA05p04269>

[21] Mjølhus, E. Theoretical model for long time stimulated electromagnetic emission generation in ionospheric radio modification experiments, *J. Geophys. Res.* **103**(A7), 14711–14729 (1998). <https://doi.org/10.1029/98JA00927>

[22] Eliasson, B. & Papadopoulos, K. Numerical study of anomalous absorption of O mode waves on magnetic field-aligned striations, *Geophys. Res. Lett.* **42**, 2603–2611 (2015).

<https://doi.org/10.1002/2015GL063751>

Continuing in the Supplementary Information, near end of page 6, immediately following the above (also at line 81 in the main article):

> Within a few milliseconds after switch-on of the transmitter, there is a drop of about 10 dB in the reflected power.

This sounds like the relaxation of what has been called the overshoot. A reference is needed, or maybe references.

Also, Duncan and Sheerin 1985

<https://doi.org/10.1029/JA090iA09p08371>

write

> These measurements resolve the miniovershoot (2-10 ms after HF on) and main overshoot (20-40 ms after HF on) plasma wave excitation features.

> The main overshoot ... enhancements of a factor of 10 greater than the steady state ...

> in one case, a small plasma line enhancement is detected at very early times (< 1 ms) distinct from the miniovershoot

> the interaction layer is observed to broaden by several hundred meters during the main overshoot relaxation.

How many ms is the "few milliseconds" that you mention? Would that be after the miniovershoot, or after the main overshoot? I guess it would be after the main overshoot, so it is a few tens of ms? Is the distinction between the pre-mini, mini, and main overshoot real? Or is there just one overshoot, as your explanation implies? Perhaps overshoot is an incompletely understood phenomenon.

Response: The drop of reflected power that we were discussing is before the overshoot effects set in. We have now reformulated the sentence "Within a few milliseconds after switch-on of the transmitter, there is a drop of about 10 dB in the reflected power" with more details on lines 164-169 as

"During a certain time after arrival of the leading edge of the reflected pump signal, the intensity of the reflected signal remains almost constant. This corresponds to the linear growth phase of the instabilities. The duration of the linear phase is of the order 4-5ms at an injected ERP of 12 MW but is significantly shorter 0.1–0.2 ms for a higher ERP of 1.7 GW [23]. After the linear phase, anomalous absorption sets in, and the intensity of the reflected signal abruptly drops by about 10-30 dB, with higher absorption at higher injected power."

The overshoot effects are interesting and seem to be still under active research and may therefore be incompletely understood. We thank Reviewer #2 for the relevant reference to Duncan and Sheerin (1985). On lines 188-191, we discuss the overshoot effects as:

"The turbulence leads to intriguing time and space dynamics of the ionospheric interaction region with overshoot phenomena on different time scales as studied using high resolution incoherent backscatter radar observations with possible explanations for these effects [50]."

References:

[23] Grach, S. M., Sergeev, E. N., Mishin, E. V. & Shindin, A. V. Dynamic properties of ionospheric plasma turbulence driven by high-power high-frequency radiowaves, *Phys. Usp.* **59**(11), 1091–1128 (2016). <https://doi.org/10.3367/UFNe.2016.07.037868>

[50] Duncan, L. M. & Sheerin, J. P. High-resolution studies of the HF ionospheric modification interaction region, *J. Geophys. Res.* **90**(A9), 8371– 8376 (1985). <https://doi.org/10.1029/JA090iA09p08371>

Continuing in the Supplementary Information, near end of page 6:

> becomes tilted

Maybe

=> becomes refracted

A reference would be good, perhaps Leyser's paper (maybe Leyser et al.) showing the E field components and Airy pattern. I don't remember for sure the reference to it, and don't have access to a copy right now, but maybe

<https://doi.org/10.1103/PhysRevLett.63.1145>

or

<https://doi.org/10.1029/JA094iA08p10111>

or referenced in one of those.

Response: Thank you for these suggestions. We have reformulated the sentence as

“During the linear phase, the wave electric field of the O mode becomes aligned with the background magnetic field (i.e. almost in the vertical direction at Tromsø) as the wave reaches the critical layer [6,14], ...”

Even though the references suggested by Reviewer #2 are interesting, we think the references to Mjølhus et al. (2003) and Lundborg & Thidé (1986) are more relevant to the structure of the wave electric field near the turning point.

References:

[6] Mjølhus, E., Helmersen E. & DuBois D. F. Geometric aspects of HF driven Langmuir turbulence in the ionosphere, *Nonlin. Proc. Geophys.* **10**, 151–177 (2003).

<https://doi.org/10.5194/npg-10-151-2003>

[14] Lundborg, B. & Thidé, B. Standing wave pattern of HF radio waves in the ionospheric reflection region: 2. Applications, *Radio Sci.* **21**(3), 486– 500 (1986).

<https://doi.org/10.1029/RS021i003p00486>

Supplementary Information, near end of page 6:

> the oscillating two-stream instability (OTSI) is ...

This could be written as

=> the OTSI is ...

(in parallel with the "PDI" abbreviation earlier in the sentence).

Response: Thank you. We have corrected this.

Continuing:

> ... OTSI ... is a 4-wave process in which the EM wave decays into two counter-propagating Langmuir waves and a purely growing (non-propagating) ion fluctuation. In the non-linear stage, large amplitude Langmuir waves with group velocities smaller than the ion-acoustic velocity may also undergo a 4-wave modulational instability [20] which, together with the OTSI, leads to strong, cavitating Langmuir turbulence [21].

This is confusing. What 4-wave modulational instability is being referred to in addition to the OTSI, which is also 4-wave modulational instability...?

[20] might explain it, but a qualitative explanation of what the second 4-wave modulational instability is, equivalent to what is given for the OTSI in the previous sentence, would be worthwhile, and would put your MI comment on an equal footing in the text with your OTSI comment. And a note about whether both the new MI, along with the OTSI, are both required in the evolution to cavitating LT.

I think the OTSI is an MI but you might be saying that they are distinct processes. An explanation here would be valuable in that regard.

I suggest references to Fejer 1988 and Robinson 1997:

Robinson 1997

<https://doi.org/10.1103/RevModPhys.69.507>

is very serious and concerning in writing:

> For historical reasons, some modulational instabilities have been termed oscillating two-stream instabilities (OTSI) in much of the older literature, and even in some recent work. This terminology has proved to be highly confusing and should be rigorously avoided.

Fejer 1988

[https://doi.org/10.1016/0273-1177\(88\)90371-7](https://doi.org/10.1016/0273-1177(88)90371-7)

agrees that OTSI is a particular case of MIs:

> In the OTSI a spatially periodic but temporally aperiodic plasma density perturbation scatters the electromagnetic pump wave into a standing Langmuir wave. The ponderomotive force due the interference pattern of the pump wave with the standing Langmuir wave generates the temporally aperiodic density perturbation and thus completes the feedback loop. Such parametric instabilities in which temporally

aperiodic density perturbations scatter the pump wave and thus spatially modulate the pump field, are usually called modulational instabilities.

This may be a simple matter, but Robinson and Fejer should be considered in regard to being clear about OTSI vs MI. This is an opportunity to clear up a old point of confusion.

Response: We agree that the terminology of various instabilities can be confusing. We now briefly discuss the history behind the OTSI and the difference between Langmuir-drive OTSI (which is a modulational instability) and the EM wave-driven OTSI (which we think is an unstable mode conversion process and not a modulational instability), but would like to avoid the issue whether the name “OTSI” is right or wrong, on lines 175-188:

“The OTSI was originally studied by Nishikawa [24] who considered a Langmuir wave as the pump wave and was shown to be a 4-wave instability with a non-resonantly growing density perturbation rather than an ion-acoustic wave. Its name originates from an analogy with a two-stream instability between electrons and ions [25]. When the Langmuir wave is the pump, the OTSI is a modulational instability since only Langmuir waves and a density perturbation are involved. It was adopted by the laser interaction community as a 4 wave non-resonant purely growing instability but driven by a transverse wave i.e. the laser beam, and also by the ionospheric community for EM waves in the radio frequency range. This makes it distinct from the Langmuir driven case. The EM wave-driven OTSI drives two counter-propagating electrostatic Langmuir waves and a purely growing density perturbation [26,27] typically having wavelengths much shorter than that of the EM wave. When the driven Langmuir waves have grown to large amplitude, they can in turn undergo modulational instability and nucleation leading to localized Langmuir wave packets trapped in ion density cavities [28,29] and to strong, cavitating Langmuir turbulence [27,30,31].”

References:

[24] Nishikawa, K. Parametric excitation of coupled waves I. General formulation, *J. Phys. Soc. Japan* **24**(4), 916–922 (1968). <https://doi.org/10.1143/JPSJ.24.916>

[25] Sanmartin, J. R. Electrostatic plasma instabilities excited by a high-frequency electric field, *Phys. Fluids* **13**(6), 1533–1542 (1970). <https://doi.org/10.1063/1.1693114>

[26] Lashmore-Davies, C. N. The coupled mode approach to nonlinear wave interactions and parametric instabilities, *Plasma Physics* **17**(4), 281–303 (1975). <https://doi.org/10.1088/0032-1028/17/4/005>

[27] Eliasson, B. & Papadopoulos, K. HF wave propagation and induced ionospheric turbulence in the magnetic equatorial region, *J. Geophys. Res. Space Physics* **121**, 2727–2742 (2016). <https://doi.org/10.1002/2015JA022323>

[28] Bingham, R. & Lashmore-Davies, C. N. On the nonlinear development of the Langmuir modulational instability, *J. Plasma Phys.* **21**(1), 51–69 (1979). <https://doi.org/10.1017/S0022377800021644>

[29] Stenflo, L. & Shukla, P. K. Parametric instabilities and nucleation of coupled Langmuir and acoustic waves in a weakly-ionized plasma, *Astrophys. Space Sci.* **172**, 317–320 (1990). <https://doi.org/10.1007/BF00643325>

[30] Fejer, J. A. Physical processes of ionospheric heating experiments, *Adv. Space Res.* **8**(1), 261–270 (1988). [https://doi.org/10.1016/0273-1177\(88\)90371-7](https://doi.org/10.1016/0273-1177(88)90371-7)

[31] Robinson, P. A. Nonlinear wave collapse and strong turbulence, *Rev. Mod. Phys.* **69**(2), 507–573 (1997). <https://doi.org/10.1103/RevModPhys.69.507>

In the Supplementary Information, near bottom of page 7:

> On a much longer time-scale of about a second, there

attributed to the development of magnetic field aligned density striations through a thermal instability at the upper hybrid layer, a few km below the critical layer.

Is this the same as discussed by Fejer 1988, who writes:

> Short-scale field aligned irregularities are produced by a thermal parametric instability of the modulational type in which, however, the irregularities scatter the pump wave into Langmuir waves, rather than into electromagnetic waves as in the case of thermal self-focusing. The theory of these instabilities was developed in /33,34,35,36,37,38,39/.

Response: Yes, it is probably the same as discussed by Fejer (1988). We now cite Fejer (1988) and a few more authors who have discussed the linear and nonlinear phases of the thermal instability:

“On a much longer timescale of the order of a second, there is often a further drop of about 10-15 dB of the observed reflected power [36-38], which is attributed to the development of magnetic field aligned density striations through thermal instabilities [30,39-41] at the upper hybrid layer, a few km below the critical layer.”

References:

[30] Fejer, J. A. Physical processes of ionospheric heating experiments, *Adv. Space Res.* **8**(1), 261-270 (1988). [https://doi.org/10.1016/0273-1177\(88\)90371-7](https://doi.org/10.1016/0273-1177(88)90371-7)

[36] Cohen, R. & Whitehead, J. D. Radio-reflectivity detection of artificial modification of the ionospheric F layer, *J. Geophys. Res.* **75**(31), 6439–6445 (1970). <https://doi.org/10.1029/JA075i031p06439>

[37] Stubbe, P., Kopka, H., Jones, T. B. & Robinson, T. Wide band attenuation of radio waves caused by powerful HF waves: Saturation and dependence on ionospheric variability, *J. Geophys. Res.* **87**(A3), 1551–1555 (1982). <https://doi.org/10.1029/JA087iA03p01551>

[38] Getmantsev, G. G., *et al.* Some results of investigations of nonlinear phenomena in the F-layer of the ionosphere, *JETP Lett.* **18**, 364–366 (1973).
http://jetpletters.ru/ps/1569/article_24047.shtml

[39] Grach, S. M., Mityakov, N. A., Rapoport, V. O. & Trakhtengertz, V. Yu. Thermal parametric turbulence in a plasma, *Physica D* **2**(1), 102–106 (1981).
[https://doi.org/10.1016/0167-2789\(81\)90063-4](https://doi.org/10.1016/0167-2789(81)90063-4)

[40] Gurevich, A. V., Zybin, K. B. & Lukyanov, A. V. Stationary striations developed in the ionospheric modification, *Phys. Rev. Lett.* **75**(13), 2622 (1995).
<https://doi.org/10.1103/PhysRevLett.75.2622>

[41] Istomin, Ya. N. & Leyser, T. B. Small-scale magnetic field-aligned density irregularities excited by a powerful electromagnetic wave, *Phys. Plasmas* **4**, 817–828 (1997).
<https://doi.org/10.1063/1.872175>

In the Supplementary Information, near top of page 8:

> However, the formation of striations is suppressed when the transmitted frequency is close to electron cyclotron harmonics [32,33]

A brief reason why would be helpful to interested readers.

Response: We have slightly reformulated the sentence and cite a few references, as

“However, the formation of striations is suppressed when the transmitted frequency is close to electron gyroharmonics [42,43] due to linear dispersion effects that restrict the existence of upper hybrid waves involved in the formation of the striations [44-46], in which case the O mode can reach and excite Langmuir turbulence at the critical layer.”

and further down in the SEE section,

“For example, the downshifted peak and broad upshifted maximum are diminished for pump frequencies very close to electron gyroharmonics due to the suppression of upper hybrid waves near the gyroharmonics [44-46], which gives information about the local magnetic field strength at the upper hybrid layer and indicates that magnetic field aligned striations are suppressed at these frequencies.”

References:

[42] Honary, F., Stocker, A. J., Robinson, T. R., Jones, T. B. & Stubbe, P. Ionospheric plasma response to HF radio waves operating at frequencies close to the third harmonic of the electron gyrofrequency, *J. Geophys. Res.* **100**(A11), 21489– 21501 (1995).
<https://doi.org/10.1029/95JA02098>

[43] Honary, F. *et al.* First direct observations of the reduced striations at pump frequencies close to the electron gyroharmonics, *Ann. Geophys.* **17**, 1235–1238 (1999).
<https://doi.org/10.1007/s00585-999-1235-6>

[44] Mjølhus E. On the small scale striation effect in ionospheric radio modification experiments near harmonics of the electron gyro frequency, *J. Atm. Terr. Phys.* **55**(6), 907-918 (1993). [https://doi.org/10.1016/0021-9169\(93\)90030-3](https://doi.org/10.1016/0021-9169(93)90030-3)

[45] Leyser, T. B., Thidé, B., Waldenvik, M., Veszelei, E., Frolov, V. L., Grach, S. M. & Komrakov, G. P. Downshifted maximum features in stimulated electromagnetic emission spectra, *J. Geophys. Res.* **99**(A10), 19555–19568 (1994).
<https://dx.doi.org/10.1029/94JA01399>

[46] Grach, S.M., Thidé, B. & Leyser, T.B. Plasma waves near the double resonance layer in the ionosphere, *Radiophys. Quantum Electron.* **37**, 392–402 (1994).
<https://doi.org/10.1007/BF01045689>

In the Supplementary Information, page 8, SEE section:

> The PDI/OTSI and resulting Langmuir turbulence near the critical layer gives rise to the narrow downshifted continuum a few kHz below the pump frequency

A reference or two here would be good -- are there references specific to this issue?

Response: We have now cited Frolov et al. (2004) and Thidé et al. (2005) who discuss the narrow downshifted continuum and attribute it to the ponderomotive parametric instabilities (i.e. the results of the PDI and OTSI) just below the reflection point of the O mode:

“The PDI/OTSI and resulting Langmuir turbulence near the critical layer gives rise to the narrow downshifted continuum a few kHz below the pump frequency [47,48] ...”

References:

[47] Frolov, V. L. *et al.*, Ponderomotive narrow continuum (NCp) component in stimulated electromagnetic emission spectra, *J. Geophys. Res.* **109**, A07304 (2004).
<https://doi.org/10.1029/2001JA005063>

[48] Thidé, B., Sergeev, E. N., Grach, S. M., Leyser, T. B. & Carozzi, T. D. Competition between Langmuir and upper-hybrid turbulence in a high-frequency-pumped ionosphere, *Phys. Rev. Lett.* **95**, 255002 (2005). <https://doi.org/10.1103/PhysRevLett.95.255002>

> the downshifted maximum and broad upshifted maximum originate from the upper hybrid layer once magnetic field aligned striations have developed. The down-shifted maximum is about 10 kHz (the lower hybrid frequency) below the pump frequency and is attributed to upper hybrid and lower hybrid turbulence. The broad upshifted maximum is several tens of kHz above the pump frequency and is generated at the upper hybrid layer due to the coupling to electron Bernstein waves at frequencies between electron gyroharmonics.

Also a specific reference or two for these important cases would be good, more specific than the general references of the reviews.

Response: We have cited Carozzi et al. (2002) who discuss the observations of the down-shifted maximum and broad upshifted maximum in great detail during frequency sweeping through the 4th electron gyroharmonic.

“...the downshifted maximum and broad upshifted maximum originate from the upper hybrid layer once magnetic field aligned striations have developed [49].”

Reference:

[49] Carozzi, T. D., *et al.* Stimulated electromagnetic emissions during pump frequency sweep through fourth electron cyclotron harmonic, *J. Geophys. Res.* **107**(A9), 1253 (2002).
<https://doi.org/10.1029/2001JA005082>

Typos:

I noticed a few small things.

Reference 20 is missing the year.

Response: Thank you. We corrected Ref. 20 in the Supplementary Information.

Supplementary Information, page 7, the equation should be S7.

Response: Thank you, we corrected the equation number.

Missing d in field:

523 => wave electric field in time

Response: We have corrected this.

No "-" in "downshifted" or "upshifted": various places, main article lines 68, 119, 165, 166, 4 on page 6, 366, 381, 390, and 173, 360, 386, 389, also Supplementary Information, page 8,

=> The downshifted maximum

Note a few lines early "downshifted" is used:

> the narrow downshifted continuum

and "upshifted" is also used in the main article and in the Supplementary Information.

Response: Thank you. We have removed the hyphen in "upshifted" and "downshifted".

No "-" in timescale:

87 > longer timescale of about

or

87 > longer time scale of about

(according to the dictionary they look to me to be the same meaning)

Response: We changed "time-scale" to "timescale" everywhere.

No "-" in daytime:

Supplementary Information mid page 5

> during daytime conditions

Response: We changed to "daytime".

This next thing may be a question of journal style, but I'd guess no () needed around equation numbers; () are part of the equation format, not part of the equation number:

77 => Eq. S5

107 => Eq. M3

307 => by Eq. 18 in Ref. [67]

362 => in Eqs. 1a and 1b.

386 => below Eqs. 1a and 1b.

467 => Equation M1 describes

469 => and Eq. M2

487 => of Eq. M4

Supplementary Information mid page 5

=> [cf. Eq. M3 in the main article]

Response: Equations should be referred to using parentheses, e.g. "equation (1)", as described under **Equations** in the guidance document

<https://www.nature.com/ncomms/submit/how-to-submit>. Reading a few published articles in Nature Communications (e.g. <https://www.nature.com/articles/s41467-021-24686-5>), the abbreviated notations "Eq. (1)" and "Eqs. (1) and (2)" are used. Therefore we leave the references to equations in the text as is.

Note parentheses () are not used around figure numbers or table numbers e.g.:

117 > Figure S5

288 > (cf. Table 1)

Response: In these two cases the parentheses are not around the figure and table numbers, but around part of the sentences, in order to increase readability. We therefore like to leave these parentheses as written.

Somewhat similar thought for the reference numbers, [] are not part of the reference number but are shorthand for "reference", so "Ref. [67]" is redundant.

307 => in Ref. 67

or

307 => in [67]

Supp Info bottom of page 2:

=> (cf. Section 4 of Ref. 5)

Supp Info bottom of page 3:

=> Ref. 7

Response: Thank you. We removed the square brackets and put the reference numbers as superscripts, following the journal style.

Supp Info bottom of page 3, since "Sect." is used just once I suggest no abbreviation:

Supp Info bottom of page 3:

=> Section 6.6 of

Response: We changed to "Section".

I'm not sure if equation, eq., reference, ref., section, figure, and table should be capitalized or small. Based only on grammar it seems to me they should not be capitalized but that's a journal style question I guess.

Response: We leave the standard notation used as is. Final adjustments may be introduced by the journal during the typesetting phase.

We thank Reviewer #2 for the final comments and suggestions, and hope that the revised manuscript is now suitable for publication in Nature Communications.

Yours sincerely

Bengt Eliasson

For and on behalf of all authors

REVIEWERS' COMMENTS

Reviewer #2 (Remarks to the Author):

I thank the authors for their consideration of my comments. Their responses, and the article, are excellent.

I include a few comments on the latest changes, that might be considered by the authors. Otherwise, in my opinion the manuscript is well-suited for publication in Nature Communications.

On lines 181 and 184 in the Supplementary Information:

181> purely growing instability

184> The EM wave-driven OTSI drives two counter-propagating electrostatic Langmuir waves and a purely growing density perturbation [26,27]

It is clear what counter-propagating means, but "purely growing" is mysterious to the uninitiated. Does it mean the perturbation monotonically increases in size until the instability somehow fails? Perhaps the references explain it, but I have never noticed a comment on this. It would be a helpful thing to clarify.

185> When the driven Langmuir waves have grown to large amplitude

Does that also apply to the purely growing density

perturbation? When the density perturbation has grown enough does it contribute to the transition to nucleation and cavitating turbulence? I have had the impression that the purely growing density perturbation becomes the cavity in which the L waves are trapped. Perhaps that could be explained or clarified with an additional sentence.

In regard to the same sentence, on line 185:

185> When the driven Langmuir waves have grown to large amplitude, they can in turn undergo modulational instability and nucleation leading to localized Langmuir wave packets trapped in ion density cavities [28,29] and to strong, cavitating Langmuir turbulence [27,30,31].

Perhaps this is meant:

185 => When the driven Langmuir waves have grown to large amplitude, they may in turn drive a modulational instability ...

But I'm not sure, and not sure how to continue on to nucleation and local L wave packets, ion density cavities, etc.

On line 179 it says "When the Langmuir wave is the pump, the OTSI is a modulational instability", implying different kinds of MI.

Although the case on line 185 sounds similar to the OTSI, with an L wave as the pump, a reader may wonder what it means to "undergo modulational instability", and will also wonder what is meant by "undergo modulational instability and nucleation", and how they are connected and in turn connected to Langmuir wave packets trapped in ion density cavities, and what the difference is between an ion density cavity and an electron density cavity vs just a density cavity. And also about strong/cavitating Langmuir turbulence. Some mystery is probably unavoidable, but a little less mystery would likely be much appreciated by readers.

MI is explained on line 179 as "When the Langmuir wave is the pump, the OTSI is a modulational instability since only Langmuir waves and a density perturbation are involved."

Modulation is mentioned on line 132: "the character of the interaction may change from a three-wave interaction to a modulational interaction [15]."

Perhaps what is modulation, what is doing the modulating, and what is being modulated, can be clarified a bit more, as well as the connection to, and what is, nucleation, and the relation to ion density cavities vs other density cavities.

It's not central to the paper, but a few additional words might help. Perhaps there is a reference that can help provide an descriptive explanation, or maybe I am forgetting about some additional explanation elsewhere in the text.

At line 339 in the main text:

339> The interaction between the O mode and the Langmuir turbulence also generates a lower amplitude Z mode wave that propagates above the O mode cut-off up to the Z mode cut-off

There is a Z mode window at the Spitz angle where it is usually thought that the O mode may convert to a Z mode. Are you saying that "interaction between the O mode and the Langmuir turbulence" creates conditions that extend or recreate this window? Since the wave is directly vertically I guess this must be the case. Some additional explanation about the nature of the interaction between the O mode and the Langmuir turbulence might help to make this a bit more understandable for interested readers.

A recent relevant reference might be

Borisov (2021)

Peculiarities of the Z-mode propagation in the ionosphere

Phys. Plasmas 28, 052906

<https://www.doi.org/10.1063/5.0046369>

In the Supplementary Information, [14] is in fact the reference I had in mind (along with Thidé and Lundborg 1996) -- I had remembered it incorrectly:

287> [14] Lundborg, B. & Thidé, B. Standing wave pattern of HF radio waves in the ionospheric reflection region: 2. Applications, Radio Sci. 21(3), 486– 500 (1986). <https://doi.org/10.1029/RS021i003p00486>

A few additional details:

442> that distinguishes it

=> that distinguish it

On line 307, 67 is a superscript, probably should be normal text?

339> The interaction between the O mode and the Langmuir turbulence also generates a lower amplitude Z mode wave that propagates above the O mode cut-off up to the Z mode cut-off

My dictionary indicates it should be "cutoff" with no hyphen.

436> beat-wave

445> beatwave

My dictionary indicates that "beat wave" should be two words.

In the Supplementary Information

177> 4-wave

perhaps should be "four-wave".

181> 4 wave

also "4-wave" or "four-wave".

See:

https://www.scribendi.com/academy/articles/when_to_spell_out_numbers_in_writing.en.html

> A simple rule for using numbers in writing is that small numbers ranging from one to ten (or one to nine, depending on the style guide) should generally be spelled out. Larger numbers (i.e., above ten) are written as numerals.

183> distinct from the Langmuir driven case.

=> distinct from the Langmuir-driven case.

13 September, 2021

We thank Reviewer #2 for the final comments and suggestions. Below we list our point-by-point response to Reviewer #2's comments and our changes to the manuscript. All changes are indicated in yellow in the manuscript:

Reviewer #2 (Remarks to the Author):

I thank the authors for their consideration of my comments. Their responses, and the article, are excellent.

I include a few comments on the latest changes, that might be considered by the authors. Otherwise, in my opinion the manuscript is well-suited for publication in Nature Communications.

Response: We thank Reviewer #2 again for the insightful and constructive comments. Below we address the remaining questions.

On lines 181 and 184 in the Supplementary Information:

181> purely growing instability

184> The EM wave-driven OTSI drives two counter-propagating electrostatic Langmuir waves and a purely growing density perturbation [26,27]

It is clear what counter-propagating means, but "purely growing" is mysterious to the uninitiated. Does it mean the perturbation monotonically increases in size until the instability somehow fails? Perhaps the references explain it, but I have never noticed a comment on this. It would be a helpful thing to clarify.

Response: We have now clarified what is meant purely growing, and the difference between the PDI and OTSI starting on line 175:

"The PDI is a three-wave process in which the EM wave resonantly drives a high-frequency Langmuir wave and a low-frequency propagating IA wave which oscillates while increasing its amplitude with time. The OTSI was originally studied by Nishikawa [24] who considered a Langmuir wave as the pump wave, and was shown to be a purely growing four-wave instability in which a non-resonant density perturbation increases monotonically with time."

185> When the driven Langmuir waves have grown to large amplitude

Does that also apply to the purely growing density perturbation? When the density perturbation has grown enough does it contribute to the transition to nucleation and cavitating turbulence? I have had the impression that the purely growing density perturbation becomes the cavity in which the L waves are trapped. Perhaps that could be explained or clarified with an additional sentence.

In regard to the same sentence, on line 185:

185> When the driven Langmuir waves have grown to large amplitude, they can in turn undergo modulational instability and nucleation leading to localized Langmuir wave packets trapped in ion density cavities [28,29] and to strong, cavitating Langmuir turbulence [27,30,31].

Perhaps this is meant:

185 => When the driven Langmuir waves have grown to large amplitude, they may in turn drive a modulational instability ...

But I'm not sure, and not sure how to continue on to nucleation and local L wave packets, ion density cavities, etc.

Response: We have formulated the text to more carefully describe what takes place when the Langmuir waves have grown to finite amplitude, starting on line 188, as

“When the counter-propagating Langmuir waves have grown to large amplitude, they can in turn beat and drive monotonically growing ion density fluctuations at half their wavelength, $\lambda_L/2$. The growing ion density fluctuations saturate by nucleation with localized Langmuir wave packets trapped in ion density cavities [28,29] resulting in strong, cavitating Langmuir turbulence [27,30-32].”

New Reference:

[32] Djuth, F. T. & D. F. DuBois, Temporal development of HF-excited Langmuir and ion turbulence at Arecibo, *Earth Moon Planets* **116**, 19–53 (2015).
<https://doi.org/10.1007/s11038-015-9458-x>. Erratum: *ibid.* **116**, 139 (2015).
<https://doi.org/10.1007/s11038-015-9478-6>

On line 179 it says "When the Langmuir wave is the pump, the OTSI is a modulational instability", implying different kinds of MI.

Although the case on line 185 sounds similar to the OTSI, with an L wave as the pump, a reader may wonder what it means to "undergo modulational instability", and will also wonder what is meant by "undergo modulational instability and nucleation", and how they are connected and in turn connected to Langmuir wave packets trapped in ion density cavities, and what the difference is between an ion density cavity and an electron density cavity vs just a density cavity. And also about strong/cavitating Langmuir turbulence. Some mystery is probably unavoidable, but a little less mystery would likely be much appreciated by readers.

MI is explained on line 179 as "When the Langmuir wave is the pump, the OTSI is a modulational instability since only Langmuir waves and a density perturbation are involved."

Response: We have clarified the sentence describing the Langmuir wave case as, starting line 181:

"When the Langmuir wave is the pump, the OTSI is a modulational instability in which the Langmuir pump wave is modulated in amplitude by the slowly time-varying ion density perturbations."

Modulation is mentioned on line 132: "the character of the interaction may change from a three-wave interaction to a modulational interaction [15]."

Perhaps what is modulation, what is doing the modulating, and what is being modulated, can be clarified a bit more, as well as the connection to, and what is, nucleation, and the relation to ion density cavities vs other density cavities.

It's not central to the paper, but a few additional words might help. Perhaps there is a reference that can help provide a descriptive explanation, or maybe I am forgetting about some additional explanation elsewhere in the text.

Response: We have clarified this sentence as, starting line 133:

"the character of the interaction may change from a three-wave to a four-wave interaction where the electromagnetic wave is modulated in amplitude by the slowly time-varying ion density fluctuations [15]."

At line 339 in the main text:

339> The interaction between the O mode and the Langmuir turbulence also generates a lower amplitude Z mode wave that propagates above the O mode cut-off up to the Z mode cut-off

There is a Z mode window at the Spitze angle where it is usually thought that the O mode may convert to a Z mode. Are you saying that "interaction between the O mode and the Langmuir turbulence" creates conditions that extend or recreate this window? Since the wave is directly vertically I guess this must be the case. Some additional explanation about the nature of the interaction between the O mode and the Langmuir turbulence might help to make this a bit more understandable for interested readers.

A recent relevant reference might be

Borisov (2021)
Peculiarities of the Z-mode propagation in the ionosphere

Phys. Plasmas 28, 052906
<https://www.doi.org/10.1063/5.0046369>

Response: Thank you for this comment and the relevant reference by Borisov (2021) which we now cite. We do not think that the turbulence extends the Z mode window, but that a plausible explanation is that mode converted Langmuir waves that propagate upwards become increasingly electromagnetic and can propagate into the into the denser plasma in the form of a Z mode wave. We now write, starting line 338:

“The Z mode wave has an electrostatic resonance located slightly below the O mode cutoff [67,69] (see the Supplementary Information) where strong Langmuir turbulence takes place. An upward propagating electrostatic Langmuir wave becomes increasingly electromagnetically polarized as it passes the Z mode resonance, and can propagate into the denser plasma in the form of a Z mode.”

References:

[67] Mjølhus, E. On linear conversion in a magnetized plasma, *Radio Science* **25**(6), 1321–1339 (1990). <https://doi.org/10.1029/RS025i006p01321>

[69] Borisov, N. Peculiarities of the Z-mode propagation in the ionosphere, *Phys. Plasmas* **28**, 052906 (2021). <https://www.doi.org/10.1063/5.0046369>

In the Supplementary Information, [14] is in fact the reference I had in mind (along with Thidé and Lundborg 1996) -- I had remembered it incorrectly:

287> [14] Lundborg, B. & Thidé, B. Standing wave pattern of HF radio waves in the ionospheric reflection region: 2. Applications, *Radio Sci.* 21(3), 486– 500 (1986). <https://doi.org/10.1029/RS021i003p00486>

Response: Thank you for confirming that we cited the correct reference.

A few additional details:

442> that distinguishes it
=> that distinguish it

Response: Thank you, we corrected this.

On line 307, 67 is a superscript, probably should be normal text?

Response: We changed “67” to normal text.

339> The interaction between the O mode and the Langmuir turbulence also generates a lower amplitude Z mode wave that propagates above the O mode cut-off up to the Z mode cut-off

My dictionary indicates it should be "cutoff" with no hyphen.

Response: Thank you. We changed "cut-off" to "cutoff" everywhere.

436> beat-wave

445> beatwave

My dictionary indicates that "beat wave" should be two words.

Response: We have everywhere changed "beatwave" and "beat wave" to "beat-wave" which is a commonly used form in the literature.

In the Supplementary Information

177> 4-wave

perhaps should be "four-wave".

181> 4 wave

also "4-wave" or "four-wave".

See:

https://www.scribendi.com/academy/articles/when_to_spell_out_numbers_in_writing_en.html

> A simple rule for using numbers in writing is that small numbers ranging from one to ten (or one to nine, depending on the style guide) should generally be spelled out. Larger numbers (i.e., above ten) are written as numerals.

Response: Thank you. We everywhere changed "4-wave" and "4 wave" to "four-wave", and corresponding changes for "three-wave".

183> distinct from the Langmuir driven case.

=> distinct from the Langmuir-driven case.

Response: Thank you, we changed this.

We thank Reviewer #2 for the final comments and suggestions, and hope that the revised manuscript is now suitable for publication in Nature Communications.

Yours sincerely

Bengt Eliasson

For and on behalf of all authors